# Provable Robustness of (Graph) Neural Networks Against Data Poisoning and Backdoors

## Abstract

Generalization of machine learning models can be severely compromised by data poisoning, where adversarial changes are applied to the training data. This vulnerability has led to interest in certifying (i.e., proving) that such changes up to a certain magnitude do not affect test predictions. We, for the *first* time, certify Graph Neural Networks (GNNs) against poisoning attacks, including backdoors, targeting the node features of a given graph. Our certificates are white-box and based upon $(i)$ the *neural tangent kernel*, which characterizes the training dynamics of sufficiently wide networks; and $(ii)$ a novel reformulation of the bilevel optimization problem describing poisoning as a mixed-integer linear program. Consequently, we leverage our framework to provide fundamental insights into the role of graph structure and its connectivity on the worst-case robustness behavior of convolution-based and PageRank-based GNNs. We note that our framework is more general and constitutes the *first* approach to derive white-box poisoning certificates for NNs, which can be of independent interest beyond graph-related tasks.

## 1 Introduction

Numerous works showcase the vulnerability of modern machine learning models to data poisoning, where adversarial changes are made to the training data (Biggio et al., 2012; Muñoz-González et al., 2017; Zügner & Günnemann, 2019a; Wan et al., 2023), as well as backdoor attacks affecting both training and test sets (Goldblum et al., 2023). Empirical defenses against such threats are continually at risk of being compromised by future attacks (Koh et al., 2022; Suciu et al., 2018). This motivates the development of *robustness certificates*, which provide formal guarantees that the prediction for a given test data point remains unchanged under an assumed perturbation model.

Robustness certificates can be categorized as providing deterministic or probabilistic guarantees, and as being white box, i.e. developed for a particular model, or black box (model-agnostic). While each approach has its strengths and applications (Li et al., 2023), we focus on *white-box* certificates as they can provide a more direct understanding into the worst-case robustness behavior of commonly used models and architectural choices (Tjeng et al., 2019; Mao et al., 2024; Banerjee et al., 2024). The literature on poisoning certificates is less developed than certifying against test-time (evasion) attacks and we provide an overview and categorization in Table 1. Notably, white-box certificates are currently available only for decision trees (Drews et al., 2020), nearest neighbor algorithms (Jia et al., 2022), and naive Bayes classification (Bian et al., 2024). In the case of Neural Networks (NNs), the main challenge in white-box poisoning certification comes from capturing their complex training dynamics. As a result, the current literature reveals that deriving white-box poisoning certificates for NNs, and by extension Graph Neural Networks (GNNs), is still an *unsolved* problem, raising the question if such certificates can at all be practically computed.

In this work, we give a positive answer to this question by developing the first approach towards white-box certification of NNs against data poisoning and backdoor attacks, and instantiate it for common convolution-based and PageRank-based GNNs. Concretely, poisoning can be modeled as a bilevel optimization problem over the training data $\mathcal{D}$ that includes training on $\mathcal{D}$ as its inner subproblem. To overcome the challenge of capturing the complex training dynamics of NNs, we consider the Neural Tangent Kernel (NTK) that characterizes the training dynamics of sufficiently wide NNs under gradient flow (Jacot et al., 2018; Arora et al., 2019). In particular, we leverage the equivalence between NNs trained using the soft-margin loss and standard soft-margin Support Vector Machines (SVMs)

Table 1: Representative selection of data poisoning and backdoor attack certificates. To the best of our knowledge, it contains all white-box works. Our work presents the *first* white-box certificate applicable to *(graph) neural networks* and *Support Vector Machines* (SVMs). Poisoning refers to (purely) training-time attacks. A backdoor attack refers to joint training-time and test-time perturbations. Certificates apply to different attack types: $(i)$ Clean-label: modifies the features of the training data; $(ii)$ Label-flipping: modifies the labels of the training data; $(iii)$ Joint: modifies both features and labels; $(iv)$ General attack: allows (arbitrary) insertion/deletion, i.e., like $(iii)$ but dataset size doesn't need to be constant; $(v)$ Node injection: particular to graph learning, refers to adding nodes with arbitrary features and malicious edges into the graph. It is most related to $(iv)$ but does not allow deletion and can't be compared with $(i)$ and $(ii)$. Note that certificates that only certify against $(iii) - (v)$ cannot certify against clean-label or label-flipping attacks individually.

| | | Deterministic | Certified Models | Pois. | Backd. | Attack Type | Applies to Node Cls. | Approach |
|---|---|---|---|---|---|---|---|---|
| | | | | | | Perturbation Model | | |
| (Ma et al., 2019) | | ✗ | Diff. Private Learners | ✓ | ✗ | Joint | ✗ | Differential Privacy |
| (Liu et al., 2023) | | ✗ | Diff. Private Learners | ✓ | ✗ | General | ✗ | Differential Privacy |
| (Wang et al., 2020) | | ✗ | Smoothed Classifier | ✗ | ✓ | Joint | ✗ | Randomized Smoothing |
| (Weber et al., 2023) | Black Box | ✗ | Smoothed Classifier | ✗ | ✓ | Clean-label | ✗ | Randomized Smoothing |
| (Zhang et al., 2022) | | ✗ | Smoothed Classifier | ✓ | ✓ | Joint | ✗ | Randomized Smoothing |
| (Lai et al., 2024) | | ✗ | Smoothed Classifier | ✓ | ✗ | Node Injection | ✓ | Randomized Smoothing |
| (Jia et al., 2021) | | ✗ | Ensemble Classifier | ✓ | ✗ | General | ✗ | Ensemble (Majority Vote) |
| (Rosenfeld et al., 2020) | | ✓ | Smoothed Classifier | ✓ | ✗ | Label Flip. | ✗ | Randomized Smoothing |
| (Levine & Feizi, 2021) | | ✓ | Ensemble Classifier | ✓ | ✗ | Label Flip./General | ✗ | Ensemble (Majority Vote) |
| (Wang et al., 2022) | | ✓ | Ensemble Classifier | ✓ | ✗ | General | ✗ | Ensemble (Majority Vote) |
| (Rezaei et al., 2023) | | ✓ | Ensemble Classifier | ✓ | ✗ | General | ✗ | Ensemble (Run-Off Election) |
| (Drews et al., 2020) | | ✓ | Decision Trees | ✓ | ✗ | General | ✗ | Abstract Interpretation |
| (Meyer et al., 2021) | White Box | ✓ | Decision Trees | ✓ | ✗ | General | ✗ | Abstract Interpretation |
| (Jia et al., 2022) | | ✓ | k-Nearest Neighbors | ✓ | ✗ | General | ✗ | Majority Vote |
| (Bian et al., 2024) | | ✓ | Naive Bayes Classifier | ✓ | ✗ | Clean-label | ✗ | Algorithmic |
| **Ours** | | ✓ | **NNs & SVMs** | ✓ | ✓ | Clean-label | ✓ | NTK & Linear Programming |

with the NN's NTKs as kernel matrix (Chen et al., 2021). Using this equivalence, we introduce a novel reformulation of the bilevel optimization problem as a mixed-integer linear program (MILP) that allows to certify test datapoints against poisoning as well as backdoor attacks for sufficiently wide NNs (see Fig. 1). Although our framework applies to wide NNs in general, solving the MILP scales with the number of labeled training samples. Thus, it is a natural fit for semi-supervised learning tasks, where one can take advantage of the low labeling rate. In this context, we focus on semi-supervised node classification in graphs, where certifying against node feature perturbations is particularly challenging due to the interconnectivity between nodes (Zügner & Günnemann, 2019b; Scholten et al., 2023). Here, our framework provides a general and elegant way to handle this interconnectivity inherent to graph learning, by using the corresponding graph NTKs (Sabanayagam et al., 2023) of various GNNs. Our **contributions** are:

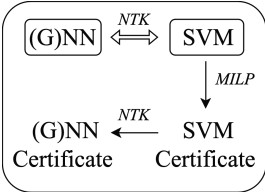

Figure 1: Illustration of our poisoning certificate.

**(i)** We are the first to certify GNNs in node-classification tasks against poisoning and backdoor attacks targeting node features. Our certification framework called QPCert is introduced in Sec. 3 and leverages the NTK to capture the complex training-dynamics of GNNs. Further, it can be applied to NNs in general and thus, it represents the first approach on *white-box* poisoning certificates for NNs.

**(ii)** Enabled by the white-box nature of our certificate, we conduct the first study into the role of graph data and architectural choices on the worst-case robustness of many widely used GNNs against data poisoning and backdoor attacks (see Sec. 4). We focus on convolution-based and PageRank-based architectures and contribute the derivation of the closed-form NTK for APPNP (Gasteiger et al., 2019), GIN (Xu et al., 2018), and GraphSAGE (Hamilton et al., 2017) in App. B.

**(iii)** We contribute a reformulation of the bilevel optimization problem describing poisoning as a MILP when instantiated with kernelized SVMs, allowing for white-box certification of SVMs. While we focus on the NTK as kernel, our strategy can be transferred to arbitrary kernel choices.

**Notation.** We represent matrices and vectors with boldfaced upper and lowercase letters, respectively. $v_i$ and $M_{ij}$ denote $i$-th and $ij$-th entries of $\mathbf{v}$ and $\mathbf{M}$, respectively. $i$-th row of $\mathbf{M}$ is $\mathbf{M}_i$. $\mathbf{I}_n$ is the identity matrix and $\mathbf{1}_{n \times n}$ is the matrix of all 1s of size $n \times n$. We use $\langle ., . \rangle$ for scalar product, $\|.\|_2$ for vector Euclidean norm and matrix Frobenius norm, $\mathbb{1}[.]$ for indicator function, $\odot$ for the Hadamard product, $\mathbb{E}[.]$ for expectation, and $\lceil z \rceil$ for the smallest integer $\geq z$ (ceil). $[n]$ denotes $\{1, 2, \ldots, n\}$.

## 2    PRELIMINARIES

We are given a partially-labeled graph $\mathcal{G} = (\boldsymbol{S}, \boldsymbol{X})$ with $n$ nodes and a graph structure matrix $\boldsymbol{S} \in \mathbb{R}_{\geq 0}^{n \times n}$, representing for example, a normalized adjacency matrix. Each node $i \in [n]$ has features $\boldsymbol{x}_i \in \mathbb{R}^d$ of dimension $d$ collected in a node feature matrix $\boldsymbol{X} \in \mathbb{R}^{n \times d}$. We assume labels $y_i \in \{1, \ldots, K\}$ are given for the first $m \leq n$ nodes. Our goal is to perform node classification, either in a transductive setting where the labels of the remaining $n - m$ nodes should be inferred, or in an inductive setting where newly added nodes at test time should be classified. The set of labeled nodes is denoted $\mathcal{V}_L$ and the set of unlabeled nodes $\mathcal{V}_U$.

**Perturbation model.** We assume that at training time the adversary $\mathcal{A}$ has control over the features of an $\epsilon$-fraction of nodes and that $\lceil (1 - \epsilon)n \rceil$ nodes are clean. For backdoor attacks, the adversary can also change the features of a test node of interest. Following the *semi-verified learning* setup introduced in (Charikar et al., 2017), we assume that $k < n$ nodes are known to be uncorrupted. We denote the verified nodes by set $\mathcal{V}_V$ and the nodes that can be potentially corrupted as set $\mathcal{U}$. We further assume that the strength of $\mathcal{A}$ to poison training or modify test nodes is bounded by a budget $\delta \in \mathbb{R}_+$. More formally, $\mathcal{A}$ can choose a perturbed $\tilde{\boldsymbol{x}}_i \in \mathcal{B}_p(\boldsymbol{x}_i) := \{\tilde{\boldsymbol{x}} \mid \|\tilde{\boldsymbol{x}} - \boldsymbol{x}_i\|_p \leq \delta\}$ for each node $i$ under control. We denote the set of all perturbed node feature matrices constructible by $\mathcal{A}$ from $\boldsymbol{X}$ as $\mathcal{A}(\boldsymbol{X})$ and $\mathcal{A}(\mathcal{G}) = \{(\boldsymbol{S}, \tilde{\boldsymbol{X}}) \mid \tilde{\boldsymbol{X}} \in \mathcal{A}(\boldsymbol{X})\}$. In data poisoning, the *goal* of $\mathcal{A}$ is to maximize misclassification in the test nodes. For backdoor attacks $\mathcal{A}$ aims to induce misclassification only in test nodes that it controls.

**Learning setup.** GNNs are functions $f_\theta$ with (learnable) parameters $\theta \in \mathbb{R}^q$ and $L$ number of layers taking the graph $\mathcal{G} = (\boldsymbol{S}, \boldsymbol{X})$ as input and outputting a prediction for each node. We consider linear output layers with weights $\boldsymbol{W}^{L+1}$ and denote by $f_\theta(\mathcal{G})_i \in \mathbb{R}^K$ the (unnormalized) logit output associated to node $i$. Note for binary classification $f_\theta(\mathcal{G})_i \in \mathbb{R}$. We define the architectures such as MLP, GCN (Kipf & Welling, 2017), SGC (Wu et al., 2019), (A)PPNP (Gasteiger et al., 2019) and others in App. A. We focus on binary classes $y_i \in \{\pm 1\}$ and refer to App. E for the multi-class case. Following Chen et al. (2021), the parameters $\theta$ are learned using the soft-margin loss

$$\mathcal{L}(\theta, \mathcal{G}) = \min_\theta \frac{1}{2} \|\mathbf{W}^{(L+1)}\|_2^2 + C \sum_{i=1}^m \max(0, 1 - y_i f_\theta(\mathcal{G})_i) \tag{1}$$

where the second term is the Hinge loss weighted by a regularization $C \in \mathbb{R}_+$. Note that due to its non-differentiability, the NN is trained by subgradient descent. Furthermore, we consider NTK parameterization (Jacot et al., 2018) in which parameters $\theta$ are initialized from a standard Gaussian $\mathcal{N}(0, 1/\text{width})$. Under NTK parameterization and sufficiently large width limit, the training dynamics of $f_\theta(\mathcal{G})$ are precisely characterized by the NTK defined between nodes $i$ and $j$ as $Q_{ij} = \boldsymbol{Q}(\mathbf{x}_i, \mathbf{x}_j) = \mathbb{E}_\theta[\langle \nabla_\theta f_\theta(\mathcal{G})_i, \nabla_\theta f_\theta(\mathcal{G})_j \rangle] \in \mathbb{R}$.

**Equivalence of NN to soft-margin SVM with NTK.** Chen et al. (2021) show that training NNs in the infinite-width limit with Eq. (1) is equivalent to training a soft-margin SVM with (sub)gradient descent using the NN's NTK as kernel. Thus, both methods converge to the same solution. We extend this equivalence to GNNs, as detailed in App. C. More formally, let the SVM be defined as $f_\theta(\mathcal{G})_i = f_\theta^{SVM}(\boldsymbol{x}_i) = \langle \boldsymbol{\beta}, \Phi(\boldsymbol{x}_i) \rangle$ where $\Phi(\cdot)$ is the feature transformation associated to the used kernel and $\theta = \boldsymbol{\beta}$ are the learnable parameters obtained by minimizing $\mathcal{L}(\theta, \mathcal{G})$. Following Chen et al. (2021), we do not include a bias term. To find the optimal $\beta^*$, instead of minimizing Eq. (1) with (sub)gradient descent, we work with the equivalent dual

$$\mathrm{P}_1(\boldsymbol{Q}): \min_{\boldsymbol{\alpha}} - \sum_{i=1}^m \alpha_i + \frac{1}{2} \sum_{i=1}^m \sum_{j=1}^m y_i y_j \alpha_i \alpha_j Q_{ij} \text{ s.t. } 0 \leq \alpha_i \leq C \ \forall i \in [m] \tag{2}$$

with the Lagrange multipliers $\boldsymbol{\alpha} \in \mathbb{R}^m$ and kernel $Q_{ij} = \boldsymbol{Q}(\boldsymbol{x}_i, \boldsymbol{x}_j) \in \mathbb{R}$ computed between all labeled nodes $i \in [m]$. and $j \in [m]$. The optimal dual solution may not be unique and we denote by $\mathcal{S}(\boldsymbol{Q})$ the set of $\boldsymbol{\alpha}$ solving $\mathrm{P}_1(\boldsymbol{Q})$. However, any $\boldsymbol{\alpha}^* \in \mathcal{S}(\boldsymbol{Q})$ corresponds to the same unique $\beta^* = \sum_{i=1}^m y_i \alpha_i^* \Phi(\mathcal{G})_i$ minimizing Eq. (1) (Burges & Crisp, 1999). Thus, the prediction of the SVM for a test node $t$ using the dual is given by $f_\theta^{SVM}(\mathbf{x}_t) = \sum_{i=1}^m y_i \alpha_i^* Q_{ti}$ for any $\boldsymbol{\alpha}^* \in \mathcal{S}(\boldsymbol{Q})$, where $Q_{ti}$ is the kernel between a test node $t$ and training node $i$. By choosing $\boldsymbol{Q}$ to be the NTK of a GNN $f_\theta$, the prediction equals $f_\theta(\mathcal{G})_t$ if the width of the GNN's hidden layers goes to infinity. Thus, a certificate for the SVM directly translates to a certificate for infinitely-wide GNNs. In the finite-width

case, where the smallest GNN's layer width is $h$, the output difference between both methods can be bounded with high probability by $\mathcal{O}(\frac{\ln h}{\sqrt{h}})$ (the probability $\to 1$ as $h \to \infty$). Thus, the certificate translates to a high probability guarantee for sufficiently wide finite networks.

# 3 QPCERT: OUR CERTIFICATION FRAMEWORK

Poisoning a clean training graph $\mathcal{G}$ can be described as a bilevel problem where an adversary $\mathcal{A}$ tries to find a perturbed $\widetilde{\mathcal{G}} \in \mathcal{A}(\mathcal{G})$ that results in a model $\theta$ minimizing an attack objective $\mathcal{L}_{att}(\theta, \widetilde{\mathcal{G}})$:

$$\min_{\widetilde{\mathcal{G}}, \theta} \mathcal{L}_{att}(\theta, \widetilde{\mathcal{G}}) \quad \text{s. t.} \quad \widetilde{\mathcal{G}} \in \mathcal{A}(\mathcal{G}) \ \wedge \ \theta \in \arg\min_{\theta'} \mathcal{L}(\theta', \widetilde{\mathcal{G}}) \tag{3}$$

Eq. (3) is called an upper-level problem and $\min_{\theta'} \mathcal{L}(\theta', \widetilde{\mathcal{G}})$ the lower-level problem. Now, a sample-wise poisoning certificate can be obtained by solving Eq. (3) with an $\mathcal{L}_{att}(\theta, \widetilde{\mathcal{G}})$ chosen to describe if the prediction for a test node $t$ changes compared to the prediction of a model trained on the clean graph. However, this approach is challenging as even the simplest bilevel problems given by a linear lower-level problem embedded in an upper-level linear problem are NP-hard (Jeroslow, 1985). Thus, in this section, we develop a general methodology to reformulate the bilevel (sample-wise) certification problem for kernelized SVMs as a mixed-integer linear program, making certification tractable through the use of highly efficient modern MILP solvers such as Gurobi (Gurobi Optimization, LLC, 2023) or CPLEX (Cplex, 2009). Our approach can be divided into three steps: **(1)** The bilevel problem is reduced to a single-level problem by exploiting properties of the quadratic dual $P_1(\mathbf{Q})$; **(2)** We model $\widetilde{\mathcal{G}} \in \mathcal{A}(\mathcal{G})$ by assuming a bound on the effect any $\widetilde{\mathcal{G}}$ can have on the elements of the kernel $\mathbf{Q}$. This introduces a relaxation of the bilevel problem from Eq. (3) and allows us to fully express certification as a MILP; **(3)** In Sec. 3.1, we choose the NTK of different GNNs as kernel and develop bounds on the kernel elements to use in the certificate. In the following, we present our certificate for binary classification where $y_i \in \{\pm 1\} \ \forall i \in [n]$ and transductive learning, where the test node is already part of $\mathcal{G}$. We generalize it to a multi-class and inductive setting in App. E.

**A single-level reformulation.** Given an SVM $f_\theta^{SVM}$ trained on the clean graph $\mathcal{G}$, its class prediction for a test node $t$ is given by $\text{sgn}(\hat{p}_t) = \text{sgn}(f_\theta^{SVM}(\mathbf{x}_t))$. If for all $\widetilde{\mathcal{G}} \in \mathcal{A}(\mathcal{G})$ the sign of the prediction does not change if the SVM should be retrained on $\widetilde{\mathcal{G}}$, then we know that the prediction for $t$ is certifiably robust. Thus, the attack objective reads $\mathcal{L}_{att}(\theta, \widetilde{\mathcal{G}}) = \text{sgn}(\hat{p}_t) \sum_{i=1}^m y_i \alpha_i \widetilde{Q}_{ti}$, where $\widetilde{Q}_{ti}$ denotes the kernel computed between nodes $t$ and $i$ on the perturbed graph $\widetilde{\mathcal{G}}$, and indicates robustness if greater than zero. Now, notice that the perturbed graph $\widetilde{\mathcal{G}}$ only enters the training objective Eq. (2) through values of the kernel matrix $\widetilde{\mathbf{Q}} \in \mathbb{R}^{n \times n}$. Thus, we introduce the set $\mathcal{A}(\mathbf{Q})$ of all kernel matrices $\widetilde{\mathbf{Q}}$, constructable from $\widetilde{\mathcal{G}} \in \mathcal{A}(\mathcal{G})$. Furthermore, we denote with $\mathcal{S}(\widetilde{\mathbf{Q}})$ the optimal solution set to $P_1(\widetilde{\mathbf{Q}})$. As a result, we can rewrite Eq. (3) for kernelized SVMs as

$$P_2(\mathbf{Q}) : \min_{\boldsymbol{\alpha}, \widetilde{\mathbf{Q}}} \text{sgn}(\hat{p}_t) \sum_{i=1}^m y_i \alpha_i \widetilde{Q}_{ti} \quad s.t. \quad \widetilde{\mathbf{Q}} \in \mathcal{A}(\mathbf{Q}) \ \wedge \ \boldsymbol{\alpha} \in \mathcal{S}(\widetilde{\mathbf{Q}}) \tag{4}$$

and certify robustness if the optimal solution to $P_2(\mathbf{Q})$ is greater than zero. Crucial in reformulating $P_2(\mathbf{Q})$ into a single-level problem are the Karush–Kuhn–Tucker (KKT) conditions of the lower-level problem $P_1(\widetilde{\mathbf{Q}})$. Concretely, the KKT conditions of $P_1(\widetilde{\mathbf{Q}})$ are

$$\forall i \in [m]: \quad \sum_{j=1}^m y_i y_j \alpha_j \widetilde{Q}_{ij} - 1 - u_i + v_i = 0 \qquad \text{(Stationarity)} \tag{5}$$

$$\alpha_i \geq 0, \ C - \alpha_i \geq 0, \ u_i \geq 0, \ v_i \geq 0 \qquad \text{(Primal and Dual feasibility)} \tag{6}$$

$$u_i \alpha_i = 0, \ v_i(C - \alpha_i) = 0 \qquad \text{(Complementary slackness)} \tag{7}$$

where $\mathbf{u} \in \mathbb{R}^m$ and $\mathbf{v} \in \mathbb{R}^m$ are Lagrange multipliers. Now, we can state (see App. F for the proof):

**Proposition 1.** *Problem* $P_1(\widetilde{\mathbf{Q}})$ *given by Eq. (2) is convex and satisfies strong Slater's constraint. Consequently, the single-level optimization problem* $P_3(\mathbf{Q})$ *arising from* $P_2(\mathbf{Q})$ *by replacing* $\boldsymbol{\alpha} \in \mathcal{S}(\widetilde{\mathbf{Q}})$ *with Eqs. (5) to (7) has the same globally optimal solutions as* $P_2(\mathbf{Q})$.

**A mixed-integer linear reformulation.** The computational bottleneck of $P_3(\widetilde{Q})$ are the non-linear product terms between continuous variables in the attack objective as well as in Eqs. (5) and (7), making $P_3(\widetilde{Q})$ a bilinear problem. Thus, we describe in the following how $P_3(\widetilde{Q})$ can be transformed into a MILP. First, the complementary slackness constraints can be linearized by recognizing that they have a combinatorial structure. In particular, $u_i = 0$ if $\alpha_i > 0$ and $v_i = 0$ if $\alpha_i < C$. Thus, introducing binary integer variables $\mathbf{s}$ and $\mathbf{t} \in \{0,1\}^m$, we reformulate the constraints in Eq. (7) with big-$M$ constraints as

$$\forall i \in [m]: \quad u_i \le M_{u_i} s_i, \ \alpha_i \le C(1-s_i), \ s_i \in \{0,1\}, \tag{8}$$
$$v_i \le M_{v_i} t_i, \ C - \alpha_i \le C(1-t_i), \ t_i \in \{0,1\}$$

where $M_{u_i}$ and $M_{v_i}$ are positive constants. In general, verifying that a certain choice of big-$Ms$ results in a valid (mixed-integer) reformulation of the complementary constraints Eq. (7), i.e., such that no optimal solution to the original bilevel problem is cut off, is at least as hard as solving the bilevel problem itself (Kleinert et al., 2020). This is problematic as heuristic choices can lead to suboptimal solutions to the original problem (Pineda & Morales, 2019). However, additional structure provided by $P_1(\widetilde{Q})$ and $P_3(Q)$ together with insights into the optimal solution set allows us to derive valid and small $M_{u_i}$ and $M_{v_i}$ for all $i \in [m]$.

Concretely, the adversary $\mathcal{A}$ can only make a bounded change to $\mathcal{G}$. Thus, the element-wise difference of any $\widetilde{Q} \in \mathcal{A}(Q)$ to $Q$ will be bounded. As a result, there exist element-wise upper and lower bounds $\widetilde{Q}_{ij}^L \le \widetilde{Q}_{ij} \le \widetilde{Q}_{ij}^U$ for all $i,j \in [m] \cup \{t\}$ and valid for any $\widetilde{Q} \in \mathcal{A}(Q)$. In Sec. 3.1 we derive concrete lower and upper bounds for the NTKs corresponding to different common GNNs. This, together with $0 \le \alpha_i \le C$, allows us to lower and upper bound $\sum_{j=1}^m y_i y_j \alpha_j \widetilde{Q}_{ij}$ in Eq. (5). Now, given an optimal solution $(\boldsymbol{\alpha}^*, \widetilde{Q}^*, \mathbf{u}^*, \mathbf{v}^*)$ to $P_3(Q)$, observe that either $u_i^*$ or $v_i^*$ are zero, or can be freely varied between any positive values as long as Eq. (5) is satisfied without changing the objective value or any other variable. As a result, one can use the lower and upper bounds on $\sum_{j=1}^m y_i y_j \alpha_j \widetilde{Q}_{ij}$ to find the minimal value range necessary and sufficient for $u_i$ and $v_i$, such that Eq. (5) can always be satisfied for any $\boldsymbol{\alpha}^*$ and $\widetilde{Q}^*$. Consequently, only redundant solutions regarding large $u_i^*$ and $v_i^*$ will be cut off and the optimal solution value stays the same as for $P_3(Q)$, not affecting the certification. The exact $M_{u_i}$ and $M_{v_i}$ depend on the signs of the involved $y_i$ and $y_j$ and are derived in App. G.

Now, the remaining non-linearities come from the product terms $\alpha_i \widetilde{Q}_{ij}$. We approach this by first introducing new variables $Z_{ij}$ for all $i,j \in [m] \cup \{t\}$ and set $Z_{ij} = \alpha_j \widetilde{Q}_{ij}$. Then, we replace all product terms $\alpha_j \widetilde{Q}_{ij}$ in Eq. (5) and in the objective in Eq. (4) with $Z_{ij}$. This alone has not changed the fact that the problem is bilinear, only that the bilinear terms have now moved to the definition of $Z_{ij}$. However, we have access to lower and upper bounds on $\widetilde{Q}_{ij}$. Thus, replacing $Z_{ij} = \alpha_j \widetilde{Q}_{ij}$ with linear constraints $Z_{ij} \le \alpha_j \widetilde{Q}_{ij}^U$ and $Z_{ij} \ge \alpha_j \widetilde{Q}_{ij}^L$ results in a relaxation to $P_3(Q)$. This resolved all non-linearities and we can write the following theorem.

**Theorem 1** (MILP Formulation). *Node $t$ is certifiably robust against adversary $\mathcal{A}$ if the optimal solution to the following MILP denoted by $P(Q)$ is greater than zero*

$$\min_{\boldsymbol{\alpha}, \mathbf{u}, \mathbf{v}, \mathbf{s}, \mathbf{t}, \mathbf{Z}} \text{sgn}(\hat{p}_t) \sum_{i=1}^m y_i Z_{ti} \quad s.t.$$

$$Z_{ij} \le \alpha_j \widetilde{Q}_{ij}^U, \ Z_{ij} \ge \alpha_j \widetilde{Q}_{ij}^L \qquad \forall i \in [m] \cup \{t\}, j \in [m]$$

$$\forall i \in [m]: \quad \sum_{j=1}^m y_i y_j Z_{ij} - 1 - u_i + v_i = 0, u_i \le M_u s_i, \ \alpha_i \le C(1-s_i), \ s_i \in \{0,1\},$$

$$\alpha_i \ge 0, \ C - \alpha_i \ge 0, \ u_i \ge 0, \ v_i \ge 0, v_i \le M_v t_i, \ C - \alpha_i \le C(1-t_i), \ t_i \in \{0,1\}$$

$P(Q)$ includes backdoor attacks through the bounds $\widetilde{Q}_{tj}^L$ and $\widetilde{Q}_{tj}^U$ for all $j \in [m]$, which for an adversary $\mathcal{A}$ who can manipulate $t$ will be set different. On computational aspects, $P(Q)$ involves $(m+1)^2 + 5m$ variables out of which $2m$ are binary. Thus, the number of binary variables, which for a particular problem type mainly defines how long it takes MILP-solvers to solve a problem, scales with the number of labeled samples.

### 3.1 QPCert for GNNs through their corresponding NTKs

To certify a specific GNN using our QPCert framework, we need to derive element-wise lower and upper bounds valid for all NTK matrices $\widetilde{Q} \in \mathcal{A}(Q)$ of the corresponding network, that are constructable by the adversary. As a first step, we introduce the NTKs for the GNNs of interest before deriving the bounds. While Sabanayagam et al. (2023) provides the NTKs for GCN and SGC with and without skip connections, we derive the NTKs for (A)PPNP, GIN and GraphSAGE in App. B. For clarity, we present the NTKs for $f_\theta(\mathcal{G})$ with hidden layers $L = 1$ here and the general case for any $L$ in the appendix. For $L = 1$, the NTKs generalize to the form $Q = \mathbf{M}(\boldsymbol{\Sigma} \odot \dot{\mathbf{E}})\mathbf{M}^T + \mathbf{M}\mathbf{E}\mathbf{M}^T$ for all

Table 2: The NTKs of GNNs have the general form $Q = \mathbf{M}(\boldsymbol{\Sigma} \odot \dot{\mathbf{E}})\mathbf{M}^T + \mathbf{M}\mathbf{E}\mathbf{M}^T$ for $L = 1$. The definitions of $\mathbf{M}, \boldsymbol{\Sigma}, \mathbf{E}$ and $\dot{\mathbf{E}}$ are given in the table. $\mathbf{Z} = \mathbf{S} + \mathbf{I}_n$ and $\mathbf{T} = ((1 + \epsilon)\mathbf{I}_n + \mathbf{A})\mathbf{X}$. $\kappa_0(z) = \frac{1}{\pi}(\pi - \arccos(z))$ and $\kappa_1(z) = \frac{1}{\pi}(z(\pi - \arccos(z)) + \sqrt{1 - z^2})$.

| GNN | M | $\boldsymbol{\Sigma}$ | $E_{ij}$ | $\dot{E}_{ij}$ |
|---|---|---|---|---|
| GCN | S | $\mathbf{S}\mathbf{X}\mathbf{X}^T\mathbf{S}^T$ | $\sqrt{\Sigma_{ii}\Sigma_{jj}}\kappa_1\left(\frac{\Sigma_{ij}}{\sqrt{\Sigma_{ii}\Sigma_{jj}}}\right)$ | $\kappa_0\left(\frac{\Sigma_{ij}}{\sqrt{\Sigma_{ii}\Sigma_{jj}}}\right)$ |
| SGC | S | $\mathbf{S}\mathbf{X}\mathbf{X}^T\mathbf{S}^T$ | $\Sigma_{ij}$ | $1$ |
| GraphSAGE | Z | $\mathbf{Z}\mathbf{X}\mathbf{X}^T\mathbf{Z}^T$ | $\sqrt{\Sigma_{ii}\Sigma_{jj}}\kappa_1\left(\frac{\Sigma_{ij}}{\sqrt{\Sigma_{ii}\Sigma_{jj}}}\right)$ | $\kappa_0\left(\frac{\Sigma_{ij}}{\sqrt{\Sigma_{ii}\Sigma_{jj}}}\right)$ |
| (A)PPNP | P | $\mathbf{X}\mathbf{X}^T + \mathbf{1}_{n \times n}$ | $\sqrt{\Sigma_{ii}\Sigma_{jj}}\kappa_1\left(\frac{\Sigma_{ij}}{\sqrt{\Sigma_{ii}\Sigma_{jj}}}\right)$ | $\kappa_0\left(\frac{\Sigma_{ij}}{\sqrt{\Sigma_{ii}\Sigma_{jj}}}\right)$ |
| GIN | $\mathbf{I}_n$ | $\mathbf{T}\mathbf{T}^T + \mathbf{1}_{n \times n}$ | $\sqrt{\Sigma_{ii}\Sigma_{jj}}\kappa_1\left(\frac{\Sigma_{ij}}{\sqrt{\Sigma_{ii}\Sigma_{jj}}}\right)$ | $\kappa_0\left(\frac{\Sigma_{ij}}{\sqrt{\Sigma_{ii}\Sigma_{jj}}}\right)$ |
| MLP | $\mathbf{I}_n$ | $\mathbf{X}\mathbf{X}^T + \mathbf{1}_{n \times n}$ | $\sqrt{\Sigma_{ii}\Sigma_{jj}}\kappa_1\left(\frac{\Sigma_{ij}}{\sqrt{\Sigma_{ii}\Sigma_{jj}}}\right)$ | $\kappa_0\left(\frac{\Sigma_{ij}}{\sqrt{\Sigma_{ii}\Sigma_{jj}}}\right)$ |

the networks, with the definitions of $\mathbf{M}, \boldsymbol{\Sigma}, \mathbf{E}$ and $\dot{\mathbf{E}}$ detailed in Table 2. Thus, it is important to note that the effect of the feature matrix $\mathbf{X}$, which the adversary can manipulate, enters into the NTK only as a product $\mathbf{X}\mathbf{X}^T$, making this the quantity of interest when bounding the NTK matrix.

Focusing on $p = \{1, 2, \infty\}$ in the perturbation model $\mathcal{B}_p(\mathbf{x})$ and $\widetilde{\mathbf{X}} \in \mathcal{A}(\mathbf{X})$, we first derive the bounds for $\widetilde{\mathbf{X}}\widetilde{\mathbf{X}}^T$ by considering $\mathcal{U} := \{i : i \notin \mathcal{V}_V\}$ to be the set of all unverified nodes that the adversary can potentially control. Particularly, we present the worst-case element-wise lower and upper bounds for $\widetilde{\mathbf{X}}\widetilde{\mathbf{X}}^T = \mathbf{X}\mathbf{X}^T + \Delta$ in terms of $\Delta$ in Lemma 1, and Lemmas 2 and 3 in App. D.

**Lemma 1** (Bounds for $\Delta$, $p = \infty$). *Given $\mathcal{B}_\infty(\mathbf{x})$ and any $\widetilde{\mathbf{X}} \in \mathcal{A}(\mathbf{X})$, then $\widetilde{\mathbf{X}}\widetilde{\mathbf{X}}^T = \mathbf{X}\mathbf{X}^T + \Delta$ where the worst-case bounds for $\Delta$, $\Delta_{ij}^L \leq \Delta_{ij} \leq \Delta_{ij}^U$ for all $i$ and $j \in [n]$, is*

$$\Delta_{ij}^L = -\delta\|\mathbf{X}_j\|_1 \mathbb{1}[i \in \mathcal{U}] - \delta\|\mathbf{X}_i\|_1 \mathbb{1}[j \in \mathcal{U}] - \delta^2 d\mathbb{1}[i \in \mathcal{U} \wedge j \in \mathcal{U} \wedge i \neq j]$$
$$\Delta_{ij}^U = \delta\|\mathbf{X}_j\|_1 \mathbb{1}[i \in \mathcal{U}] + \delta\|\mathbf{X}_i\|_1 \mathbb{1}[j \in \mathcal{U}] + \delta^2 d\mathbb{1}[i \in \mathcal{U} \wedge j \in \mathcal{U}] \qquad (9)$$

The NTK bounds $\widetilde{Q}_{ij}^L$ and $\widetilde{Q}_{ij}^U$, are now derived by simply propagating the bounds for $\widetilde{\mathbf{X}}\widetilde{\mathbf{X}}^T$ through the NTK formulation since the multipliers and addends are positive. To elaborate, we compute $\widetilde{Q}_{ij}^L$ by substituting $\mathbf{X}\mathbf{X}^T = \mathbf{X}\mathbf{X}^T + \Delta^L$, and likewise for $\widetilde{Q}_{ij}^U$. Only bounding $E_{ij}$ and $\dot{E}_{ij}$ needs special care and our respective approach is discussed in App. D.1. Further, we prove that the bounds are tight in the worst-case in App. D.2.

**Theorem 2** (NTK bounds are tight). *The worst-case NTK bounds are tight for GNNs with linear activations such as SGC and (A)PPNP, and MLP with $\sigma(z) = z$ for $p = \{1, 2, \infty\}$ in $\mathcal{B}_p(\mathbf{x})$.*

## 4 Experimental results

We present $(i)$ the effectiveness of QPCert in certifying different GNNs using their corresponding NTKs against node feature poisoning and backdoor attacks; $(ii)$ insights into the role of graph data in worst-case robustness of GNNs, specifically the importance of graph information and its connectivity; $(iii)$ a study of the impact of different architectural components in GNNs on their provable robustness.

**Dataset.** We use the real-world graph dataset *Cora-ML* (Bojchevski & Günnemann, 2018), where we generate continuous 384-dim. embeddings of the abstracts with a modern sentence transformer[1]. Furthermore, for binary classification, we use Cora-ML and another real-world graph WikiCS (Mernyei & Cangea, 2022) and extract the subgraphs defined by the two largest classes. We call

---

[1]all-MiniLM-L6-v2 from `https://huggingface.co/sentence-transformers/all-MiniLM-L6-v2`

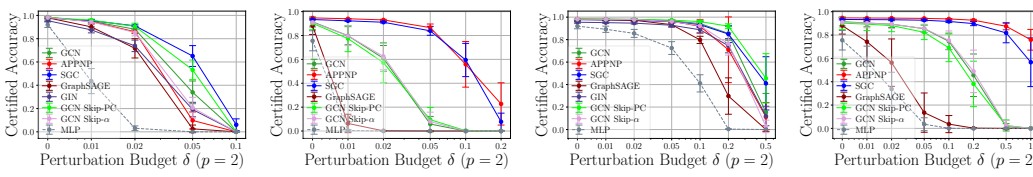

(a) Cora-MLb: $p_{adv} = 1$    (b) WikiCSb: $p_{adv} = 1$    (c) Cora-MLb: $p_{adv} = 0.1$   (d) WikiCSb: $p_{adv} = 0.1$

Figure 2: Poison Labeled ($PL$) setting for Cora-MLb and WikiCS. (a)-(b): QPCert effectively provides non-trivial guarantees. (a)-(d): All GNNs show higher certified accuracy than an MLP.

the resulting datasets *Cora-MLb* and *WikiCSb*, respectively. Lastly, we use graphs generated from Contextual Stochastic Block Models (CSBM) (Deshpande et al., 2018) for controlled experiments on graph parameters. We give dataset statistics and information on the random graph generation scheme in H.1. For Cora-MLb and WikiCSb, we choose 10 nodes per class for training, leaving 1215 and 4640 unlabeled nodes, respectively. For Cora-ML, we choose 20 training nodes per class resulting in 2925 unlabeled nodes. From the CSBM, we sample graphs with 200 nodes and choose 40 per class for training, leaving 120 unlabeled nodes. All results are averaged over 5 seeds (Cora-ML: 3 seeds) and reported with standard deviation. We do not need a separate validation set, as we perform 4-fold cross-validation (CV) for hyperparameter tuning.

**GNNs and attack.** We evaluate GCN, SGC, (A)PPNP, GIN, GraphSAGE, MLP, and the skip connection variants GCN Skip-$\alpha$ and GCN Skip-PC (see App. A). All results concern the infinite-width limit and thus, are obtained through training an SVM with the corresponding GNN's NTK and, if applicable, applying QPCert using Gurobi to solve the MILP from Theorem 1. We fix the hidden layers to $L = 1$, and the results for $L = \{2, 4\}$ are provided in App. I.2. For CSBMs we fix $C = 0.01$ for comparability between experiments and models in the main section. We find that changing $C$ has little effect on the accuracy but can strongly affect the robustness of different architectures. Other parameters on CSBM and all parameters on real-world datasets are set using 4-CV (see App. H.2 for details). The SVM's quadratic dual problem is solved using QPLayer (Bambade et al., 2023), a differentiable quadratic programming solver. Thus, for evaluating tightness regarding graph poisoning, we use APGD (Croce & Hein, 2020) with their reported hyperparameters as attack, but differentiate through the learning process using two different strategies: ($i$) QPLayer, and ($ii$) the surrogate model proposed in MetaAttack (Zügner & Günnemann, 2019a). To evaluate backdoor tightness, we use the clean-label backdoor attack from Xing et al. (2024) as well as the above APGD attack, but at test time additionally attacking the target node.

**Adversarial evaluation settings.** We categorize four settings of interest. **(1)** *Poison Labeled (PL)*: The adversary $\mathcal{A}$ can potentially poison the labeled data $\mathcal{V}_L$. **(2)** *Poison Unlabeled (PU):* Especially interesting in a semi-supervised setting is the scenario when $\mathcal{A}$ can poison the unlabeled data $\mathcal{V}_U$, while the labeled data, usually representing a small curated set of high quality, is known to be clean (Shejwalkar et al., 2023). **(3)** *Backdoor Labeled (BL):* Like (1) but the test node is also controlled by $\mathcal{A}$. **(4)** *Backdoor Unlabeled (BU):* Like (2) but again, the test node is controlled by $\mathcal{A}$. Settings (1) and (2) are evaluated transductively, i.e. on the unlabeled nodes $\mathcal{V}_U$ already known at training time. Note that this means for (2) that some test nodes may be corrupted. For the backdoor attack settings (3) and (4) the test node is removed from the graph during training and added inductively at test time. The size of the untrusted potential adversarial node set $\mathcal{U}$ is set in percentage $p_{adv} \in \{0.01, 0.02, 0.05, 0.1, 0.2, 0.5, 1\}$ of the scenario-dependent attackable node set and resampled for each seed. We consider node feature perturbations $\mathcal{B}_p(\mathbf{x})$ with $p = \{1, 2, \infty\}$ and provide all results concerning $p = 1$ in App. I.5 and J.4. In the case of CSBM, $\delta$ is set in percentage of $2\boldsymbol{\mu}$ of the underlying distribution, and for real data to absolute values. Our main evaluation metric is *certified accuracy*, referring to the percentage of correctly classified nodes without attack that are provably robust against data poisoning / backdoor attacks of the assumed adversary $\mathcal{A}$. We note that we are the first work to study certificates for clean-label attacks on node features in graphs. In particular, all current black-box certificates do not apply to graph learning or $\ell_p$ perturbation models (see App. M). Thus, there is no baseline prior work. However, we still compare the certified accuracies presented below with two common poisoning defenses in App. I.7.

**Non-trivial certificates and On the importance of graph information.** We evaluate the effectiveness of our certificates in providing non-trivial robustness guarantees. Consider the $PL$ setting where $\mathcal{A}$ can poison *all* labeled nodes ($p_{adv} = 1$) for which a trivial certificate would return 0% certified

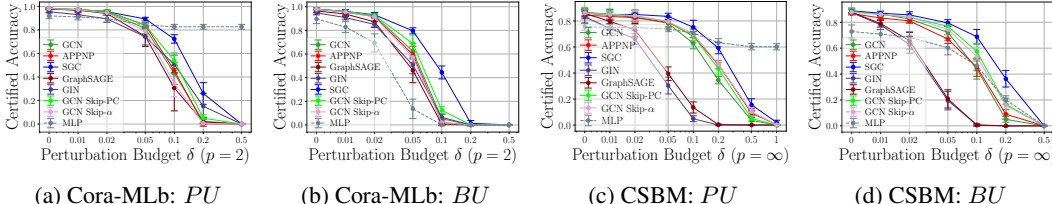

(a) Cora-MLb: $PU$    (b) Cora-MLb: $BU$    (c) CSBM: $PU$    (d) CSBM: $BU$

Figure 3: Certifiable robustness in the Poisoning Unlabeled ($PU$) and Backdoor Unlabeled ($BU$) setting with $p_{adv} = 0.1$ for Cora-MLb and $p_{adv} = 0.2$ for CSBM. We refer to App. L for WikiCSb.

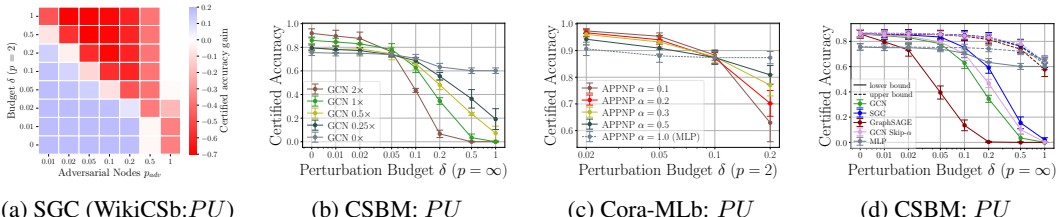

(a) SGC (WikiCSb:$PU$)    (b) CSBM: $PU$    (c) Cora-MLb: $PU$    (d) CSBM: $PU$

Figure 4: Poison Unlabeled for WikiCSb ($p_{adv} = 0.02$), Cora-MLb ($p_{adv} = 0.1$), CSBM ($p_{adv} = 0.2$). (a) Certified accuracy gain: difference of certified acc. to an MLP. (b) Graph connectivity analysis where $c\times$ is $cp$ and $cq$ in CSBM model. (c) APPNP analysis based on $\alpha$. (d) Tightness of QPCert.

accuracy. Figs. 2a and 2b prove that QPCert returns non-trivial guarantees. Further, they highlight an interesting insight: All GNNs have *significantly better* worst-case robustness behavior than the certified accuracy of an MLP. Thus, leveraging the graph connectivity, significantly improves their certified accuracy, even when faced with perturbations on all labeled nodes. In Figs. 2c and 2d we show that this observation stays consistent for other $p_{adv}$. Similar results for Cora-ML (App. K) and CSBM (App. I.1) establish that this behavior is *not* dataset-specific.

In Fig. 3, we evaluate the poison unlabeled ($PU$) and backdoor unlabeled ($BU$) settings for different datasets. When poisoning only unlabeled data ($PU$), the MLP's training process is not affected by the adversary, as the MLP does not access the unlabeled nodes during training. Thus, this provides a good baseline for our certificate to study GNNs. Again, QPCert provides non-trivial certified robustness beyond the MLP baseline. Close to all GNNs show certified accuracy exceeding the one of an MLP for small to intermediate perturbation budgets ($\delta \leq 0.1$) for Cora-MLb (Fig. 3a) and CSBM (Fig. 3a), with a similar picture for WikiCSb (App. L) and Cora-ML (App. K). Note that the drop in certified accuracy for an MLP stems from the transductive learning setting, in which the MLP is confronted with the potentially perturbed unlabeled training nodes at test time. For WikiCSb, Fig. 4a elucidates in detail the certified accuracy gain of an SGC to an MLP and for other GNNs, see App. L (and App. J.1 for Cora-MLb, App. I.1 for CSBM). Concerning backdoor attacks on unlabeled nodes Figs. 3b and 3d show that most GNNs show significantly better certified robustness than an MLP, even so MLP training is not affected by $\mathcal{A}$. We observe similar results for a $BL$ setting for Cora-MLb (App. J.1), WikiCS (App. L), and CSBM (App. I.1). These results show that leveraging graph information can *significantly improve* certified accuracy across all attack settings. Further, across all evaluation settings and datasets, we find GIN and GraphSAGE to provide the lowest certified accuracies of all GNNs; their most important design difference is choosing a sum-aggregation scheme. We note that a comparison across architecture can be affected by the certificate's tightness and we hypothesize that the high worst-case robustness of SGC compared to other models may be due to the certificate being tighter (Theorem 2). However, this still allows us to derive architectural insights for a specific GNN.

**On graph connectivity and architectural insights.** We exemplify study directions enabled through our certification framework. By leveraging CSBMs, we study the effect of graph connectivity in the poisoning unlabeled setting in Fig. 4b for GCN. Interestingly, we observe an inflection point at perturbation strength $\delta = 0.05$, where higher connectivity leads to higher certified accuracy against small perturbations, whereas higher connectivity significantly worsens certified accuracy for strong perturbations. These trends are consistent across various architectures and attack settings (App. I.2).

Secondly, we study the effect of different $\alpha$ choices in APPNP on its certified accuracy in poison unlabeled setting in Fig. 4c. Interestingly, it also shows an inflection point in the perturbation strength

($\delta = 0.1$), where higher $\alpha$ increases the provable robustness for larger $\delta$, whereas worsens the provable robustness for smaller $\delta$ in Cora-MLb. Notably, this phenomenon is unique to the $PU$ setting (see App. J.2) and is similarly observed in CSBM as shown in App. I.2. Although this setup seems to be similar to the connectivity analysis, it is different as the $\alpha$ in APPNP realizes weighted adjacency rather than changing the connectivity of the graph, that is, increasing or decreasing the number of edges in the graph. We compare different normalization choices for $\mathbf{S}$ in GCN and SGC in App. J.3. Through these analyses, it is significant to note that our certification framework enables informed architectural choices from the perspective of robustness.

## 5 DISCUSSION AND RELATED WORK

**How tight is QPCert?** We compute an upper bound on provable robustness using APGD by differentiating through the learning process. The results in Fig. 4d show that the provable robustness bounds are tight for small pertubation budgets $\delta$ but less tight for larger $\delta$, demonstrating one limitation (other settings and attacks in App. I.3). While theoretically, the NTK bounds are tight (Theorem 2), the approach of deriving element-wise bounds on $\mathbf{Q}$ to model $\mathcal{A}$ leading to a relaxation of $\mathrm{P}_3(\mathbf{Q})$ can explain the gap between provable robustness and empirical attack. Thus, we are excited about opportunities for future work to improve our approach for modeling $\mathcal{A}$ in the MILP $P(\mathbf{Q})$.

**Is QPCert deterministic or probabilistic?** Our certification framework is inherently deterministic, offering deterministic guarantees for kernelized SVMs using the NTK as the kernel. When the width of a NN approaches infinity, QPCert provides a deterministic robustness guarantee for the NN due to the exact equivalence between an SVM with the NN's NTK as kernel and the infinitely wide NN. For sufficiently wide but finite-width NNs this equivalence holds with high probability (Chen et al., 2021), making our certificate probabilistic in this context. However, note that this high-probability guarantee is qualitatively different from other methods such as randomized smoothing (Cohen et al., 2019), in which the certification approach itself is probabilistic and heavily relies on the number of samplings and thus, inherently introduces randomness.

**Generality of QPCert.** While we focus on (G)NNs for graph data, our framework enables white-box poisoning certification of NNs on any data domain. QPCert can be extended to other architecture given the criteria outlined in App. N.4 and other tasks such as graph classification (see App. N.5). Further, it allows for certifying general kernelized SVMs for arbitrary kernel choices if respective kernel bounds as in Sec. 3.1 are derived. To the best of our knowledge, this makes our work the first white-box poisoning certificate for kernelized SVMs. Moreover, the reformulation of the bilevel problem to MILP is directly applicable to any quadratic program that satisfies strong Slater's constraint and certain bounds on the involved variables, hence the name QPCert. Thus extensions to certify quadratic programming layers in NN (Amos & Kolter, 2017) or other quadratic learners are thinkable. Therefore, we believe that our work opens up numerous new avenues of research into the provable robustness against data poisoning.

**Perturbation model.** We study semi-verified learning (Charikar et al., 2017). This is particularly interesting for semi-supervised settings, where often a small fraction of nodes are manually verified and labeled (Shejwalkar et al., 2023), or when learning from the crowd Meister & Valiant (2018); Zeng & Shen (2023). However, this may produce overly pessimistic bounds when large fractions of the training data are unverified, but the adversary can only control a small part of it. We study clean-label attacks bounded by $\ell_p$-threat models instead of arbitrary perturbations to nodes controlled by $\mathcal{A}$. We refer to App. N.1 for a discussion with which commonly studied empirical attacks this threat model aligns. Goldblum et al. (2023) distinctively names studying bounded clean-label attacks as an *open problem*, as most works assume unrealistically large input perturbations. Exemplary, in Fig. 2a, QPCert allows us to certify robustness against $\ell_p$-bounded perturbations applied to all labeled data. Most works on poisoning certification work with so-called 'general attacks' allowing arbitrary modifications of data controlled by the adversary. In the setting studied in Fig. 2a, this would always lead to 0 certified accuracy and being unable to provide non-trivial guarantees.

**Related work.** There is little literature on white-box poisoning certificates (see Table 1), and existing techniques (Drews et al., 2020; Meyer et al., 2021; Jia et al., 2022; Bian et al., 2024) cannot be extended to NNs. We summarize the most important related work and refer to App. M for more details. The bilevel problem Eq. (3) is investigated by several works in the context of developing a poisoning attack or empirical defense, including for SVMs (Biggio et al., 2012; Xiao et al., 2015; Koh

& Liang, 2017; Jagielski et al., 2018). Notably Mei & Zhu (2015) reformulate the bilevel problem $\mathrm{P}_2(\boldsymbol{Q})$ for SVMs to a bilinear single-level problem similar to $\mathrm{P}_3(\boldsymbol{Q})$ but only solve it heuristically for attack generation and do not realize the possibility of a MILP reformulation and certification. There are no poisoning certificates for clean-label attacks against GNNs. (Lai et al., 2024) is the only work on poisoning certification of GNNs, but differ incomparably in their threat model and are black-box as well as not applicable to backdoors. Lingam et al. (2024) develops a label poisoning attack for GNNs using the bilevel problem with a regression objective and including NTKs as surrogate models. We note that (Steinhardt et al., 2017) develops statistical bounds on the loss that are not applicable to certify classification.

**Conclusion.** We derive the first white-box poisoning certificate framework for NNs through their NTKs and demonstrate its effective applicability to semi-supervised node classification tasks common in graph learning. In particular, we show that our certificate generates non-trivial robustness guarantees and insights into the worst-case poisoning robustness to feature perturbations of a wide range of GNNs. The study on node feature perturbations is of practical concern in many application areas of GNNs such as spam detection (Li et al., 2019) or fake news detection (Hu et al., 2024) (see App. N.3 for a more detailed discussion), and certification against them poses unique graph-related challenges due to the interconnectedness of nodes. While we address the robustness to node feature perturbations, certifying against structural perturbations to the graph itself remains an open, complex, but important problem and we refer to App. N.2 for a technical discussion on the arising challenges. Thus, this offers a valuable direction for future research. Furthermore, as is the case with all deterministic certificates (Li et al., 2023), scaling to large datasets remains challenging. Consequently, research on scaling deterministic certificates is an impactful avenue for future work.

## 6 ETHICS STATEMENT

Our method represents a robustness certificate for white-box models. This allows a more informed decision when it comes to the safety aspects of currently used models. However, insights into worst-case robustness can be used for good but potentially also by malicious actors. We strongly believe that research about the limitations of existing models is crucial in making models safer and thus, outweighs potential risks. We are not aware of any direct risks coming from our work.

## 7 REPRODUCIBILITY STATEMENT

We detail all the experimental setups with the network architectures and hyperparameters in Sec. 4 and app. H.2. The used datasets are open source as mentioned in App. H.1, and the hardware details are discussed in App. H.3. We provide the complete code base with datasets and configuration files to reproduce the experiments in `https://figshare.com/s/e155ced9910eb7b3a531`. The randomness in the experiments is controlled by setting fixed seeds which are given in the experiment configuration files. The code will be made public upon acceptance.

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

# A  ARCHITECTURE DEFINITIONS

We consider GNNs as functions $f_\theta$ with (learnable) parameters $\theta \in \mathbb{R}^q$ and $L$ number of layers taking the graph $\mathcal{G} = (\boldsymbol{S}, \boldsymbol{X})$ as input and outputs a prediction for each node. We consider linear output layers with weights $\boldsymbol{W}^{L+1}$ and denote by $f_\theta(\mathcal{G})_i \in \mathbb{R}^K$ the (unnormalized) logit output associated to node $i$. In the following, we formally define the (G)NNs such as MLP, GCN (Kipf & Welling, 2017), SGC (Wu et al., 2019) and (A)PPNP (Gasteiger et al., 2019) considered in our study.

**Def. 1** (MLP). *The $L$-layer Multi-Layer Perceptron is defined as $f_\theta(\mathcal{G})_i = f_\theta^{MLP}(\boldsymbol{x}_i) = \boldsymbol{W}^{L+1}\phi_\theta^{(L)}(\boldsymbol{x}_i)$. With $\phi_\theta^l(\boldsymbol{x}_i) = \sigma(\boldsymbol{W}^{(l)}\phi_\theta^{(l-1)}(\boldsymbol{x}_i) + \boldsymbol{b}^{(l)})$ and $\phi_\theta^{(0)}(\boldsymbol{x}_i) = \boldsymbol{x}_i$. $\boldsymbol{W}^{(l)} \in \mathbb{R}^{d_{l+1} \times d_l}$ and $\boldsymbol{b}^{(l)} \in \mathbb{R}^{d_l}$ are the weights/biases of the $l$-th layer with $d_0 = d$ and $d_{L+1} = K$. $\sigma(\cdot)$ is an element-wise activation function. If not mentioned otherwise, we choose $\sigma(z) = ReLU(z) = \max\{0, z\}$.*

**Def. 2** (GCN & SGC). *A Graph Convolution Network $f_\theta^{GCN}(\mathcal{G})$ (Kipf & Welling, 2017) of depth $L$ is defined as $f_\theta(\mathcal{G}) = \phi_\theta^{(L+1)}(\mathcal{G})$ with $\phi_\theta^{(l)}(\mathcal{G}) = \mathbf{S}\sigma(\phi_\theta^{(l-1)}(\mathcal{G}))\mathbf{W}^{(l)}$ and $\phi_\theta^{(1)}(\mathcal{G}) = \mathbf{SXW}^{(1)}$. $\mathbf{W}^{(l)} \in \mathbb{R}^{d_{l-1} \times d_l}$ are the $l$-th layer weights, $d_0 = d$, $d_{L+1} = K$, and $\sigma(z) = ReLU(z)$ applied element-wise. A Simplified Graph Convolution Network $f_\theta^{SGC}(\mathcal{G})$ (Wu et al., 2019) is a GCN with linear $\sigma(z) = z$.*

**Def. 3** (GraphSAGE). *The $L$-layer GraphSAGE $f_\theta^{GSAGE}(\mathcal{G})$ (Hamilton et al., 2017) is defined as $f_\theta(\mathcal{G}) = \phi_\theta^{(L+1)}(\mathcal{G})$ with $\phi_\theta^{(l)}(\mathcal{G}) = \sigma(\phi_\theta^{(l-1)}(\mathcal{G}))\mathbf{W}_1^{(l)} + \mathbf{S}\sigma(\phi_\theta^{(l-1)}(\mathcal{G}))\mathbf{W}_2^{(l)}$ and $\phi_\theta^{(1)}(\mathcal{G}) = \mathbf{XW}_1^{(1)} + \mathbf{SXW}_2^{(1)}$. $\mathbf{W}_1^{(l)}, \mathbf{W}_2^{(l)} \in \mathbb{R}^{d_{l-1} \times d_l}$ are the $l$-th layer weights, $d_0 = d$, $d_{L+1} = K$, and $\sigma(z) = ReLU(z)$ applied element-wise. $\mathbf{S}$ is fixed to row normalized adjacency (mean aggregator), $\mathbf{D}^{-1}\mathbf{A}$.*

**Def. 4** (GIN). *A one-layer Graph Isomorphism Network $f_\theta^{GIN}(\mathcal{G})$ (Xu et al., 2018) is defined as $f_\theta(\mathcal{G}) = f_\theta^{MLP}(\widetilde{\mathbf{G}})$ with $\widetilde{\mathbf{G}} = ((1 + \epsilon)\mathbf{I} + \mathbf{A})\mathbf{X}$ where $\epsilon$ is a fixed constant and an one-layer ReLU as MLP.*

**Def. 5** ((A)PPNP). *The Personalized Propagation of Neural Predictions Network $f_\theta^{PPNP}(\mathcal{G})$ Gasteiger et al. (2019) is defined as $f_\theta(\mathcal{G}) = \mathbf{PH}$ where $\mathbf{H}_{i,:} = f_\theta^{MLP}(\mathbf{x}_i)$ and $\mathbf{P} = \alpha(\mathbf{I}_n - (1 - \alpha)\mathbf{S})^{-1}$. The Approximate PPNP is defined with $\mathbf{P} = (1 - \alpha)^K\mathbf{S}^K + \alpha\sum_{i=0}^{K-1}(1 - \alpha)^i\mathbf{S}^i$ where $\alpha \in [0, 1]$ and $K \in \mathbb{N}$ is a fixed constant.*

For APPNP and GIN, we consider an MLP with one-layer ReLU activations as given in the default implementation of APPNP.

Along with the GNNs presented in Definitions 1 to 5, we consider two variants of popular skip connections in GNNs as given a name in Sabanayagam et al. (2023): Skip-PC (pre-convolution), where the skip is added to the features before applying convolution (Kipf & Welling, 2017); and Skip-$\alpha$, which adds the features to each layer without convolving with $\mathbf{S}$ (Chen et al., 2020). To facilitate skip connections, we need to enforce constant layer size, that is, $d_i = d_{i-1}$. Therefore, the input layer is transformed using a random matrix $\mathbf{W}$ to $\mathbf{H}_0 := \mathbf{XW}$ of size $n \times h$ where $\mathbf{W}_{ij} \sim \mathcal{N}(0, 1)$ and $h$ is the hidden layer size. Let $\mathbf{H}_i$ be the output of layer $i$ using which we formally define the skip connections as follows.

**Def. 6** (Skip-PC). *In a Skip-PC (pre-convolution) network, the transformed input $\mathbf{H}_0$ is added to the hidden layers before applying the graph convolution $\mathbf{S}$, that is, $\forall i \in [L], \phi_\theta^{(l)}(\mathcal{G}) = \mathbf{S}\left(\sigma(\phi_\theta^{(l-1)}(\mathcal{G})) + \sigma_s(\mathbf{H}_0)\right)\mathbf{W}^{(l)}$, where $\sigma_s(z)$ can be linear or ReLU.*

Skip-PC definition deviates from Kipf & Welling (2017) because we skip to the input layer instead of the previous one. We define Skip-$\alpha$ as defined in Sabanayagam et al. (2023) similar to Chen et al. (2020).

**Def. 7** (Skip-$\alpha$). *Given an interpolation coefficient $\alpha \in (0, 1)$, a Skip-$\alpha$ network is defined such that the transformed input $\mathbf{H}_0$ and the hidden layer are interpolated linearly, that is, $\phi_\theta^{(l)}(\mathcal{G}) = \left((1 - \alpha)\mathbf{S}\phi_\theta^{(l-1)}(\mathcal{G}) + \alpha\sigma_s(\mathbf{H}_0)\right)\mathbf{W}_i \ \forall i \in [L]$, where $\sigma_s(z)$ can be linear or ReLU.*

## B  DERIVATION OF NTKS FOR (A)PPNP, GIN AND GRAPHSAGE

In this section, we derive the NTKs for (A)PPNP, GIN and GraphSAGE, and state the NTKs for GCN and SGC from Sabanayagam et al. (2023).

### B.1  NTK FOR (A)PPNP

We derive the closed-form NTK expression for (A)PPNP $f_\theta(\mathcal{G})$ (Gasteiger et al., 2019) in this section. The learnable parameters $\theta$ are only part of $\mathbf{H}$. In practice, $\mathbf{H} = \mathrm{ReLU}(\mathbf{X}\mathbf{W}_1 + \mathbf{B}_1)\mathbf{W}_2 + \mathbf{B}_2$ where node features $\mathbf{X}, \theta = \{\mathbf{W}_1 \in \mathbb{R}^{d \times h}, \mathbf{W}_2 \in \mathbb{R}^{h \times K}, \mathbf{B}_1 \in \mathbb{R}^{n \times h}, \mathbf{B}_2 \in \mathbb{R}^{n \times K}\}$. Note that in the actual implementation of the MLP, $\mathbf{B}_1$ is a vector and we consider it to be a matrix by having the same columns so that we can do matrix operations easily. Same for $\mathbf{B}_2$ as well. We give the full architecture with NTK parameterization in the following,

$$f_\theta(\mathcal{G}) = \mathbf{P}(\frac{c_\sigma}{\sqrt{h}}\sigma(\mathbf{X}\mathbf{W}_1 + \mathbf{B}_1)\mathbf{W}_2 + \mathbf{B}_2) \tag{10}$$

where $h \to \infty$ and all parameters in $\theta$ are initialized as standard Gaussian $\mathcal{N}(0, 1)$. $c_\sigma$ is a constant to preserve the input norm (Sabanayagam et al., 2023). We derive for $K = 1$ as all the outputs are equivalent in expectation. The NTK between nodes $i$ and $j$ is $\mathbb{E}_{\theta \sim \mathcal{N}(0,1)}[\langle \nabla_\theta f_\theta(\mathcal{G})_i, \nabla_\theta f_\theta(\mathcal{G})_j \rangle]$.

Hence, we first write down the gradients for node $i$ following (Arora et al., 2019; Sabanayagam et al., 2023):

$$\frac{\partial f_\theta(\mathcal{G})_i}{\partial \mathbf{W}_2} = \frac{c_\sigma}{\sqrt{h}}(\mathbf{P}_i \sigma(\mathcal{G}_1))^T \qquad\qquad ; \mathcal{G}_1 = \mathbf{X}\mathbf{W}_1 + \mathbf{B}_1$$

$$\frac{\partial f_\theta(\mathcal{G})_i}{\partial \mathbf{B}_2} = (\mathbf{P}_i)^T \mathbf{1}_n$$

$$\frac{\partial f_\theta(\mathcal{G})_i}{\partial \mathbf{W}_1} = \frac{c_\sigma}{\sqrt{h}}\mathbf{X}^T(\mathbf{P}_i^T \mathbf{1}_n \mathbf{W}_2^T \odot \dot\sigma(\mathcal{G}_1))$$

$$\frac{\partial f_\theta(\mathcal{G})_i}{\partial \mathbf{B}_1} = \frac{c_\sigma}{\sqrt{h}}\mathbf{P}_i^T \mathbf{1}_n \mathbf{W}_2^T \odot \dot\sigma(\mathcal{G}_1)$$

We note that $\mathbf{B}_2$ has only one learnable parameter for $K = 1$, but is represented as a vector of size $n$ with all entries the same. Hence, the derivative is simply adding all entries of $\mathbf{P}_i$. First, we compute the covariance between nodes $i$ and $j$ in $\mathcal{G}_1$.

$$\mathbb{E}\left[(G_1)_{ik}(G_1)_{jk'}\right] = \mathbb{E}\left[(\mathbf{X}\mathbf{W}_1 + \mathbf{B}_1)_{ik}(\mathbf{X}\mathbf{W}_1 + \mathbf{B}_1)_{jk'}\right]$$

Since the expectation is over $\mathbf{W}_1$ and $\mathbf{B}_1$ and all entries are $\sim \mathcal{N}(0, 1)$, and i.i.d, the cross terms will be 0 in expectation. Also, for $k \neq k'$, it is 0. Therefore, it gets simplified to

$$\mathbb{E}\left[(G_1)_{ik}(G_1)_{jk}\right] = \mathbb{E}\left[\mathbf{X}_i \mathbf{W}_1 \mathbf{W}_1^T \mathbf{X}_j^T + (\mathbf{B}_1 \mathbf{B}_1^T)_{ij}\right]$$

$$= (\mathbf{X}\mathbf{X}^T)_{ij} + 1 = (\Sigma_1)_{ij} \tag{11}$$

Thus, $\mathbf{\Sigma}_1 = \mathbf{X}\mathbf{X}^T + \mathbf{1}_{n \times n}$ and let $(E_1)_{ij} = \mathbb{E}\left[\sigma(\mathcal{G}_1)_i \sigma(\mathcal{G}_1)_j^T\right]$ and $\left(\dot E_1\right)_{ij} = \mathbb{E}\left[\dot\sigma(\mathcal{G}_1)_i \dot\sigma(\mathcal{G}_1)_j^T\right]$ computed using the definitions in Theorem 3 for ReLU non-linearity. Now, we can compute the NTK for each parameter matrix and then sum it up to get the final kernel.

$$\left\langle \frac{\partial f_\theta(\mathcal{G})_i}{\partial \mathbf{W}_2}, \frac{\partial f_\theta(\mathcal{G})_j}{\partial \mathbf{W}_2} \right\rangle = \frac{c_\sigma^2}{h}\mathbf{P}_i \sigma(\mathcal{G}_1)\sigma(\mathcal{G}_1)^T \mathbf{P}_j^T$$

$$\stackrel{h \to \infty}{=} c_\sigma^2 \mathbf{P}_i \mathbb{E}\left[\sigma(\mathcal{G}_1)\sigma(\mathcal{G}_1)^T\right]\mathbf{P}_j^T = c_\sigma^2 \mathbf{P}_i \mathbf{E}_1 \mathbf{P}_j^T \tag{12}$$

$$\left\langle \frac{\partial f_\theta(\mathcal{G})_i}{\partial \mathbf{B}_2}, \frac{\partial f_\theta(\mathcal{G})_j}{\partial \mathbf{B}_2} \right\rangle = \mathbf{P}_i \mathbf{1}_{n \times n}\mathbf{P}_j^T \tag{13}$$

$$\left\langle \frac{\partial f_\theta(\mathcal{G})_i}{\partial \mathbf{B}_1}, \frac{\partial f_\theta(\mathcal{G})_j}{\partial \mathbf{B}_1} \right\rangle \overset{h \to \infty}{=} c_\sigma^2 \mathbf{P}_i(\mathbb{E}\left[\dot{\sigma}(\mathcal{G}_1)\dot{\sigma}(\mathcal{G}_1)\right])\mathbf{P}_j^T = c_\sigma^2 \mathbf{P}_i \dot{\mathbf{E}}_1 \mathbf{P}_j^T \tag{14}$$

$$\left\langle \frac{\partial f_\theta(\mathcal{G})_i}{\partial \mathbf{W}_1}, \frac{\partial f_\theta(\mathcal{G})_j}{\partial \mathbf{W}_1} \right\rangle \overset{f,h}{=} \frac{c_\sigma^2}{h} \sum_{p,q}^{f,h} (\mathbf{X}^T(\mathbf{P}_i^T \mathbf{1}_n \mathbf{W}_2^T \odot \dot{\sigma}(\mathcal{G}_1)))_{pq} (\mathbf{X}^T(\mathbf{P}_j^T \mathbf{1}_n \mathbf{W}_2^T \odot \dot{\sigma}(\mathcal{G}_1)))_{pq}$$

$$= \frac{c_\sigma^2}{h} \sum_{p=1}^{d} \sum_{q=1}^{h} \Big[ \sum_{a=1}^{n} (\mathbf{X}^T)_{pa}(\mathbf{P}_i^T \mathbf{W}_2^T)_{aq}\dot{\sigma}(\mathcal{G}_1)_{aq}$$

$$\sum_{b=1}^{n} (\mathbf{X}^T)_{pb}(\mathbf{P}_j^T \mathbf{W}_2^T)_{bq}\dot{\sigma}(\mathcal{G}_1)_{bq}\Big]$$

$$\overset{h \to \infty}{=} c_\sigma^2 \sum_{a=1,b=1}^{n,n} (\mathbf{X}\mathbf{X}^T)_{ab}\mathbf{P}_{ia}(\mathbf{P}^T)_{bj}\mathbb{E}\left[\dot{\sigma}(\mathcal{G}_1)\dot{\sigma}(\mathcal{G}_1)\right]_{ab}$$

$$= c_\sigma^2 \mathbf{P}_i(\mathbf{X}\mathbf{X}^T \odot \mathbb{E}\left[\dot{\sigma}(\mathcal{G}_1)\dot{\sigma}(\mathcal{G}_1)\right])\mathbf{P}_j^T = c_\sigma^2 \mathbf{P}_i(\mathbf{X}\mathbf{X}^T \odot \dot{\mathbf{E}})_1\mathbf{P}_j^T \tag{15}$$

Finally, the NTK matrix for the considered (A)PPNP is sum of Eqs. (12) to (15) as shown below.

$$\mathbf{Q} = c_\sigma^2 \left( \mathbf{PE}_1\mathbf{P}^T + \mathbf{P}\mathbf{1}_{n \times n}\mathbf{P}^T + \mathbf{P}\dot{\mathbf{E}}_1\mathbf{P} + \mathbf{P}\left(\mathbf{X}\mathbf{X}^T \odot \dot{\mathbf{E}}_1\right)\mathbf{P}^T \right)$$

$$= c_\sigma^2 \left( \mathbf{P}\left(\mathbf{E}_1 + \mathbf{1}_{n \times n}\right)\mathbf{P}^T + \mathbf{P}\left(\left(\mathbf{X}\mathbf{X}^T + \mathbf{1}_{n \times n}\right) \odot \dot{\mathbf{E}}_1\right)\mathbf{P}^T \right)$$

$$= c_\sigma^2 \left( \mathbf{P}\left(\mathbf{E}_1 + \mathbf{1}_{n \times n}\right)\mathbf{P}^T + \mathbf{P}\left(\mathbf{\Sigma}_1 \odot \dot{\mathbf{E}}_1\right)\mathbf{P}^T \right) \tag{16}$$

Note that $c_\sigma$ is a constant, and it only scales the NTK, so we set it to 1 in our experiments. Since we use a linear output layer without bias term at the end, that is, $\mathbf{B}_2 = 0$, the NTK we use for our experiments is reduced to

$$\mathbf{Q} = \left( \mathbf{PE}_1\mathbf{P}^T + \mathbf{P}\left(\mathbf{\Sigma}_1 \odot \dot{\mathbf{E}}_1\right)\mathbf{P}^T \right).$$

$\square$

## B.2 NTK FOR GIN

The GIN architecture Definition 4 is similar to APPNP: $\mathbf{P}$ and $\mathbf{X}$ in APPNP are Identity and $\widetilde{\mathbf{G}}$ in GIN, respectively. Hence the NTK is exactly the same as APPNP with these matrices. Thus, the NTK for GIN is

$$\mathbf{Q} = \mathbf{E}_1 + \left(\mathbf{\Sigma}_1 \odot \dot{\mathbf{E}}_1\right)$$

with $\mathbf{\Sigma}_1 = \widetilde{\mathbf{G}}\widetilde{\mathbf{G}}^T + \mathbf{1}_{n \times n}$, $\mathbf{E}_1 = \underset{\mathbf{F} \sim \mathcal{N}(\mathbf{0}, \mathbf{\Sigma}_1)}{\mathbb{E}} \left[\sigma(\mathbf{F})\sigma(\mathbf{F})^T\right]$ and $\dot{\mathbf{E}}_1 = \underset{\mathbf{F} \sim \mathcal{N}(\mathbf{0}, \mathbf{\Sigma}_1)}{\mathbb{E}} \left[\dot{\sigma}(\mathbf{F})\dot{\sigma}(\mathbf{F})^T\right]$. $\square$

## B.3 NTKS FOR GCN AND SGC

We restate the NTK derived in Sabanayagam et al. (2023) for self containment. The GCN of depth $L$ with width $d_l \to \infty \ \forall l \in \{1, \ldots, L\}$, the network converges to the following kernel when trained with gradient flow.

**Theorem 3** (NTK for Vanilla GCN). *For the GCN defined in Definition 2, the NTK $\mathbf{Q}$ at depth $L$ and $K = 1$ is*

$$\mathbf{Q}^{(L)} = \sum_{k=1}^{L+1} \mathbf{S}\underbrace{\left(\ldots\mathbf{S}\left(\mathbf{S}\left(\mathbf{\Sigma}_k \odot \dot{\mathbf{E}}_k\right)\mathbf{S}^T \odot \dot{\mathbf{E}}_{k+1}\right)\mathbf{S}^T \odot \ldots \odot \dot{\mathbf{E}}_L\right)}_{L+1-k \ terms}\mathbf{S}^T. \tag{17}$$

*Here $\mathbf{\Sigma}_k \in \mathbb{R}^{n \times n}$ is the co-variance between nodes of layer $k$, and is given by $\mathbf{\Sigma}_1 = \mathbf{S}\mathbf{X}\mathbf{X}^T\mathbf{S}^T$, $\mathbf{\Sigma}_k = \mathbf{S}\mathbf{E}_{k-1}\mathbf{S}^T$ with $\mathbf{E}_k = c_\sigma \underset{\mathbf{F} \sim \mathcal{N}(\mathbf{0}, \mathbf{\Sigma}_k)}{\mathbb{E}} \left[\sigma(\mathbf{F})\sigma(\mathbf{F})^T\right]$, $\dot{\mathbf{E}}_k = c_\sigma \underset{\mathbf{F} \sim \mathcal{N}(\mathbf{0}, \mathbf{\Sigma}_k)}{\mathbb{E}} \left[\dot{\sigma}(\mathbf{F})\dot{\sigma}(\mathbf{F})^T\right]$ and $\dot{\mathbf{E}}_{L+1} = \mathbf{1}_{n \times n}$.*

$$\left(E_k\right)_{ij} = \sqrt{(\Sigma_k)_{ii}\,(\Sigma_k)_{jj}}\;\kappa_1\left(\frac{(\Sigma_k)_{ij}}{\sqrt{(\Sigma_k)_{ii}\,(\Sigma_k)_{jj}}}\right)$$

$$\left(\dot{E}_k\right)_{ij} = \kappa_0\left(\frac{(\Sigma_k)_{ij}}{\sqrt{(\Sigma_k)_{ii}\,(\Sigma_k)_{jj}}}\right),$$

*where* $\kappa_0(z) = \dfrac{1}{\pi}\left(\pi - arccos\,(z)\right)$ *and* $\kappa_1(z) = \dfrac{1}{\pi}\left(z\left(\pi - arccos\,(z)\right) + \sqrt{1-z^2}\right).$

### B.4 NTK FOR GRAPHSAGE

From the definition of GraphSAGE Definition 3, it is very similar to GCN with row normalization adjacency $\mathbf{S} = \widehat{\mathbf{D}}^{-1}\widehat{\mathbf{A}}$ where $\widehat{\mathbf{A}}$ and $\widehat{\mathbf{D}}$ are adjacency matrix with self-loop and its corresponding degree matrix. The differences in GraphSAGE are the following: there is no self-loop to the adjacency as $\mathbf{S} = \mathbf{D}^{-1}\mathbf{A}$, and the neighboring node features are weighted differently compared to the node itself using $\mathbf{W}_1$ and $\mathbf{W}_2$. Given that the NTK is computed as the expectation over weights at initialization and infinite width, both $\mathbf{W}_1$ and $\mathbf{W}_2$ behave similarly. Hence, these weights can be replaced with a single parameter $\mathbf{W}$ which transforms the network definition of GraphSAGE to $\phi_\theta^1(\mathcal{G}) = (\mathbf{I} + \mathbf{S})\mathbf{X}\mathbf{W}^{(1)}$ and similarly $\phi_\theta^l(\mathcal{G}) = (\mathbf{I} + \mathbf{S})\sigma(\phi_\theta^{(l-1)(\mathcal{G})})\mathbf{W}^{(l)}$ with $\mathbf{S} = \mathbf{D}^{-1}\mathbf{A}$. Thus, the NTK for GraphSAGE is the same as GCN with the difference in the graph normalization $\mathbf{S}$. $\square$

## C EQUIVALENCE OF GNNS TO SVMS

We show the equivalence between GNNs and SVMs by extending the result from Chen et al. (2021), which showed that an infinite-width NN trained by gradient descent on a soft-margin loss has the same training dynamics as that of an SVM with the NN's NTK as the kernel. The fulcrum of their proof that directly depends on the NN is that the NTK stays constant throughout the training (refer to (Chen et al., 2021, Theorem 3.4)). As we consider the same learning setup with only changing the network to GNNs, it is enough to show that the graph NTKs stay constant throughout training for the equivalence to hold in this case.

**Constancy of Graph NTKs.** This constancy of the NTK in the case of infinitely-wide NNs is deeply studied in Liu et al. (2020) and derived the conditions for the constancy as stated in Theorem 4.

**Theorem 4** ((Liu et al., 2020)). *The constancy of the NTK throughout the training of the NN holds if and only if* $(i)$ *the last layer of the NN is linear;* $(ii)$ *the Hessian spectral norm* $\|\mathbf{H}\|$ *of the neural network with respect to the parameters is small, that is,* $\to 0$ *with the width of the network;* $(iii)$ *the parameters of the network* $\mathbf{w}$ *during training and at initialization is bounded, that is, parameters at time* $t$, $\mathbf{w}_t$, *satisfies* $\|\mathbf{w}_t - \mathbf{w}_0\|_2 \le \epsilon$.

Now, we prove the constancy of graph NTKs of the GNNs by showing the three conditions.

$(i)$ **Linear last layer.** The GNNs considered in Definitions 1 and 7 have a linear last layer.

$(ii)$ **Small Hessian spectral norm.** Recollect that we use NTK parameterization for initializing the network parameters, that is, $\mathcal{N}(0, 1/\text{width})$. This is equivalent to initializing the network with standard normal $\mathcal{N}(0, 1)$ and appropriately normalizing the layer outputs (Arora et al., 2019; Sabanayagam et al., 2023). To exemplify, the APPNP network definition with the normalization is given in Eq. (10). Similarly for other GNNs, the normalization results in scaling $\phi_\theta^{(l)}$ as $\frac{c_\sigma}{\sqrt{h}}\phi_\theta^{(l)}$ where $h$ is the width of the layer $l$. As all our GNNs have a simple matrix multiplication of the graph structure without any bottleneck layer, the Hessian spectral norm is $\mathcal{O}(\ln h/\sqrt{h})$ as derived for the multilayer fully connected networks in Liu et al. (2020). Therefore, as $h \to \infty$ the spectral norm $\to 0$.

$(iii)$ **Bounded parameters.** This is dependent only on the optimization of the loss function as derived in Chen et al. (2021, Lemma D.1). We directly use this result as our loss and the optimization are the same as (Chen et al., 2021).

With this, we show that the considered GNNs trained by gradient descent on soft-margin loss is equivalent to SVM with the graph NTK as the kernel. □

# D DERIVATION OF NTK BOUNDS AND THEOREM 2

In this section, we first present the bounds for $\Delta$ in the case of $p = 2$ and $p = 1$ in $\mathcal{B}_p(\mathbf{x})$ (Lemma 2 and Lemma 3), and then derive Lemmas 1, 2 and 3 and Theorem 2 stated in Sec. 3.1.

**Lemma 2** (Bounds for $\Delta$, $p = 2$). *Given $\mathcal{B}_2(\mathbf{x})$ and any $\widetilde{\mathbf{X}} \in \mathcal{A}(\mathbf{X})$, then $\widetilde{\mathbf{X}}\widetilde{\mathbf{X}}^T = \mathbf{X}\mathbf{X}^T + \Delta$ where the worst-case bounds for $\Delta$, $\Delta_{ij}^L \leq \Delta_{ij} \leq \Delta_{ij}^U$ for all $i$ and $j \in [n]$, is*

$$\Delta_{ij}^L = -\delta\|\mathbf{X}_j\|_2 \mathbb{1}[i \in \mathcal{U}] - \delta\|\mathbf{X}_i\|_2 \mathbb{1}[j \in \mathcal{U}] - \delta^2 \mathbb{1}[i \in \mathcal{U} \wedge j \in \mathcal{U} \wedge i \neq j]$$
$$\Delta_{ij}^U = \delta\|\mathbf{X}_j\|_2 \mathbb{1}[i \in \mathcal{U}] + \delta\|\mathbf{X}_i\|_2 \mathbb{1}[j \in \mathcal{U}] + \delta^2 \mathbb{1}[i \in \mathcal{U} \wedge j \in \mathcal{U}] \quad (18)$$

**Lemma 3** (Bounds for $\Delta$, $p = 1$). *Given $\mathcal{B}_2(\mathbf{x})$ and any $\widetilde{\mathbf{X}} \in \mathcal{A}(\mathbf{X})$, then $\widetilde{\mathbf{X}}\widetilde{\mathbf{X}}^T = \mathbf{X}\mathbf{X}^T + \Delta$ where the worst-case bounds for $\Delta$, $\Delta_{ij}^L \leq \Delta_{ij} \leq \Delta_{ij}^U$ for all $i$ and $j \in [n]$, is*

$$\Delta_{ij}^L = -\delta\|\mathbf{X}_j\|_\infty \mathbb{1}[i \in \mathcal{U}] - \delta\|\mathbf{X}_i\|_\infty \mathbb{1}[j \in \mathcal{U}] - \delta^2 \mathbb{1}[i \in \mathcal{U} \wedge j \in \mathcal{U} \wedge i \neq j]$$
$$\Delta_{ij}^U = \delta\|\mathbf{X}_j\|_\infty \mathbb{1}[i \in \mathcal{U}] + \delta\|\mathbf{X}_i\|_\infty \mathbb{1}[j \in \mathcal{U}] + \delta^2 \mathbb{1}[i \in \mathcal{U} \wedge j \in \mathcal{U}] \quad (19)$$

To derive Lemmas 1, 2 and 3, we consider the perturbed feature matrix $\widetilde{\mathbf{X}} \in \mathcal{A}(\mathbf{X})$ and derive the worst-case bounds for $\widetilde{\mathbf{X}}\widetilde{\mathbf{X}}^T$ based on the perturbation model $\mathcal{B}_p(\mathbf{x})$ where $p = \infty$, $p = 2$ and $p = 1$ in our study. Let's say $\mathcal{U}$ is the set of nodes that are potentially controlled by the adversary $\mathcal{A}(\mathbf{X})$ and $\widetilde{\mathbf{X}} = \mathbf{X} + \mathbf{\Gamma} \in \mathbb{R}^{n \times d}$ where $\mathbf{\Gamma}_i$ is the adversarial perturbations added to node $i$ by the adversary, therefore, $\|\mathbf{\Gamma}_i\|_p \leq \delta$ and $\mathbf{\Gamma}_i > 0$ for $i \in \mathcal{U}$ and $\mathbf{\Gamma}_i = 0$ for $i \notin \mathcal{U}$. Then

$$\widetilde{\mathbf{X}}\widetilde{\mathbf{X}}^T = (\mathbf{X} + \mathbf{\Gamma})(\mathbf{X} + \mathbf{\Gamma})^T$$
$$= \mathbf{X}\mathbf{X}^T + \mathbf{\Gamma}\mathbf{X}^T + \mathbf{X}\mathbf{\Gamma}^T + \mathbf{\Gamma}\mathbf{\Gamma}^T = \mathbf{X}\mathbf{X}^T + \Delta. \quad (20)$$

As a result, it suffices to derive the worst-case bounds for $\Delta$, $\Delta^L \leq \Delta \leq \Delta^U$, for different perturbations. To do so, our strategy is to bound the scalar products $\langle \mathbf{\Gamma}_i, \mathbf{X}_j \rangle$ and $\langle \mathbf{\Gamma}_i, \mathbf{\Gamma}_j \rangle$ element-wise, hence derive $\Delta_{ij}^L \leq \Delta_{ij} \leq \Delta_{ij}^U$. In the following, we derive $\Delta_{ij}^L$ and $\Delta_{ij}^U$ for the cases when $p = \infty$, $p = 2$ and $p = 1$ in $\mathcal{B}_p(\mathbf{x})$.

**Case $(i)$: Derivation of Lemma 1 for $p = \infty$.** In this case, the perturbation allows $\|\tilde{\mathbf{X}}_i - \mathbf{X}_i\|_\infty \leq \delta$, then by Hölder's inequality $\langle \mathbf{a}, \mathbf{b} \rangle \leq \|\mathbf{a}\|_p\|\mathbf{b}\|_q$ where $\frac{1}{p} + \frac{1}{q} = 1$ for all $p, q \in [1, \infty]$ we have

$$|\langle \mathbf{\Gamma}_i, \mathbf{X}_j \rangle| \leq \|\mathbf{\Gamma}_i\|_\infty\|\mathbf{X}_j\|_1 \leq \delta\|\mathbf{X}_j\|_1$$
$$|\langle \mathbf{\Gamma}_i, \mathbf{\Gamma}_j \rangle| \leq \|\mathbf{\Gamma}_i\|_2\|\mathbf{\Gamma}_j\|_2 \leq d\|\Delta_i\|_\infty\|\Delta_j\|_\infty \leq d\delta^2. \quad (21)$$

Using Eq. (21), the worst-case lower bound $\Delta_{ij}^L$ is the lower bound of $\mathbf{\Gamma}\mathbf{X}^T + \mathbf{X}\mathbf{\Gamma}^T + \mathbf{\Gamma}\mathbf{\Gamma}^T$:

$$\Delta_{ij}^L = \begin{cases} 0+ & \text{if } i, j \notin \mathcal{U} \\ -\delta\|\mathbf{X}_j\|_1+ & \text{if } i \in \mathcal{U} \\ -\delta\|\mathbf{X}_i\|_1+ & \text{if } j \in \mathcal{U} \\ -\delta^2 d & \text{if } i, j \in \mathcal{U} \text{ and } i \neq j. \end{cases} \quad (22)$$

The last case in Eq. (22) is due to the fact that $\langle \mathbf{\Gamma}_i, \mathbf{\Gamma}_i \rangle \geq 0$, hence $\Delta_{ii}^L = 0$. Finally, the Eq. (22) can be succinctly written using the indicator function as

$$\Delta_{ij}^L = -\delta\|\mathbf{X}_j\|_1 \mathbb{1}[i \in \mathcal{U}] - \delta\|\mathbf{X}_i\|_1 \mathbb{1}[j \in \mathcal{U}] - \delta^2 d \mathbb{1}[i \in \mathcal{U} \wedge j \in \mathcal{U} \wedge i \neq j],$$

deriving the lower bound in Lemma 1. Similarly, applying the Hölder's inequality for the worst-case upper bound, we get

$$\Delta_{ij}^{U} = \begin{cases} 0+ & \text{if } i,j \notin \mathcal{U} \\ \delta \|\mathbf{X}_j\|_1+ & \text{if } i \in \mathcal{U} \\ \delta \|\mathbf{X}_i\|_1+ & \text{if } j \in \mathcal{U} \\ \delta^2 d & \text{if } i,j \in \mathcal{U}. \end{cases} \tag{23}$$

Thus, we derive Lemma 1 by succinctly writing it as

$$\Delta_{ij}^{U} = \delta\|\mathbf{X}_j\|_1 \mathbb{1}[i \in \mathcal{U}] + \delta\|\mathbf{X}_i\|_1 \mathbb{1}[j \in \mathcal{U}] + \delta^2 d \mathbb{1}[i \in \mathcal{U} \wedge j \in \mathcal{U}].$$

$\square$

**Case $(ii)$: Derivation of Lemma 2 for $p = 2$.** The worst-case lower and upper bounds of $\Delta_{ij}$ for $p = 2$ is derived in the similar fashion as $p = \infty$. Here, the perturbation allows $\|\tilde{\mathbf{X}}_i - \mathbf{X}_i\|_2 \leq \delta$. Hence,

$$|\langle \boldsymbol{\Gamma}_i, \mathbf{X}_j \rangle| \leq \|\boldsymbol{\Gamma}_i\|_2 \|\mathbf{X}_j\|_2 \leq \delta \|\mathbf{X}_j\|_2$$
$$|\langle \boldsymbol{\Gamma}_i, \boldsymbol{\Gamma}_j \rangle| \leq \|\boldsymbol{\Gamma}_i\|_2 \|\boldsymbol{\Gamma}_j\|_2 \leq \delta^2. \tag{24}$$

Using Eq. (24), we derive the lower and upper bounds of $\Delta_{ij}$:

$$\Delta_{ij}^{L} = \begin{cases} 0+ & \text{if } i,j \notin \mathcal{U} \\ -\delta\|\mathbf{X}_j\|_2+ & \text{if } i \in \mathcal{U} \\ -\delta\|\mathbf{X}_i\|_2+ & \text{if } j \in \mathcal{U} \\ -\delta^2 & \text{if } i,j \in \mathcal{U} \end{cases} \qquad \Delta_{ij}^{U} = \begin{cases} 0+ & \text{if } i,j \notin \mathcal{U} \\ \delta\|\mathbf{X}_j\|_2+ & \text{if } i \in \mathcal{U} \\ \delta\|\mathbf{X}_i\|_2+ & \text{if } j \in \mathcal{U} \\ \delta^2 & \text{if } i,j \in \mathcal{U} \end{cases}$$

$\square$

**Case $(iii)$: Derivation of Lemma 3 for $p = 1$.** The worst-case lower and upper bounds of $\Delta_{ij}$ for $p = 1$ is derived in the similar fashion as $p = \infty$. Here, the perturbation allows $\|\tilde{\mathbf{X}}_i - \mathbf{X}_i\|_1 \leq \delta$. Hence,

$$|\langle \boldsymbol{\Gamma}_i, \mathbf{X}_j \rangle| \leq \|\boldsymbol{\Gamma}_i\|_1 \|\mathbf{X}_j\|_\infty \leq \delta \|\mathbf{X}_j\|_\infty$$
$$|\langle \boldsymbol{\Gamma}_i, \boldsymbol{\Gamma}_j \rangle| \leq \|\boldsymbol{\Gamma}_i\|_2 \|\boldsymbol{\Gamma}_j\|_2 \leq \|\boldsymbol{\Gamma}_i\|_1 \|\boldsymbol{\Gamma}_j\|_1 \leq \delta^2. \tag{25}$$

Using Eq. (25), we derive the lower and upper bounds of $\Delta_{ij}$:

$$\Delta_{ij}^{L} = \begin{cases} 0+ & \text{if } i,j \notin \mathcal{U} \\ -\delta\|\mathbf{X}_j\|_\infty+ & \text{if } i \in \mathcal{U} \\ -\delta\|\mathbf{X}_i\|_\infty+ & \text{if } j \in \mathcal{U} \\ -\delta^2 & \text{if } i,j \in \mathcal{U} \end{cases} \qquad \Delta_{ij}^{U} = \begin{cases} 0+ & \text{if } i,j \notin \mathcal{U} \\ \delta\|\mathbf{X}_j\|_\infty+ & \text{if } i \in \mathcal{U} \\ \delta\|\mathbf{X}_i\|_\infty+ & \text{if } j \in \mathcal{U} \\ \delta^2 & \text{if } i,j \in \mathcal{U} \end{cases}$$

$\square$

### D.1 BOUNDING $E_{ij}$ AND $\dot{E}_{ij}$ IN THE NTK

NTKs for GNNs with non-linear ReLU activation have $\mathbf{E}$ and $\dot{\mathbf{E}}$ with non-linear $\kappa_1(z)$ and $\kappa_0(z)$ functions in their definitions, respectively. In order to bound the NTK, we need a strategy to bound these quantities as well. In this section, we discuss our approach to bound $E_{ij}$ and $\dot{E}_{ij}$ through bounding the functions for any GNN with $L$ layers. For ease of exposition, we ignore the layer indexing for the terms of interest and it is understood from the context. Recollect that the definitions of $\mathbf{E}$ and $\dot{\mathbf{E}}$ are based on $\boldsymbol{\Sigma}$, which is a linear combination of $\mathbf{S}$ and the previous layer. So, we consider that at this stage, we already have $\boldsymbol{\Sigma}$, $\boldsymbol{\Sigma}^L$ and $\boldsymbol{\Sigma}^U$. Now, we expand the functions in the

definition and write $E_{ij}$ and $\dot{E}_{ij}$ using their corresponding $\Sigma$ as follows:

$$E_{ij} = \frac{\sqrt{\Sigma_{ii}\Sigma_{jj}}}{\pi} \left( \frac{\Sigma_{ij}}{\sqrt{\Sigma_{ii}\Sigma_{jj}}} \left( \pi - \arccos\left( \frac{\Sigma_{ij}}{\sqrt{\Sigma_{ii}\Sigma_{jj}}} \right) \right) + \sqrt{1 - \frac{\Sigma_{ij}^2}{\Sigma_{ii}\Sigma_{jj}}} \right) \qquad (26)$$

$$\dot{E}_{ij} = \frac{1}{\pi} \left( \pi - \arccos\left( \frac{\Sigma_{ij}}{\sqrt{\Sigma_{ii}\Sigma_{jj}}} \right) \right) \qquad (27)$$

We derive the lower and upper bounds for $E_{ij}$ and $\dot{E}_{ij}$ in Algorithm 1.

---

**Algorithm 1** Procedure to compute $E_{ij}^L$, $E_{ij}^U$, $\dot{E}_{ij}^L$ and $\dot{E}_{ij}^U$

---

Given $\Sigma$, $\Sigma^L$ and $\Sigma^U$
Let $s^l = \sqrt{\Sigma_{ii}^L \Sigma_{jj}^L}$, $s^u = \sqrt{\Sigma_{ii}^U \Sigma_{jj}^U}$
**if** $\Sigma_{ij}^L > 0$ **then**
$\quad a^l = \dfrac{\Sigma_{ij}^L}{s^u}, a^u = \dfrac{\Sigma_{ij}^U}{s^l}$
**else**
$\quad a^l = \dfrac{\Sigma_{ij}^L}{s^l}, a^u = \dfrac{\Sigma_{ij}^U}{s^u}$
**end if**
**if** $|\Sigma_{ij}^U| > |\Sigma_{ij}^L|$ **then**
$\quad b^l = \left( \dfrac{\Sigma_{ij}^L}{s^u} \right)^2, b^u = \left( \dfrac{\Sigma_{ij}^U}{s^l} \right)^2$
**else**
$\quad b^l = \left( \dfrac{\Sigma_{ij}^L}{s^l} \right)^2, b^u = \left( \dfrac{\Sigma_{ij}^U}{s^u} \right)^2$
**end if**
$E_{ij}^L = \frac{s^l}{\pi} \left( a^l \left( \pi - \arccos\left( a^l \right) \right) + \sqrt{1 - b^u} \right)$
$E_{ij}^U = \frac{s^u}{\pi} \left( a^u \left( \pi - \arccos\left( a^u \right) \right) + \sqrt{1 - b^l} \right)$
$\dot{E}_{ij}^L = \frac{1}{\pi} \left( \pi - \arccos\left( a^l \right) \right)$
$\dot{E}_{ij}^L = \frac{1}{\pi} \left( \pi - \arccos\left( a^u \right) \right)$

---

### D.2 DERIVATION OF THEOREM 2: NTK BOUNDS ARE TIGHT

We analyze the tightness of NTK bounds by deriving conditions on graph $\mathcal{G} = (\mathbf{S}, \mathbf{X})$ when $\Delta_{ij}^L$ and $\Delta_{ij}^U$ are attainable exactly. As our NTK bounding strategy is based on bounding the adversarial perturbation $\widetilde{\mathbf{X}}\widetilde{\mathbf{X}}^T$ and the non-linear functions $\kappa_0(z)$ and $\kappa_1(z)$, it is easy to see that the bounds with non-linearities cannot be tight. So, we consider only linear GCN (=SGC), (A)PPNP and MLP with linear activations.

Now, we focus on deriving conditions for the given node features $\mathbf{X}$ using the classic result on the equality condition of Hölder's inequality (Steele, 2004), and then analyze the NTK bounds. Steele (2004, Fig. 9.1) shows that the bounds on $\langle \mathbf{a}, \mathbf{b} \rangle$ using the Höder's inequality is reached when $|\mathbf{a}_i|^p = |\mathbf{b}_i|^q \frac{\|\mathbf{a}\|_p^p}{\|\mathbf{b}\|_q^q}$. Using this, we analyze

$$\Delta_{ij} = \langle \boldsymbol{\Gamma}_i, \mathbf{X}_j \rangle + \langle \boldsymbol{\Gamma}_j, \mathbf{X}_i \rangle + \langle \boldsymbol{\Gamma}_i, \boldsymbol{\Gamma}_j \rangle \qquad (28)$$

in which we call $\langle \boldsymbol{\Gamma}_i, \boldsymbol{\Gamma}_j \rangle$ as interaction term. Following this analysis, the tightness of NTK bounds is derived below for $p = \infty$ and $p = 2$.

**Case** $(i)$**:** $p = \infty$**.** In this case, the feature bounds in Eq. (21) are tight,

$$\forall j, \ \mathbf{X}_j \neq 0 \text{ and } \forall i, k \ \boldsymbol{\Gamma}_{ik} = c_i$$

where $c_i$ is some constant such that $\|\mathbf{\Gamma}_i\|_\infty \leq \delta$ so the perturbation budget is satisfied. As a result, the upper bound of $\Delta_{ij}$ in Lemma 1 is achieved exactly in the following cases,

**(a)** Number of adversarial nodes = 1: Here the interaction term in Eq. (28) is 0 for all $i$ and $j$. Then for the one adversarial node $i$, there exists $\mathbf{X}_j \in \mathbb{R}^d_+$, one can set $\mathbf{\Gamma}_i = +\delta \mathbf{1}_d$ to achieve the upper bound.

**(b)** Number of adversarial nodes > 1: Here the interaction term is $\neq 0$ for all the adversarial nodes $i$ and $j$. Then, for the adversarial nodes $i$ and $j$ if there exist $\mathbf{X}_i \in \mathbb{R}^d_+$ and $\mathbf{X}_j \in \mathbb{R}^d_+$ then for $\mathbf{\Gamma}_i = \mathbf{\Gamma}_j = +\delta \mathbf{1}_d$ upper bounds are achieved.

The NTKs with linear activations $Q_{ij}$ achieve the upper bound in these cases. Similarly, the lower bound in Lemma 1 is achieved exactly as discussed in the following,

**(a)** Number of adversarial nodes = 1: Here the interaction term in Eq. (28) is 0 for all $i$ and $j$. Then for the adversarial node $i$, there exists $\mathbf{X}_j \in \mathbb{R}^d_+$, one can set $\mathbf{\Gamma}_i = -\delta \mathbf{1}_d$ to achieve the lower bound.

**(b)** Number of adversarial nodes > 1: Here the interaction term is $\neq 0$ for all the adversarial nodes $i$ and $j$. Then, for the adversarial nodes $i$ and $j$ if there exist $\mathbf{X}_i \in \mathbb{R}^d_+$ and $\mathbf{X}_j \in \mathbb{R}^d_-$ then for $\mathbf{\Gamma}_i = -\delta \mathbf{1}_d$ and $\mathbf{\Gamma}_j = +\delta \mathbf{1}_d$,

leading to tight lower bounds of Lemma 1. The lower and upper tight bounds of $\Delta$ together lead to tight NTK bounds for linear activations. Note that there is no need to impose any structural restriction on the graph $\mathbf{S}$ to achieve the tight bounds for NTK.

**Case** $(ii)$**:** $p = 2$**.** In this case, the feature bounds in Eq. (24) are tight,

$$\forall i, j, \ \mathbf{X}_j \text{ and } \mathbf{\Gamma}_i \text{ are linearly dependent}$$

and $\|\mathbf{\Gamma}_i\|_2 \leq \delta$ so the perturbation budget is satisfied. As a result, the upper bound of $\Delta_{ij}$ in Lemma 2 is achieved exactly in the following,

**(a)** Number of adversarial nodes = 1: Here the interaction term in Eq. (28) is 0 for all $i$ and $j$. Then for the one adversarial node $i$, and any $\mathbf{X}_j \in \mathbb{R}^d$, one can set $\mathbf{\Gamma}_i = +\delta \frac{\mathbf{X}_j}{\|\mathbf{X}_j\|_2}$ to achieve the upper bound.

**(b)** Number of adversarial nodes > 1: Here the interaction term is $\neq 0$ for all the adversarial nodes $i$ and $j$. Then, for the adversarial nodes $i$ and $j$, if there exist $\mathbf{X}_i \in \mathbb{R}^d_+$ and $\mathbf{X}_j \in \mathbb{R}^d_+$ are linearly dependent, then for $\mathbf{\Gamma}_i = +\delta \frac{\mathbf{X}_j}{\|\mathbf{X}_j\|_2}$ and $\mathbf{\Gamma}_j = +\delta \frac{\mathbf{X}_i}{\|\mathbf{X}_i\|_2}$ tight upper bound is achieved.

The NTKs with linear activations $Q_{ij}$ achieve the worst-case upper bound in these cases. Similarly, the lower bound in Lemma 2 is achieved exactly as discussed in the following,

**(a)** Number of adversarial nodes = 1: Here the interaction term in Eq. (28) is 0 for all $i$ and $j$. Then for the adversarial node $i$, and any $\mathbf{X}_j \in \mathbb{R}^d$, one can set $\mathbf{\Gamma}_i = -\delta \frac{\mathbf{X}_j}{\|\mathbf{X}_j\|_2}$ to achieve the lower bound.

**(b)** Number of adversarial nodes > 1: Here the interaction term is $\neq 0$ for all the adversarial nodes $i$ and $j$. Then, for the adversarial nodes $i$ and $j$, if there exist $\mathbf{X}_i \in \mathbb{R}^d_+$ and $\mathbf{X}_j \in \mathbb{R}^d_-$ are linearly dependent, then for $\mathbf{\Gamma}_i = -\delta \frac{\mathbf{X}_j}{\|\mathbf{X}_j\|_2}$ and $\mathbf{\Gamma}_i = +\delta \frac{\mathbf{X}_i}{\|\mathbf{X}_j\|_2}$,

leading to tight lower bounds of Lemma 2. The lower and upper tight bounds of $\Delta$ together leads to tight NTK bounds for linear activations. Note that there is no need to impose any structural restriction on the graph $\mathbf{S}$ to achieve the tight bounds for NTK, same as the $p = \infty$ case. We further note that only one instance of achieving the worst-case bound is stated, and one can construct similar cases, for example by considering opposite signs for the features and perturbations.

**Case** $(iii)$**:** $p = 1$**.** In this case, the feature bounds in Eq. (25) are tight,

$$\forall j, \ \mathbf{X}_j \text{ and } \forall i, k \ \mathbf{\Gamma}_{ik} = c \mathbb{1}[k = \arg\max_{k'} \mathbf{X}_j]$$

where $c = \delta$ to satisfy $\|\mathbf{\Gamma}_i\|_1 \leq \delta$. As a result, the upper bound of $\Delta_{ij}$ in Lemma 3 is achieved exactly in the following cases,

**(a)** Number of adversarial nodes = 1: Here the interaction term in Eq. (28) is 0 for all $i$ and $j$. Then for the one adversarial node $i$, for any $j$, $\mathbf{X}_j \neq 0$ and $\arg\max \mathbf{X}_j = k$, one can set $\mathbf{\Gamma}_{ik} = \text{sgn}(X_{jk})\delta$ and $\mathbf{\Gamma}_{ik'} = 0 \ \forall \ k' \neq k$ to achieve the upper bound.

**(b)** Number of adversarial nodes $> 1$: Here the interaction term is $\neq 0$ for all the adversarial nodes $i$ and $j$. Then, for the adversarial nodes $i$ and $j$ if there exist $\arg\max \mathbf{X}_i = \arg\max \mathbf{X}_j = k$ and $\mathrm{sgn}(X_{ik}) = \mathrm{sgn}(X_{jk})$ then for $\mathbf{\Gamma}_{ik} = \mathbf{\Gamma}_{jk} = \mathrm{sgn}(X_{ik})\delta$ and $\forall k' \neq k$, $\mathbf{\Gamma}_{ik'} = \mathbf{\Gamma}_{jk'} = 0$ upper bounds are achieved.

The NTKs with linear activations $Q_{ij}$ achieve the upper bound in these cases. Similarly, the lower bound in Lemma 3 is achieved exactly as discussed in the following,

**(a)** Number of adversarial nodes $= 1$: Here the interaction term in Eq. (28) is $0$ for all $i$ and $j$. Then for the one adversarial node $i$, for any $j$, $\mathbf{X}_j \neq 0$ and $\arg\max \mathbf{X}_j = k$, one can set $\mathbf{\Gamma}_{ik} = -\mathrm{sgn}(X_{jk})\delta$ and $\mathbf{\Gamma}_{ik'} = 0 \ \forall \ k' \neq k$ to achieve the lower bound.

**(b)** Number of adversarial nodes $> 1$: Here the interaction term is $\neq 0$ for all the adversarial nodes $i$ and $j$. Then, for the adversarial nodes $i$ and $j$ if there exist $\arg\max \mathbf{X}_i = \arg\max \mathbf{X}_j = k$ and $\mathrm{sgn}(X_{ik}) = -\mathrm{sgn}(X_{jk})$ then for $\mathbf{\Gamma}_{ik} = -\mathrm{sgn}(X_{jk})\delta$, $\mathbf{\Gamma}_{jk} = -\mathrm{sgn}(X_{jk})\delta$ and $\forall k' \neq k$, $\mathbf{\Gamma}_{ik'} = \mathbf{\Gamma}_{jk'} = 0$,

leading to tight lower bounds of Lemma 3. The lower and upper tight bounds of $\Delta$ together leads to tight NTK bounds for linear activations. Again, there is no need to impose any structural restriction on the graph $\mathbf{S}$ to achieve the tight bounds for NTK. $\qquad\square$

# E  MULTI-CLASS CERTIFICATION

In this section, we discuss the certification for multi-class. We abstract the NN and work with NTK here. Hence, to do multi-class classification using SVM with NTK, we choose One-Vs-All strategy, where we learn $K$ classifiers. Formally, we learn $\boldsymbol{\beta}^1, \ldots, \boldsymbol{\beta}^K$ which has corresponding duals $\boldsymbol{\alpha}^1, \ldots, \boldsymbol{\alpha}^K$. In order to learn $\boldsymbol{\beta}^c$, all samples with class label $c$ are assumed to be positive and the rest negative. Assume from hence on that for all $c$, $\boldsymbol{\beta}^c$ corresponds to the optimal solution with the corresponding dual $\boldsymbol{\alpha}^c$. Then the prediction for a node $t$ is $c^* = \arg\max_c \hat{p}_t^c$ where $\hat{p}_t^c = \sum_{i=1}^m y_i \alpha_i^c Q_{ti}$ where $\boldsymbol{Q}$ is the NTK matrix.

Given this, we propose a simple extension of our binary certification where to certify a node $t$ as provably robust, we minimize the MILP objective in Theorem 1 for the predicted class $c^*$ and maximize the objective for the remaining $K - 1$ classes. Finally, certify $t$ to be provably robust only if the objective for $c^*$ remains maximum. Formally, we state the objective below.

**Theorem 5.** *Node $t$ with original predicted class $c^*$ is certifiably robust against adversary $\mathcal{A}$ if $c' = c^*$ where $c'$ is defined in the following. Using the MILP $P(\boldsymbol{Q})$ in Theorem 1, we define*

$$P(\boldsymbol{Q})_c := P(\boldsymbol{Q}) \text{ using } \boldsymbol{\alpha}^c, \text{ with the only change in obj. to } (-1)^{\mathbb{1}[c \neq c^*]} \sum_{i=1}^m y_i Z_{ti}$$

$$c_t' = \arg\max_{c \in [K]} P(\boldsymbol{Q})_c. \tag{29}$$

# F  PROOF OF PROPOSITION 1

We restate Proposition 1.

**Proposition 1.** *Problem $\mathrm{P}_1(\widetilde{\boldsymbol{Q}})$ given by Eq. (2) is convex and satisfies strong Slater's constraint. Consequently, the single-level optimization problem $\mathrm{P}_3(\boldsymbol{Q})$ arising from $\mathrm{P}_2(\boldsymbol{Q})$ by replacing $\boldsymbol{\alpha} \in \mathcal{S}(\widetilde{\boldsymbol{Q}})$ with Eqs. (5) to (7) has the same globally optimal solutions as $\mathrm{P}_2(\boldsymbol{Q})$.*

Given any $\widetilde{\boldsymbol{Q}} \in \mathcal{A}(\boldsymbol{Q})$. We prove two lemmas, leading us towards proving Proposition 1.

**Lemma 4.** *Problem $\mathrm{P}_1(\widetilde{\boldsymbol{Q}})$ is convex.*

*Proof.* The dual problem $\mathrm{P}_1^{\mathrm{b}}(\widetilde{\boldsymbol{Q}})$ associated do an SVM with bias term reads

$$\mathrm{P}_1^{\mathrm{b}}(\widetilde{\boldsymbol{Q}}): \ \min_{\boldsymbol{\alpha}} -\sum_{i=1}^{m} \alpha_i + \frac{1}{2}\sum_{i=1}^{m}\sum_{j=1}^{m} y_i y_j \alpha_i \alpha_j \widetilde{Q}_{ij} \ \text{s.t.} \sum_{i=1}^{m} \alpha_i y_i = 0 \ \ 0 \le \alpha_i \le C \ \forall i \in [m] \quad (30)$$

It is a known textbook result that $\mathrm{P}_1^{\mathrm{b}}(\widetilde{\boldsymbol{Q}})$ is convex and we refer to Mohri et al. (2018) for a proof. A necessary and sufficient condition for an optimization problem to be convex is that the objective function as well as all inequality constraints are convex and the equality constraints affine functions. Furthermore, the domain of the variable over which is optimized must be a convex set. As removing the bias term of an SVM results in a dual problem $\mathrm{P}_1(\widetilde{\boldsymbol{Q}})$ which is equivalent to $\mathrm{P}_1^{\mathrm{b}}(\widetilde{\boldsymbol{Q}})$ only with the constraint $\sum_{i=1}^{m} \alpha_i y_i = 0$ removed, the necessary and sufficient conditions for convexity stay fulfilled. $\qquad\square$

Now, we define strong Slater's condition for $\mathrm{P}_1(\widetilde{\boldsymbol{Q}})$ embedded in the upper-level problem $\mathrm{P}_2(\boldsymbol{Q})$ defined in Eq. (4), which we from here on will call strong Slater's constraint qualification (Dempe & Dutta, 2012).

**Def. 8** (Slater's CQ). *The lower-level convex optimization problem $\mathrm{P}_1(\widetilde{\boldsymbol{Q}})$ fulfills strong Slater's Constraint Qualification, if for any upper-level feasible $\widetilde{\boldsymbol{Q}} \in \mathcal{A}(\boldsymbol{Q})$, there exists a point $\boldsymbol{\alpha}(\widetilde{\boldsymbol{Q}})$ in the feasible set of $\mathrm{P}_1(\widetilde{\boldsymbol{Q}})$ such that no constraint in $\mathrm{P}_1(\widetilde{\boldsymbol{Q}})$ is active, i.e. $0 < \alpha(\widetilde{\boldsymbol{Q}})_i < C$ for all $i \in [m]$.*

**Lemma 5.** *Problem $\mathrm{P}_1(\widetilde{\boldsymbol{Q}})$ fulfills strong Slater's constraint qualification.*

*Proof.* We prove Lemma 5 through a constructive proof. Given any upper-level feasible $\widetilde{\boldsymbol{Q}} \in \mathcal{A}(\boldsymbol{Q})$. Let $\boldsymbol{\alpha}$ be an optimal solution to $\mathrm{P}_1(\widetilde{\boldsymbol{Q}})$. We restrict ourselves to cases, where $\mathrm{P}_1(\widetilde{\boldsymbol{Q}})$ is non-degenerate, i.e. the optimal solution to the SVM $f_\theta^{SVM}$ corresponds to a weight vector $\boldsymbol{\beta} \ne \boldsymbol{0}$. Then, at least for one index $i \in [m]$ it must hold that $\boldsymbol{\alpha}_i > 0$.

Assume that $j$ is the index in $[m]$ with the smallest $\boldsymbol{\alpha}_j > 0$. Let $\epsilon = \boldsymbol{\alpha}_j/m + 1 > 0$. Now, we construct a new $\boldsymbol{\alpha}'$ from $\boldsymbol{\alpha}$ by for each $i \in [m]$ setting:

- If $\alpha_i = 0$, set $\alpha_i' = \epsilon$.

- If $\alpha_i = C$, set $\alpha_i' = C - \epsilon$.

The new $\boldsymbol{\alpha}'$ fulfills $0 < \boldsymbol{\alpha}'(\widetilde{\boldsymbol{Q}})_i < C$ for all $i \in [m]$. If $\mathrm{P}_1(\widetilde{\boldsymbol{Q}})$ is degenerate, set $\boldsymbol{\alpha}'(\widetilde{\boldsymbol{Q}})_i = C/2$ for all $i \in [m]$. This concludes the proof. $\qquad\square$

(Dempe & Dutta, 2012) establish that any bilevel optimization problem $U$ whose lower-level problem $L$ is convex and fulfills strong Slater's constraint qualification for any upper-level feasible point has the same global solutions as another problem defined by replacing the lower-level problem $L$ in $U$ with $L$'s Karash Kuhn Tucker conditions. This, together with Lemmas 4 and 5 concludes the proof for Proposition 1. $\qquad\square$

# G  SETTING BIG-$M$ CONSTRAINTS

**Proposition 2** (Big-$M$'s). *Replacing the complementary slackness constraints Eq. (7) in $\mathrm{P}_3(\boldsymbol{Q})$ with the big-$M$ constraints given in Eq. (8) does not cut away solution values of $\mathrm{P}_3(\boldsymbol{Q})$, if for any $i \in [m]$, the big-$M$ values fulfill the following conditions. For notational simplicity $j : Condition(j)$ denotes $j \in \{j \in [m] : Condition(j)\}$.*

*If $y_i = 1$ then*

$$M_{u_i} \ge \sum_{j:y_j=1 \wedge \tilde{Q}_{ij}^U \ge 0} C\widetilde{Q}_{ij}^U - \sum_{j:y_j=-1 \wedge \tilde{Q}_{ij}^L \le 0} C\widetilde{Q}_{ij}^L - 1 \quad (31)$$

$$M_{v_i} \ge \sum_{j:y_j=-1 \wedge \tilde{Q}_{ij}^U \ge 0} C\widetilde{Q}_{ij}^U - \sum_{j:y_j=1 \wedge \tilde{Q}_{ij}^L \le 0} C\widetilde{Q}_{ij}^L + 1 \quad (32)$$

*If $y_i = -1$ then*

$$M_{u_i} \geq \sum_{j:y_j=-1 \wedge \widetilde{Q}_{ij}^U \geq 0} C\widetilde{Q}_{ij}^U - \sum_{j:y_j=1 \wedge \widetilde{Q}_{ij}^L \leq 0} C\widetilde{Q}_{ij}^L - 1 \tag{33}$$

$$M_{v_i} \geq \sum_{j:y_j=1 \wedge \widetilde{Q}_{ij}^U \geq 0} C\widetilde{Q}_{ij}^U - \sum_{j:y_j=-1 \wedge \widetilde{Q}_{ij}^L \leq 0} C\widetilde{Q}_{ij}^L + 1 \tag{34}$$

*To obtain the tightest formulation for $P(\boldsymbol{Q})$ from the above conditions, we set the big-$M$'s to equal the conditions.*

*Proof.* Denote by $UB$ an upper bound to $\sum_{j=1}^m y_i y_j Z_{ij}$ and by $LB$ a lower bound to $\sum_{j=1}^m y_i y_j Z_{ij}$. The existence of these bounds follows from $y_i$ and $y_j \in \{-1, 1\}$ and $Z_{ij} = \alpha_j \widetilde{Q}_{ij}$ with $0 \leq \alpha_j \leq C$ and $\widetilde{Q}_{ij}^L \leq \widetilde{Q}_{ij} \leq \widetilde{Q}_{ij}^U$, i.e. the boundedness of all variables.

$u_i$ and $v_i$ need to be able to be set such that $\sum_{j=1}^m y_i y_j Z_{ij} - u_i + v_i = 1$ (see Eq. (5)) can be satisfied given any $\boldsymbol{\alpha}^*$ and $\widetilde{\boldsymbol{Q}}^*$ part of an optimal solution to $P_3(\boldsymbol{Q})$. By using $UB$ and $LB$ we get the following inequalities:

$$UB - u_i + v_i \geq 1 \tag{35}$$

and

$$LB - u_i + v_i \leq 1 \tag{36}$$

Denote $\sum_{j=1}^m y_i y_j Z_{ij}$ by $T$. Thus, if $T \geq 1$, setting $v_i = 0$ and $u_i \leq UB - 1 \wedge u_i \geq LB - 1$ allows to satisfy Eq. (5). If $T < 1$, setting $u_i = 0$ and $v_i \leq 1 - LB \wedge v_i \geq 1 - UB$ allows to satisfy Eq. (5). Note that for a given $i$, we are free to set $u_i$ and $v_i$ to arbitrary positive values, as long as they satisfy Eq. (5), as they don't affect the optimal solution value nor the values of other variables.

Thus, adding $u_i \leq UB - 1$ and $v_i \leq 1 - LB$ as constraints to $P_3(\boldsymbol{Q})$ does not affect its optimal solution. Consequently, setting $M_{u_i} \geq UB - 1$ and $M_{v_i} \geq 1 - LB$, are valid big-$M$ constraints in the mixed-integer reformulation of the complementary slackness constraints $Eq.$ (7). The $UB$ and $LB$ values depend on the sign of $y_i$, $y_j$ and the bounds on $\alpha_j$ and $\widetilde{Q}_{ij}$ and the right terms in Eqs. (31) to (34) represent the respective $UB$ and $LB$ arising. This concludes the proof. $\qquad\square$

# H ADDITIONAL EXPERIMENTAL DETAILS

## H.1 DATASETS

The CSBM implementation is taken from (Gosch et al., 2023) publicly released under MIT license. Cora-ML taken from (Bojchevski & Günnemann, 2018) is also released under MIT license. Cora-ML has 2995 nodes with 8158 edges, and 7 classes. It traditionally comes with a 2879 dimensional discrete bag-of-words node feature embedding from the paper abstract. As we focus on continuous perturbation models, we use the abstracts provided by (Bojchevski & Günnemann, 2018) together with all-MiniLM-L6-v2, a modern sentence transformer from `https://huggingface.co/sentence-transformers/all-MiniLM-L6-v2` to generate 384-dimensional continuous node-feature embeddings. From Cora-ML, we extract the subgraph defined by the two most largest classes, remove singleton nodes, and call the resulting binary-classification dataset Cora-MLb. It has 1235 nodes and 2601 edges. WikiCSb, created from extracting the two largest classes from WikiCS, is the largest used dataset with 4660 nodes and 72806 edges.

### H.1.1 CSBM SAMPLING SCHEME

A CSBM graph $\mathcal{G}$ with $n$ nodes is iteratively sampled as: (a) Sample label $y_i \sim Bernoulli(1/2) \; \forall i \in [n]$; (b) Sample feature vectors $\mathbf{X}_i | y_i \sim \mathcal{N}(y_i \boldsymbol{\mu}, \sigma^2 \mathbf{I}_d)$; (c) Sample adjacency $A_{ij} \sim Bernoulli(p)$

if $y_i = y_j$, $A_{ij} \sim Bernoulli(q)$ otherwise, and $A_{ji} = A_{ij}$. Following Gosch et al. (2023) we set $p, q$ through the maximum likelihood fit to Cora (Sen et al., 2008) ($p = 3.17\%$, $q = 0.74\%$), and $\boldsymbol{\mu}$ element-wise to $K\sigma/2\sqrt{d}$ with $d = \lfloor n/\ln^2(n) \rfloor$, $\sigma = 1$, and $K = 1.5$, resulting in an interesting classification scheme where both graph structure and features are necessary for good generalization. We sample $n = 200$ and choose 40 nodes per class for training, leaving 120 unlabeled nodes.

## H.2 ARCHITECTURES.

We fix $\mathbf{S}$ to $\mathbf{S}_{\text{row}}$ for GCN, SGC, GCN Skip-$\alpha$ and GCN Skip-PC following (Sabanayagam et al., 2023), $\mathbf{S}_{\text{sym}}$ for APPNP as its default implementation. From the GNN definitions App. A, the graph structure for GIN is $(1 + \epsilon)\mathbf{I} + \mathbf{A}$, for GraphSAGE is $\mathbf{I} + \mathbf{D}^{-1}\mathbf{A}$.

We outline the hyperparameters for Cora-MLb, for CSBM all parameters are mentioned in Sec. 4 except the Skip-$\alpha$ for GCN Skip-$\alpha$ which was set to 0.2.

- GCN (Row Norm.): $C = 0.75$
- GCN (Sym. Norm.): $C = 1$
- SGC (Row Norm.): $C = 0.75$
- SGC (Sym Norm.): $C = 0.75$
- APPNP (Sym. Norm.): $C = 1$, $\alpha = 0.1$
- MLP: $C = 0.5$
- GCN Skip-$\alpha$: $C = 1$, $\alpha = 0.1$
- GCN Skippc: $C = 0.5$

For Cora-ML, the following hyperparameters were set:

- GCN (Row Norm.): $C = 0.05$
- SGC (Row Norm.): $C = 0.0575$
- MLP: $C = 0.004$

For WikiCSb:

- GCN (Row Norm.): $C = 1$
- SGC (Row Norm.): $C = 5$
- APPNP (Sym. Norm.): $C = 0.75$, $\alpha = 0.1$
- MLP: $C = 0.175$
- GCN Skip-$\alpha$: $C = 0.1$, $\alpha = 0.1$
- GCN Skippc: $C = 1$

Hyperparameters were set using 4-fold cross-validation and and choosing the result with lowest $C$ in the standard deviation of the best validation accuracy, to reduce runtime of the MILP certification process.

## H.3 HARDWARE.

Experiments are run on CPU using Gurobi on an internal cluster. Experiments for CSBM, Cora-MLb and WikiCSb do not require more than 15GB of RAM. Cora-ML experiments do not require more than 20GB of RAM. The time to certify a node depends on the size of MILP as well as the structure of the concrete problem. On our hardware, for CSBM and Cora-MLb certifying one node typically takes several seconds up to one minute on a single CPU. For Cora-ML, certifying a node can take between one minute and several hours ($\leq 10$) using two CPUs depending on the difficulty of the associated MILP.

# I  ADDITIONAL RESULTS: CSBM

## I.1  EVALUATING QPCERT AND IMPORTANCE OF GRAPH INFORMATION

Fig. 5a shows the same result as Fig. 2a from Sec. 4 establishing that including graph information boosts worst-case robustness in CSBM too. This also shows that the result is not dataset-specific. In Fig. 5, we provide the remaining settings in correspondence to Fig. 3, Poison Labeled $PL$ and Backdoor Labeled $BL$ for CSBM. Similarly, the heatmaps showing the certified accuracy gain with respect to MLP is presented in Fig. 6.

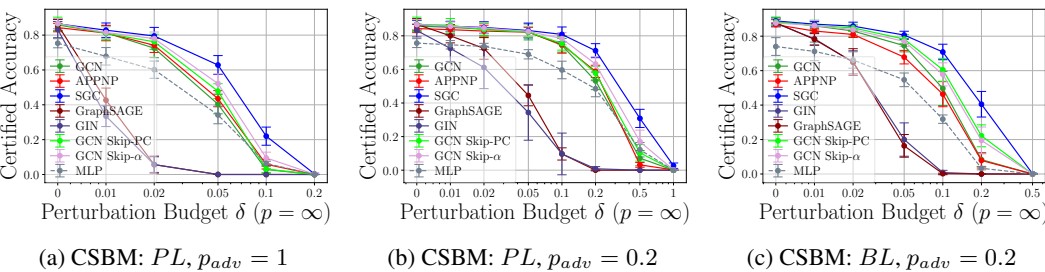

(a) CSBM: $PL$, $p_{adv} = 1$    (b) CSBM: $PL$, $p_{adv} = 0.2$    (c) CSBM: $BL$, $p_{adv} = 0.2$

Figure 5: Certifiable robustness for different (G)NNs in Poisoning Labeled (PL) and Backdoor Labeled (BL) setting.

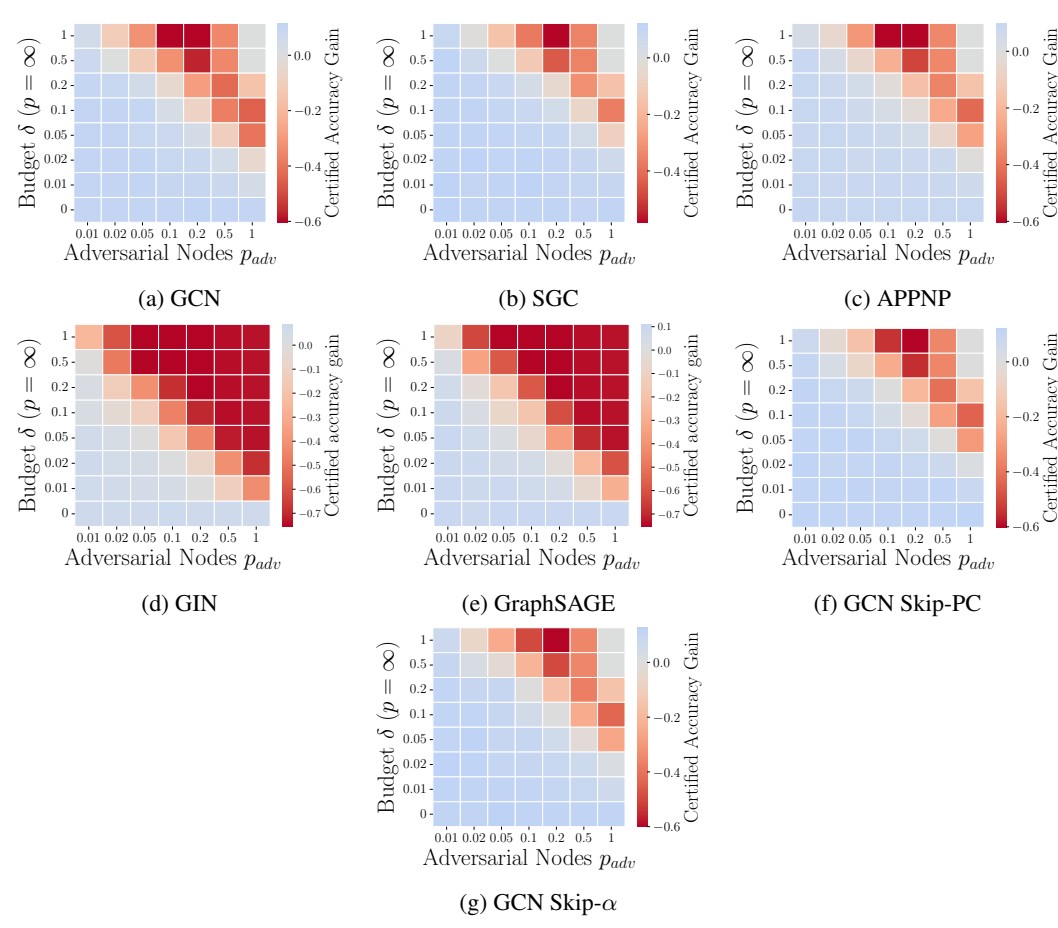

(a) GCN    (b) SGC    (c) APPNP

(d) GIN    (e) GraphSAGE    (f) GCN Skip-PC

(g) GCN Skip-$\alpha$

Figure 6: Heatmaps of different GNNs for Poison Unlabeled ($PU$) setting.

## I.2 ON GRAPH CONNECTIVITY AND ARCHITECTURAL INSIGHTS

We present the sparsity analysis for SGC and APPNP in (a) and (b) of Fig. 7, showing a similar observation to GCN in App. I.2. The APPNP $\alpha$ analysis for $PU$ and $PL$ are provided in (c) and (d) of Fig. 7, showing the inflection point in $PU$ but not in $PL$. Additionally, we show the influence of depth, linear vs ReLU, regularization $C$ and row vs symmetric normalized adjacency in Fig. 8.

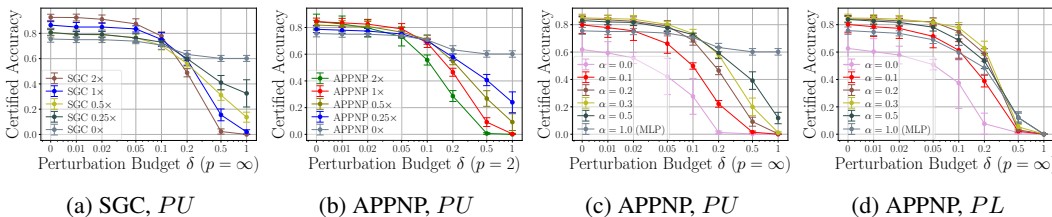

| (a) SGC, $PU$ | (b) APPNP, $PU$ | (c) APPNP, $PU$ | (d) APPNP, $PL$ |

Figure 7: (a)-(b): Graph connectivity analysis where $c\times$ is $cp$ and $cq$ in CSBM model. GCN is provided in Fig. 4b. (c)-(d): APPNP analysis based on $\alpha$.

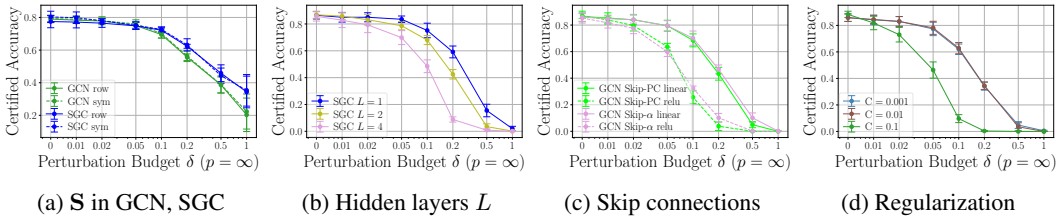

| (a) **S** in GCN, SGC | (b) Hidden layers $L$ | (c) Skip connections | (d) Regularization |

Figure 8: (a): Symmetric and row normalized adjacencies as the choice for **S** in GCN and SGC. (b): Effect of number of hidden layers $L$. (c): Linear and relu for the Skip-PC and Skip-$\alpha$. (d): Regularization $C$ in GCN. All experiments in $PU$ setting and $p_{adv} = 0.2$.

## I.3 TIGHTNESS OF QPCERT

First, we present the tightness of QPCert in Figs. 9 and 10 evaluated with our strongest employed attacks for each setting: For graph poisoning (Fig. 9), APGD is employed with direct differentiation through the learning process (QPLayer) for the $PU$ setting in Fig. 9b and for the $PL$ setting in Fig. 9a. For the backdoor attack setting (Fig. 10), first a poisoning attack is carried out with APGD (QPLayer) and then, the respective test node is additionally attacked with APGD in an evasion setting. Fig. 10b shows the result for the $BU$ setting and Fig. 10a for the $BL$ setting. Interstingly, QPCert seems to be more tight in a backdoor setting than in a pure poisoning setting. However, this could also be explained by the fact that an evasion attack is easier to perform than a poisoning attack and thus, APGD potentially provided lower upper bounds to the actual robustness than for the pure poisoning setting. Another interesting observation is that for the backdoor settings, the rankings of the GNNs regarding certified robustness seems to roughly correspond to the robust accuracies obtained by the backdoor attack.

In Fig. 11 performing a gradient-based attack (APGD) using either exact gradients with QPLayer or meta-gradients obtained through MetaAttack's surrogate model is compared. For both the PL (Fig. 11a) and PU (Fig. 11b) setting using exact gradients results in a lower upper bound to the robust accuracy (i.e., a stronger attack). Thus, we use the exact gradients from QPLayer to measure the tightness of QPCert. Meta-gradients from MetaAttack are obtained by adapting Algorithm 2 in (Zügner & Günnemann, 2019a) to feature perturbations, through setting a maximum number of iterations as the stop criterion and instead of choosing an edge with maximal score, update the feature matrix with the meta-gradient using APGD. In MetaAttack, $\lambda$ trading of the self-supervised with the training loss is set to $0.5$. Interestingly, for small budgets, MetaAttack can lead to the opposite intended effect. Exemplary, for a GCN in the PL setting with $\delta = 0.1$, the generalization performance is slightly increased. This indicates that for small perturbation budgets, the meta-gradient of MetaAttack's surrogate model does not transfer well to the infinite-width networks. However, for

larger budgets, MetaAttack still provides a strong, albeit weaker attack than exact gradients. In Figure Fig. 12 we compare performing the above mentioned gradient-based backdoor attack with the simple backdoor strategy proposed by Xing et al. (2024) with a trigger size of $0.5$. We observe that Xing et al. (2024)'s attack is significantly weaker compared to the gradient-based attack and only starts to reduce accuracy of the models for high attack budgets. This can be explained by several observations: For small $\ell_p$-budgets, the backdoor trigger is often distorted in the backdoored nodes by having to project the perturbation back into the allowed $\ell_p$-ball and secondly, the attack is simple, static and not adaptive. Concretely, it simply copies certain features to other nodes without considering the attacked model. We want to note that similar to MetaAttack, for small budgets, for BU we can observe for MLP that the change actually results in slightly higher generalization of the model under attack, showing that for small budgets, the backdoor strategy in Xing et al. (2024) is not effective.

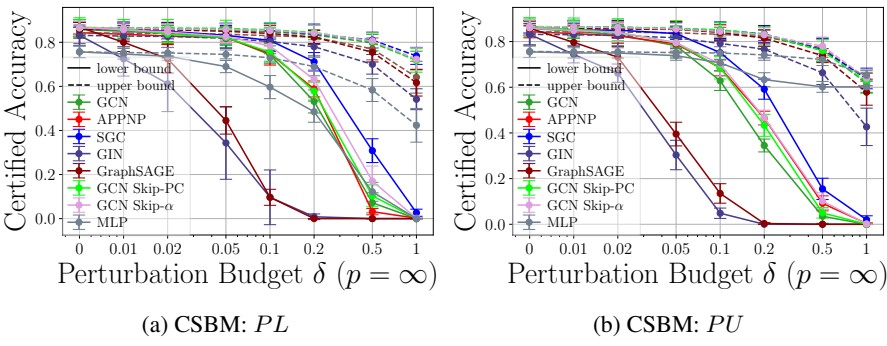

(a) CSBM: $PL$        (b) CSBM: $PU$

Figure 9: Tightness of our certificate for data poisoning. Both $PU$ and $PL$ with $p_{adv} = 0.2$ evaluated with APGD (QPLayer).

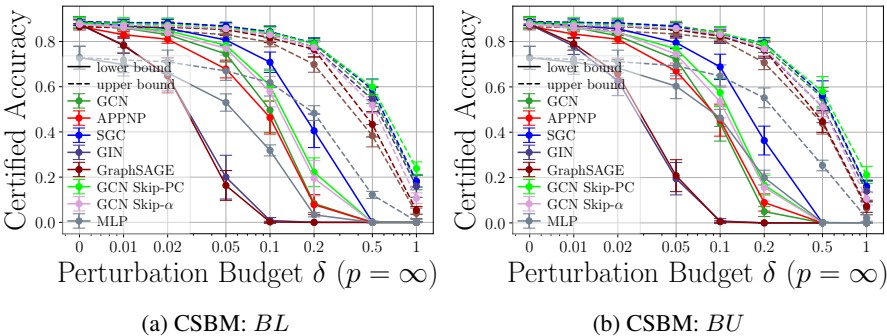

(a) CSBM: $BL$        (b) CSBM: $BU$

Figure 10: Tightness of our certificate for backdoor attacks. Both $BU$ and $BL$ again with $p_{adv} = 0.2$.

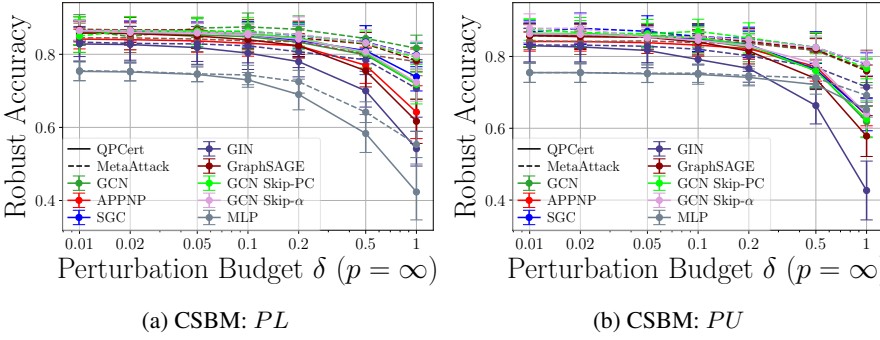

(a) CSBM: $PL$        (b) CSBM: $PU$

Figure 11: Comparison of performing a gradient-based attack (APGD) using either exact gradients using QPLayer or using surrogate meta-gradients using MetaAttack's surrogate model. Both $PL$ and $PU$ with $p_{adv} = 0.2$.

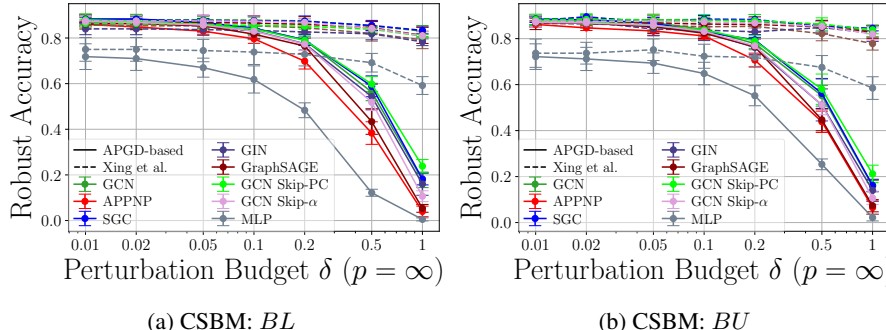

(a) CSBM: $BL$          (b) CSBM: $BU$

Figure 12: Comparison of performing a gradient-based backdoor attack versus the simple backdoor attack proposed in Xing et al. (2024).

### I.4 RESULTS FOR $p = 2$ PERTURBATION BUDGET

We present the results for $p = 2$ perturbation budget evaluated on CSBM and all the GNNs considered. We focus on Poison Unlabeled setting. Fig. 13 show the results of the certifiable robustness for all GNNs and the heatmaps showing the accuracy gain with respect to MLP is in Fig. 13. All the results are in identical to $p = \infty$ setting and we do not see any discrepancy.

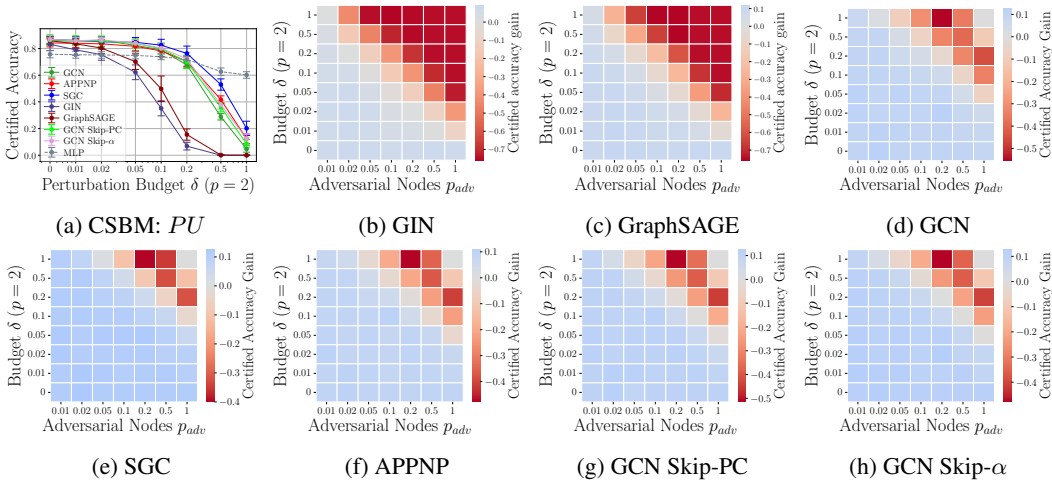

(a) CSBM: $PU$     (b) GIN     (c) GraphSAGE     (d) GCN

(e) SGC     (f) APPNP     (g) GCN Skip-PC     (h) GCN Skip-$\alpha$

Figure 13: (a): Certifiable robustness for different (G)NNs in Poisoning Unlabeled (PU) for $p = 2$. (b)-(h): Certified accuracy gain for heatmap for all GNNs. All experiments with Poisoning Unlabeled (PU) and $p_{adv} = 0.2$

### I.5 RESULTS FOR $p = 1$ PERTURBATION BUDGET

Similar to $p = 2$, we also present the results for $p = 1$ perturbation budget evaluated on CSBM and all the GNNs considered for Poison Unlabeled setting in Fig. 14. All the results are in identical to $p = \infty$ setting and we do not see any discrepancy.

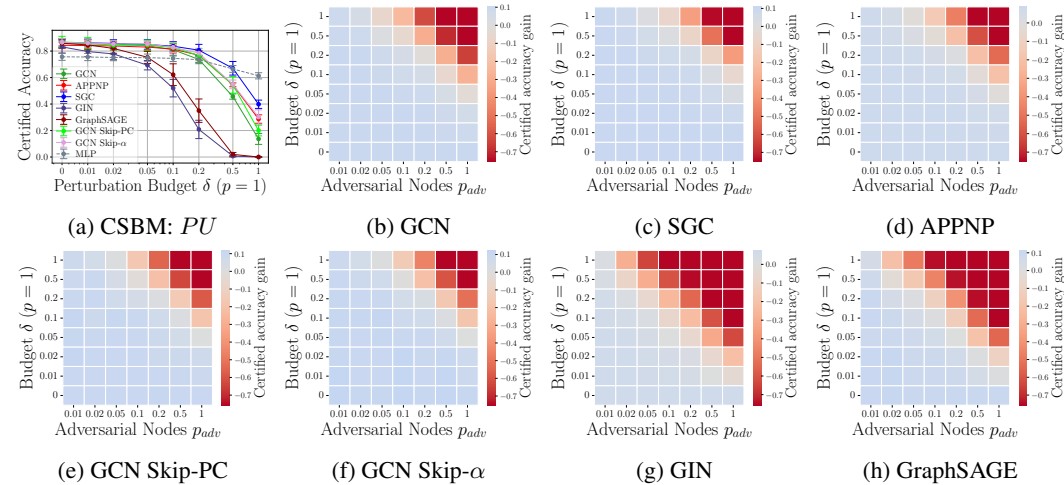

(a) CSBM: $PU$        (b) GCN        (c) SGC        (d) APPNP

(e) GCN Skip-PC     (f) GCN Skip-$\alpha$     (g) GIN     (h) GraphSAGE

Figure 14: (a): Certifiable robustness for different (G)NNs in Poisoning Unlabeled (PU) for $p = 1$. (b)-(h): Certified accuracy gain for heatmap for all GNNs. All experiments with Poisoning Unlabeled (PU) and $p_{adv} = 0.2$.

## I.6 COMPARISON BETWEEN $p = \infty$ AND $p = 2$

We provide a comparison between $p = \infty$ and $p = 2$ perturbation budget, showing that $p = 2$ is tighter than $p = \infty$ for the same budget as expected.

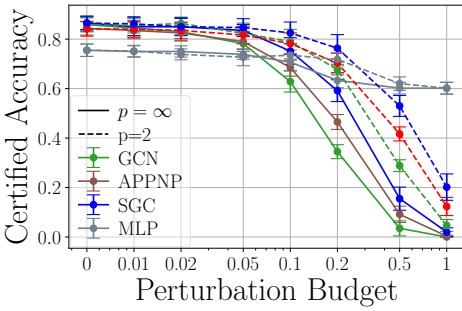

Figure 15: Comparison between $p = \infty$ and $p = 2$ for Poison Unlabeled setting. $p_{adv} = 0.2$.

## I.7 COMPARISON TO COMMON POISONING DEFENSES

In Fig. 16 we compare two common poisoning defenses namely GNNGuard (Zhang & Zitnik, 2020) and ElasticGNN (Liu et al., 2021) with the certified accuracy provided by QPCert. While the accuracies provided by a defense are not certified accuracies (i.e., they are only upper bounds to the true robust accuracy) and hence, can only be compared partly with the certified accuracy which represents a true lower bound to the robust accuracy. However, a comparison is still interesting as it allows to answer the question, of how big the gap between the best-certified accuracy to the robust accuracy provided by defenses is and if we could even get a certified accuracy result comparable to a poisoning defense's accuracy. Interestingly, Fig. 16 shows that for small to intermediate budgets, the certified accuracy of an infinite-width GCN as provided by QPCert is higher than the robust accuracy provided by the defense baselines. This can be explained by the fact that even ElasticGNN and GNNGuard show lower base clean accuracy despite significant hyperparameter tuning (experimental details see below paragraph) paired with a few very brittle predictions. We hypothesize that this is due to the difficult learning problem a CSBM poses (despite being a small dataset) paired with the fact that both poisoning defenses have GCN-like base models where the graph / propagation scheme is adapted to be more robust to poisoning while potentially trading off clean accuracy.

Both poisoning defenses are trained using the non-negative likelihood loss and the ADAM optimizer following Zhang & Zitnik (2020). GNNGuard uses a 2-layer GCN as a baseline model and hyperparameters are searched in the grid: $(i)$ number of filters $\{8, 16, 32\}$, $(ii)$ dropout $\{0, 0.2, 0.5\}$, $(iii)$ learning rate $\{0.01, 0.001\}$, $(iv)$ weight decay $\{5e-3, 1e-3, 5e-4, 1e-4\}$ over 10 seeds resulting in 720 models. For ElasticGNN the hyperparameter grid reported in Liu et al. (2021) is explored over 10 seeds resulting in 11520 models due to ElasticGNN having more hyperparameters to tune. It's hidden layer size is fixed to 32. Both baseline defenses are attacked using MetaAttack adapted to feature perturbations as done in App. I.3. The infinite-width GCN is attacked using the exact gradient obtained from the QPLayer implementation.

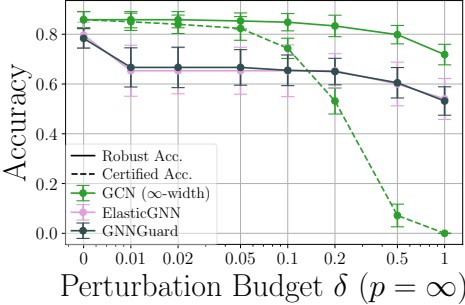

Figure 16: Comparison of different poisoning defenses with the certified accuracy obtained by QPCert.

## J ADDITIONAL RESULTS: CORA-MLB

### J.1 EVALUATING QPCERT

Fig. 17a shows the certified accuracy on Cora-MLb for the $BL$ settings for $p_{cert} = 0.1$. Figs. 17b to 17d and 18 show a detailed analysis into the certified accuracy difference of different GNN architectures for PU setting for $p_{cert} = 0.1$.

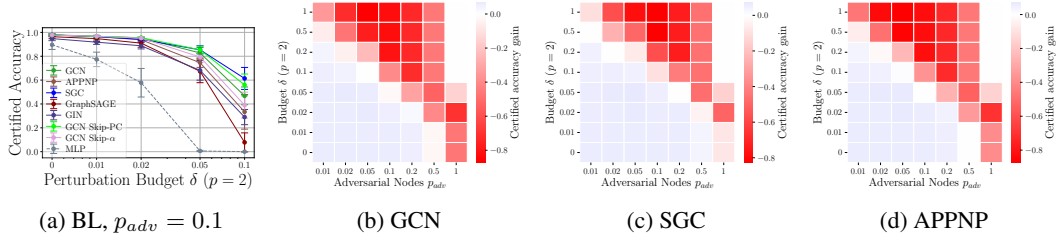

| (a) BL, $p_{adv} = 0.1$ | (b) GCN | (c) SGC | (d) APPNP |

Figure 17: (a) Backdoor Labeled ($BL$) Setting. (b)-(d) Heatmaps of GCN, SGC, and APPNP for Poison Unlabeled ($PU$) setting on Cora-MLb with $p_{adv} = 0.1$.

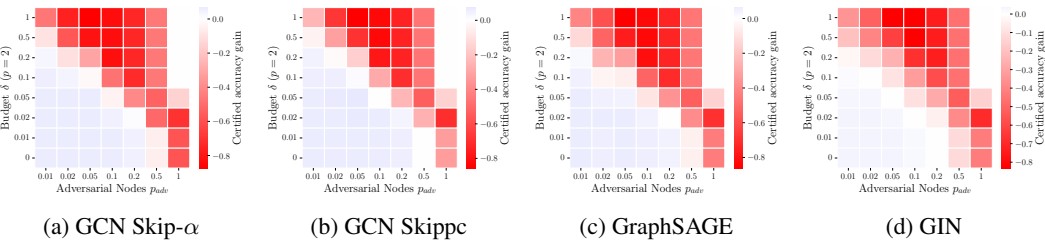

| (a) GCN Skip-$\alpha$ | (b) GCN Skippc | (c) GraphSAGE | (d) GIN |

Figure 18: Heatmaps of GCN Skip-$\alpha$, GCN Skippc, GraphSAGE, and GIN for Poison Unlabeled ($PU$) setting on Cora-MLb with $p_{adv} = 0.1$.

## J.2    APPNP

Fig. 19 shows that the inflection point observed in Fig. 4c is not observed in the other settings.

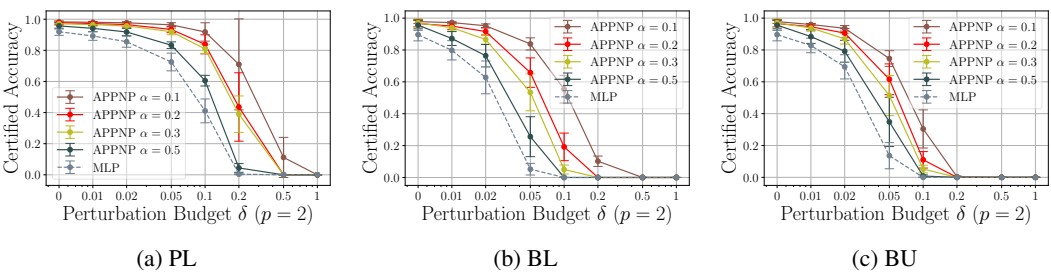

(a) PL                                    (b) BL                                    (c) BU

Figure 19: Cora-MLb, all settings with $p_{adv} = 0.05$.

## J.3    SYMMETRIC VS. ROW NORMALIZATION OF THE ADJACENCY MATRIX

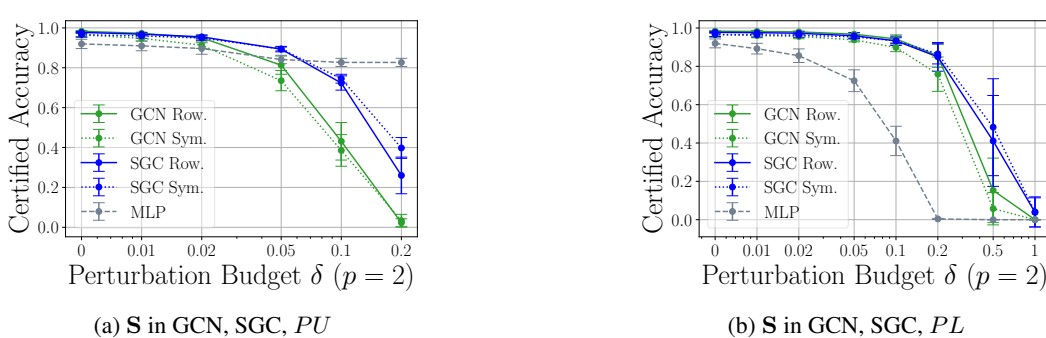

(a) **S** in GCN, SGC, $PU$                          (b) **S** in GCN, SGC, $PL$

Figure 20: Influence of symmetric and row normalized adjacency in GCN and SGC for poison unlabeled and poison labeled settings.

## J.4    RESULTS ON $p = 1$ ADVERSARY

Fig. 21 shows the certifiable robustness to $p = 1$ adversary on Cora-MLb dataset. The observation is consistent to the CSBM case.

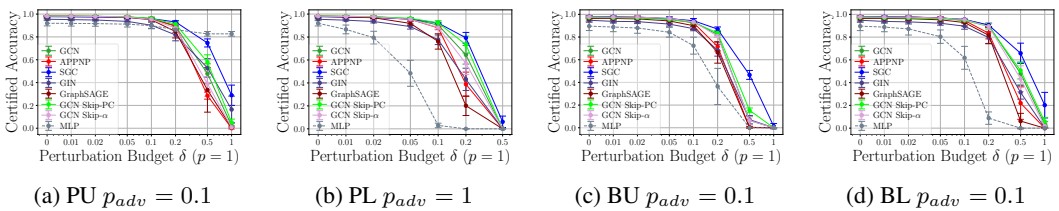

(a) PU $p_{adv} = 0.1$       (b) PL $p_{adv} = 1$       (c) BU $p_{adv} = 0.1$       (d) BL $p_{adv} = 0.1$

Figure 21: Cora-MLb results for PL, PU, BL and BU under $p = 1$ perturbation.

# K    ADDITIONAL RESULTS: CORA-ML

For Cora-ML we choose 100 test nodes at random and investigate in Fig. 22a the poison labeled ($PL$) setting with a strong adversary $p_{adv} = 1.0$ for GCN, SGC and MLP. It shows that QPCert can provide non-trivial robustness guarantees even in multiclass settings. Fig. 22b shows the results for poison unlabeled ($PU$) and $p_{adv} = 0.05$. Only SGC shows better worst-case robustness than MLP. This, together with both plots showing that the certified radii are lower compared to the binary-case, highlights that white-box certification of (G)NNs for the multiclass case is a more challenging task.

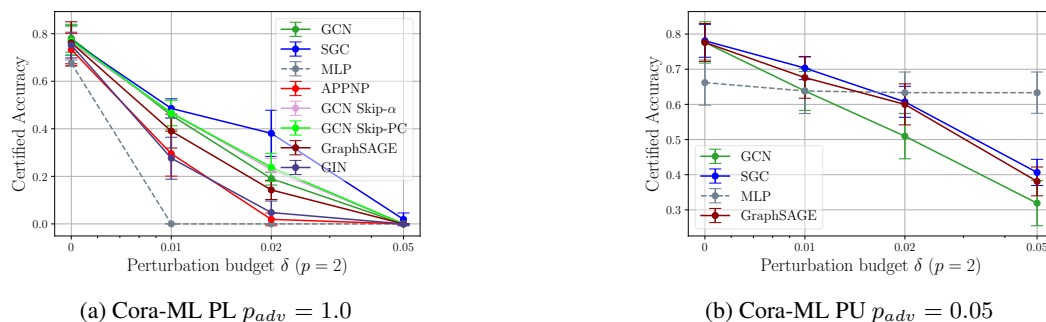

(a) Cora-ML PL $p_{adv} = 1.0$        (b) Cora-ML PU $p_{adv} = 0.05$

Figure 22: Cora-ML results for PL and PU.

## L  ADDITIONAL RESULTS: WIKICSB

Fig. 23a shows for the poisoned unlabeled setting that until a certain perturbation budget, GNNs lead to higher certified accuracy as an MLP. However, as $p_{adv} = 0.02$ it also shows that the certified accuracy of GNNs can be highly susceptible even to few perturbed nodes. Figs. 24b to 24d and 25 show a more detailed analysis into the certified accuracy difference of different GNN architectures for PU setting for $p_{cert} = 0.02$. We want to note the especially good performance of choosing linear activations (SGC). Fig. 23b shows that all GNNs achieve better certified accuracy as an MLP. Lastly, Fig. 24a shows the certified accuracy on WikiCSb for the $BL$ settings for $p_{cert} = 0.1$.

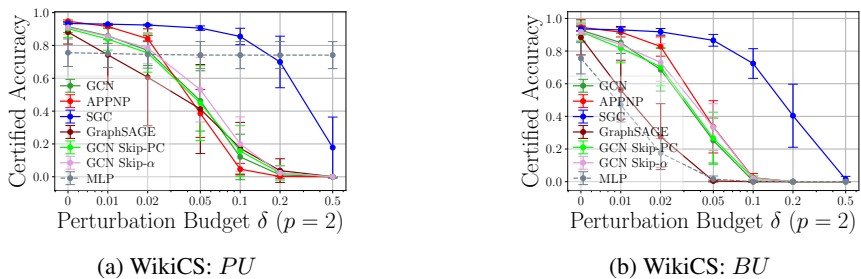

(a) WikiCS: $PU$        (b) WikiCS: $BU$

Figure 23: Certifiable robustness for different (G)NNs in Poisoning Unlabeled ($PU$) and Backdoor Unlabeled ($BU$) setting with $p_{adv} = 0.02$ for WikiCSb.

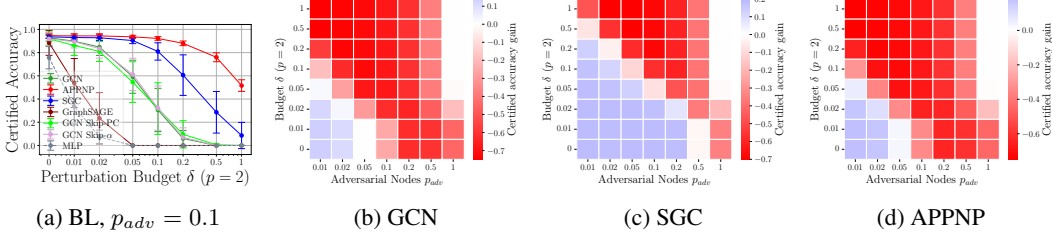

(a) BL, $p_{adv} = 0.1$     (b) GCN     (c) SGC     (d) APPNP

Figure 24: (a) Backdoor Labeled ($BL$) Setting. (b)-(d) Heatmaps of GCN, SGC, and APPNP for Poison Unlabeled ($PU$) setting on WikiCSb with $p_{adv} = 0.02$.

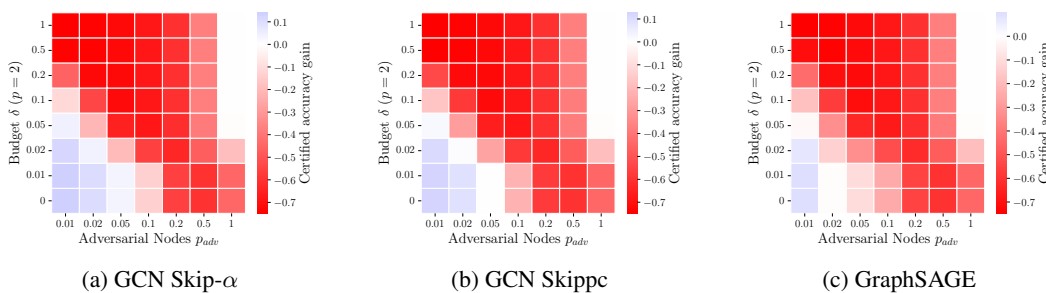

(a) GCN Skip-$\alpha$        (b) GCN Skippc        (c) GraphSAGE

Figure 25: Heatmaps of GCN Skip-$\alpha$, GCN Skippc, GraphSAGE, and GIN for Poison Unlabeled ($PU$) setting on WikiCSb with $p_{adv} = 0.02$.

## M    RELATED WORKS

**Poisoning certificates.** The literature on poisoning certificates is significantly less developed than certifying against test-time (evasion) attacks (Li et al., 2023) and we provide an overview in Table 1. Black-box certificates for poisoning are derived following three different approaches: $(i)$ Randomized smoothing, a popular probabilistic test-time certificate strategy (Cohen et al., 2019), in which randomization performed over the training dataset (Rosenfeld et al., 2020; Weber et al., 2023; Zhang et al., 2022). Other than data partitioning, a common defense is to sanitize the data, and Hong et al. (2024) certifies diffusion-based data sanitation via randomized smoothing. $(ii)$ Ensembles: Creating separate partitions of the training data, training individual base classifiers on top of them and certifying a constructed ensemble classifier (Levine & Feizi, 2021; Jia et al., 2021; Wang et al., 2022; Rezaei et al., 2023); Jia et al. (2021) and Chen et al. (2022) offer certificates and collective certificates, respectively, for bagging, while Levine & Feizi (2021) and Wang et al. (2022) derive certificates for aggregation-based methods tailored for black-box classifiers. $(iii)$ Differential Privacy[2] (DP): Ma et al. (2019) show that any $(\epsilon, \delta)$-DP learner enjoys a certain provable poisoning robustness. Liu et al. (2023) extend this result to more general notions of DP. Xie et al. (2023) derives guarantees against arbitrary data poisoning in DP federated learning setup. However, white-box deterministic poisoning certificates remain sparse. Drews et al. (2020) and Meyer et al. (2021) derive poisoning certificates for decision trees using abstract interpretations, while Jia et al. (2022) provides a poisoning certificate for nearest neighbor algorithms based on their inherent majority voting principle. Recently, Bian et al. (2024) derives a poisoning certificate for naive Bayes classification.

**Poisoning attacks and defense using the bilevel problem.** Biggio et al. (2012) and Xiao et al. (2015) use the bilevel problem with SVM hinge loss and regularized ERM to generate poison samples, and solve it using iterative gradient ascent. Mei & Zhu (2015) also recognize the possibility to transform the bilevel problem into a single-level one. However, they only reformulate the problem into a single-level bilinear one and solve it heuristically w.r.t. to the training data to generate poisoning attacks. Koh & Liang (2017) also considers the bilevel problem to detect and also generate poisoned samples using influence functions (gradient and Hessian vector product).

**Graphs.** Currently, there are no poisoning certificates for clean-label attacks specifically developed for GNNs or the task of node classification. (Lai et al., 2024) is the only work on poisoning certification of GNNs, but differ incomparably in their threat model and are black-box as well as not applicable to backdoors. However, there are many works on certifying against test-time attacks on graphs and Günnemann (2022) provides an overview.

---

[2] The mechanism to derive a poisoning certificate from a certain privacy guarantee is model agnostic, thus we count it as black-box. However, the calculated privacy guarantees may depend on white-box knowledge.

# N FURTHER DISCUSSIONS

## N.1 APPLICABILITY TO COMMONLY STUDIED PERTURBATION MODELS AND ATTACKS

QPCert applies to any poisoning or backdoor attack that performs $\ell_p$-bounded feature perturbations. As such, QPCert is directly applicable to clean-label (graph) backdoor attacks as proposed by Turner et al. (2019) and Xing et al. (2024), and clean-label poisoning attacks such as Huang et al. (2020) and Geiping et al. (2021). It is not directly applicable to poisoning of the graph structure as performed by MetaAttack (Zügner & Günnemann, 2019a). However, the MetaAttack strategy can be easily adapted to poison node features as done in App. I.3 and we discuss the challenges to extend QPCert to structure perturbations in App. N.2. The backdoor attack proposed by Dai et al. (2023) changes the node features and training labels jointly and thus, our method is only applicable if the training labels will be kept constant or the poisoned nodes are sampled only from the class, the training label should be changed to. Similar, Xi et al. (2021) develops a backdoor attack that changes the features and graph structure jointly and thus, QPCert is not applicable given the graph structure changes.

## N.2 CERTIFYING AGAINST GRAPH STRUCTURE PERTURBATIONS

In the following, we discuss how to approach certifying against poisoning of the graph structure and the open challenges that arise in the process. To certify against poisoning the graph structure, again Eq. (3) has to be solved but now, the adversary $\mathcal{A}$ can change the graph structure instead of the node features, meaning the optimization in Eq. (3) is performed w.r.t. the graph structure matrix $\mathbf{S}$ and thus, reads

$$\min_{\widetilde{\mathbf{S}},\theta} \mathcal{L}_{att}(\theta, \widetilde{\mathcal{G}}) \quad \text{s. t.} \quad \widetilde{\mathbf{S}} \in \mathcal{A}(\mathbf{S}) \ \wedge \ \theta \in \arg\min_{\theta'} \mathcal{L}(\theta', \widetilde{\mathcal{G}}) \tag{37}$$

with $\widetilde{\mathbf{S}} \in \mathcal{A}(\mathbf{S})$ representing the perturbed graph structure matrices constructable by the adversary and $\widetilde{\mathcal{G}} = (\widetilde{\mathbf{S}}, \mathbf{X})$. Indeed, it will be possible to reformulate this problem into a single-level problem similar to the description in Sec. 3. While in theory, QPCert Theorem 1 also applies to structure perturbations, the bounding strategy from Sec. 3.1 does result in loose bounds for structure perturbations, as an untrusted node will always result in a lower bound in the respective adjacency matrix entry of $0$ and an upper bound of $1$ - thus, spanning the whole space of possible entries.

To overcome this, one can approach certifying graph structure perturbations by including the NTK computation into the optimization problem with the drawback that each type of GNN architecture will require slight adaptations of the optimization problem depending on its corresponding NTK. Assuming that the chosen model is an $L = 1$ layer GCN (the formulation can be easily extended to arbitrary layers, see App. B.3), the bilevel optimization problem reads as follows

$$\min_{\boldsymbol{\alpha},\widetilde{\mathbf{S}},\mathbf{Q},\boldsymbol{\Sigma}_1,\boldsymbol{\Sigma}_2,\mathbf{E}_1,\dot{\mathbf{E}}_1,\dot{\mathbf{E}}_2} \mathrm{sgn}(\hat{p}_t) \sum_{i=1}^{m} y_i \alpha_i Q_{ti} \quad s.t. \quad \widetilde{\mathbf{S}} \in \mathcal{A}(\mathbf{S}) \ \wedge \ \boldsymbol{\alpha} \in \mathcal{S}(\mathbf{Q}) \tag{38}$$

$$\mathbf{Q} = \widetilde{\mathbf{S}}(\boldsymbol{\Sigma}_1 \odot \dot{\mathbf{E}}_1)\widetilde{\mathbf{S}}^T + \boldsymbol{\Sigma}_2 \odot \dot{\mathbf{E}}_2 \tag{39}$$

$$\boldsymbol{\Sigma}_1 = \widetilde{\mathbf{S}}\mathbf{X}\mathbf{X}\widetilde{\mathbf{S}}^T \tag{40}$$

$$\boldsymbol{\Sigma}_2 = \widetilde{\mathbf{S}}\mathbf{E}_1\widetilde{\mathbf{S}}^T \tag{41}$$

$$\mathbf{E}_1 = c_\sigma \mathop{\mathbb{E}}_{\mathbf{F}\sim\mathcal{N}(\mathbf{0},\boldsymbol{\Sigma}_l)} \left[\sigma(\mathbf{F})\sigma(\mathbf{F})^T\right] \tag{42}$$

$$\dot{\mathbf{E}}_1 = c_\sigma \mathop{\mathbb{E}}_{\mathbf{F}\sim\mathcal{N}(\mathbf{0},\boldsymbol{\Sigma}_l)} \left[\dot{\sigma}(\mathbf{F})\dot{\sigma}(\mathbf{F})^T\right] \tag{43}$$

$$\dot{\mathbf{E}}_2 = \mathbf{1}_{n \times n} \tag{44}$$

This (non-linear) bilevel problem can be transformed into a single-level problem as described in Sec. 3, as the inner-level problem $\boldsymbol{\alpha} \in \mathcal{S}(\mathbf{Q})$ is the same as in Eq. (4) and the same strategy can be applied to linearly model the resulting constraints from the KKT conditions. However, a crucial

difference to Eq. (4) are the additional non-linear constraints arising from optimizing over the NTK computation. Eqs. (39) to (41) are multilinear constraints that can be reduced to bilinear constraints by introducing additional variables as follows (for brevity, again writing the problem in its bilevel form):

$$\min \ \mathrm{sgn}(\hat{p}_t) \sum_{i=1}^{m} y_i \alpha_i Q_{ti} \quad s.t. \quad \widetilde{\boldsymbol{S}} \in \mathcal{A}(\boldsymbol{S}) \ \wedge \ \boldsymbol{\alpha} \in \mathcal{S}(\boldsymbol{Q}) \tag{45}$$

$$\boldsymbol{Q} = \boldsymbol{H}_1'' + \boldsymbol{H}_2 \tag{46}$$

$$\boldsymbol{H}_1 = \boldsymbol{\Sigma}_1 \odot \dot{\boldsymbol{E}}_1 \tag{47}$$

$$\boldsymbol{H}_1' = \boldsymbol{H}_1 \boldsymbol{S}^T \tag{48}$$

$$\boldsymbol{H}_1'' = \boldsymbol{S} \boldsymbol{H}_1' \tag{49}$$

$$\boldsymbol{H}_2 = \boldsymbol{\Sigma}_2 \odot \dot{\boldsymbol{E}}_2 \tag{50}$$

$$\boldsymbol{\Sigma}_1 = \boldsymbol{M}_1 \boldsymbol{M}_1^T \tag{51}$$

$$\boldsymbol{M}_1 = \boldsymbol{S} \boldsymbol{X} \tag{52}$$

$$\boldsymbol{M}_2 = \boldsymbol{E}_1 \boldsymbol{S}^T \tag{53}$$

$$\boldsymbol{\Sigma}_2 = \boldsymbol{S} \boldsymbol{M}_2 \tag{54}$$

$$\mathbf{E}_1 = c_\sigma \mathop{\mathbb{E}}_{\mathbf{F} \sim \mathcal{N}(\mathbf{0}, \boldsymbol{\Sigma}_1)} \left[ \sigma(\mathbf{F}) \sigma(\mathbf{F})^T \right] \tag{55}$$

$$\dot{\mathbf{E}}_1 = c_\sigma \mathop{\mathbb{E}}_{\mathbf{F} \sim \mathcal{N}(\mathbf{0}, \boldsymbol{\Sigma}_1)} \left[ \dot{\sigma}(\mathbf{F}) \dot{\sigma}(\mathbf{F})^T \right] \tag{56}$$

$$\dot{\mathbf{E}}_2 = \mathbf{1}_{n \times n} \tag{57}$$

where the optimization is over the same variables as in the previous problem, and additionally the variables $\boldsymbol{H}_1$, $\boldsymbol{H}_1'$, $\boldsymbol{H}_1''$, $\boldsymbol{H}_2$, $\boldsymbol{M}_1$, and $\boldsymbol{M}_2$. Eqs. (46) and (52) are linear constraints, the rest of the newly introduced constraints represent bilinear terms. The same bilinearization strategy can be applied given the NTK computation over arbitrary layers. The remaining non-linear and non-bilinear terms are Eqs. (55) and (56). They can be solved in closed-form resulting in relatively well-behaved functions as shown in App. D.1. Thus, a convex relaxation of the expectation terms can be derived by e.g., choosing linear functions that lower and upper bound the expectation terms.

Assume for now that one can linearly model $\widetilde{\boldsymbol{S}} \in \mathcal{A}(\boldsymbol{S})$, this can be achieved by e.g., choosing the adjacency matrix without normalization as graph structure matrix as done by Hojny et al. (2024). Then, the crucial question is:

*How to effectively solve the arising bilinear problem?*

In particular, the bilinearities arise in both, the constraints and objective, and thereby this contrasts e.g., with Zügner & Günnemann (2020) who only have to deal with a bilinear objective but have linear constraints. The problem can be slightly simplified if $\boldsymbol{S}$ is chosen to be the unnormalized adjacency matrix, as then any $\widetilde{\boldsymbol{S}}$ is discrete and thus Eqs. (48), (49) and (53) represent multiplications of a continuous with a discrete variable and thus, can be linearly modeled using standard modeling techniques. However, the objective and Eqs. (47), (50) and (51) remain products of continuous variables and thus, can fundamentally not be modeled linearly. One potential way to tackle this, is to use techniques of convex relaxations of bilinear functions as e.g., the so called McCormic envelope (McCormick, 1976). However, it is not clear if common bilinear relaxation techniques can scale to problems of the size necessary to compute practical certificates for machine learning datasets, nor is it clear if the relaxations introduced in the process result in tight enough formulations to yield non-trivial certificates. This is complicated by the fact that problems that are studied in the bilinear optimization literature are often significantly smaller than the problem size we can expect from certifying graph structure perturbations. However, it is not unlikely that further progress in bilinear optimization will make this problem tractable.

We want to note that linearly modeling $\widetilde{\boldsymbol{S}} \in \mathcal{A}(\boldsymbol{S})$ by choosing an unnormalized adjacency matrix as the graph structure matrix results in a restriction of possible architecture to certify. It could be possible

to adapt Zügner & Günnemann (2020)'s modeling technique for the symmetric-degree normalized adjacency matrix they use to certify a finite-width GCN to the above optimization problem without increasing its difficulty as it is already bilinear.

## N.3 PRACTICAL IMPLICATION OF FEATURE AND STRUCTURE PERTURBATIONS

Both feature and structure perturbations can find applications in real-world scenarios. In particular, important application areas for graph learning methods with adversarial actors are fake news detection (Hu et al., 2024) and spam detection (Li et al., 2019). Regarding fake news detection, feature perturbations can model changes to the fake news content or to (controlled) user account comments and profiles to mislead detectors (Hu et al., 2024; Le et al., 2020). Structure perturbations allow to model a change in the propagation patterns (e.g., through changing a retweet graph) (Wang et al., 2023). The qualitative difference in the application of feature compared to structure perturbations is similar for spam detection. Here, feature perturbation can model spammers trying to adapt their comments to avoid detection (Li et al., 2019; Wang et al., 2012). However, structure perturbations can model behavioral changes in the posting patterns of spammers to imitate real users (Soliman & Girdzijauskas, 2017; Wang et al., 2012).

## N.4 QPCERT FOR OTHER GNNS

While our analysis focused on commonly used GNNs with and without skip connections, QPCert is broadly applicable to any GNNs with a well-defined analytical form of the NTK. The following challenges and considerations have to be taken into account when extending QPCert to other architectures:

1. **NTK-network equivalence:** The equivalence between the network and NTK breaks down when the network has non-linear last layer or bottleneck layers (Liu et al., 2020). Consequently, our certificates do not hold for such networks.

2. **Analytical form of NTK:** Deriving a closed-form expression for NTK is needed to derive bounds on the kernel. This might be challenging for networks with batch-normalizations or advanced pooling layers.

3. **Tight bounds for the NTK:** Ensuring non-trivial certificates requires deriving tight bounds for the NTK. Depending on the NTK, additional adaptation of our bounding strategy may be necessary.

Despite these considerations, most message-passing networks satisfy these criteria, making QPCert readily applicable to a wide range of architectures.

## N.5 ADAPTATION TO QPCERT FOR GRAPH CLASSIFICATION

Our work can be extended to graph classification using the graph NTK of the corresponding neural network trained for the task. Note that in this case, the kernel is computed between all pairwise graphs instead of nodes. Du et al. (2019) derived one such NTK for graph classification. Using this NTK and our MILP, robustness certificate can be derived. We elaborate on the technical details below.

**Adversary.** First, we extend our adversary setting to include multiple graphs and allow for node feature perturbation. We have $n$ graphs $\mathcal{G}_i = (\mathbf{S}_i, \mathbf{X}_i), \mathbf{S}_i \in \mathbb{R}^{n_i \times n_i}, \mathbf{X}_i \in \mathbb{R}^{n_i \times d}$ where $n_i$ is the number of nodes in graph $i$ with $d$ dimensional features, for all $i \in [n]$ and the adversary $\mathcal{A}$ can perturb the features $\mathbf{X}$ where $\in \mathcal{B}_p(\mathbf{x})$. As we consider semi-verified learning setting, let $\mathcal{U}_i$ be the set of nodes in $\mathcal{G}_i$ that are potentially controlled by the adversary $\mathcal{A}$.

**MILP.** The certification framework applies directly without any change given the NTK $\mathbf{Q}$ and its element-wise bounds. Therefore, the important adaptation here is to derive bounds on the NTK. Since, in this setting, the NTK computation involves all pairwise feature matrix covariance, that is, $\mathbf{X}_i \mathbf{X}_j^T$ for all $i, j$ pairs. Similar to node classification setting, we consider $\widetilde{\mathbf{X}}_i \in \mathcal{A}(\mathbf{X}_i)$ and derive $\widetilde{\mathbf{X}}_i \widetilde{\mathbf{X}}_j^T = \mathbf{X}_i \mathbf{X}_j^T + \Delta_{ij}$. We derive the bounds for $\Delta_{ij}$ for the perturbation $\mathcal{B}_p(\mathbf{x})$ with $p = 2$. It is easy to extend to $p = 1$ and $p = \infty$ in a similar way.

**Bounds for $\Delta_{ij}$ and $p = 2$.** We consider the perturbed feature matrix $\widetilde{\mathbf{X}}_i \in \mathcal{A}(\mathbf{X}_i)$ for all graphs $i$, and $\widetilde{\mathbf{X}}_i = \mathbf{X}_i + \mathbf{\Gamma}_i \in \mathbb{R}^{n_i \times d}$ where $\mathbf{\Gamma}_{ij}$ is the adversarial perturbations added to node $j$ of graph $i$ by the adversary, therefore, $\|\mathbf{\Gamma}_{ij}\|_p \leq \delta$ and $\mathbf{\Gamma}_{ij} > 0$ for $j \in \mathcal{U}_i$ and $\mathbf{\Gamma}_{ij} = 0$ for $j \notin \mathcal{U}_i$. Then

$$\widetilde{\mathbf{X}}_i \widetilde{\mathbf{X}}_j^T = (\mathbf{X}_i + \mathbf{\Gamma}_i)(\mathbf{X}_j + \mathbf{\Gamma}_j)^T$$
$$= \mathbf{X}_i \mathbf{X}_j^T + \mathbf{\Gamma}_i \mathbf{X}_j^T + \mathbf{X}_i \mathbf{\Gamma}_j^T + \mathbf{\Gamma}_i \mathbf{\Gamma}_j^T = \mathbf{X}_i \mathbf{X}_j^T + \Delta_{ij}. \tag{58}$$

As a result, it suffices to derive the element-wise worst-case bounds for $\Delta_{ij} \in \mathbb{R}^{n_i \times n_j}$, $(\Delta_{ij})_{ab}^L \leq (\Delta_{ij})_{ab} \leq (\Delta_{ij})_{ab}^U$, for different perturbations. To do so, our strategy is to bound the scalar products $\langle (\mathbf{\Gamma}_i)_a, (\mathbf{X}_j)_b \rangle$ and $\langle (\mathbf{\Gamma}_i)_a, (\mathbf{\Gamma}_j)_b \rangle$ element-wise.

$$|\langle (\mathbf{\Gamma}_i)_a, (\mathbf{X}_j)_b \rangle| \leq \|(\mathbf{\Gamma}_i)_a\|_2 \|(\mathbf{X}_j)_b\|_2 \leq \delta \|(\mathbf{X}_j)_b\|_2$$
$$|\langle (\mathbf{\Gamma}_i)_a, (\mathbf{\Gamma}_j)_b \rangle| \leq \|(\mathbf{\Gamma}_i)_a\|_2 (\|\mathbf{\Gamma}_j)_b\|_2 \leq \delta^2. \tag{59}$$

Using Eq. (59), we derive the lower and upper bounds of $(\Delta_{ij})_{ab}$:

$$(\Delta_{ij})_{ab}^L = \begin{cases} 0+ & \text{if } a \notin \mathcal{U}_\rangle, b \notin \mathcal{U}_j \\ -\delta\|(\mathbf{X}_j)_b\|_2+ & \text{if } a \in \mathcal{U}_i \\ -\delta\|(\mathbf{X}_i)_a\|_2+ & \text{if } b \in \mathcal{U}_j \\ -\delta^2 & \text{if } a \in \mathcal{U}_i, b \in \mathcal{U}_j \end{cases} \qquad (\Delta_{ij})_{ab}^L = \begin{cases} 0+ & \text{if } a \notin \mathcal{U}_\rangle, b \notin \mathcal{U}_j \\ \delta\|(\mathbf{X}_j)_b\|_2+ & \text{if } a \in \mathcal{U}_i \\ \delta\|(\mathbf{X}_i)_a\|_2+ & \text{if } b \in \mathcal{U}_j \\ \delta^2 & \text{if } a \in \mathcal{U}_i, b \in \mathcal{U}_j \end{cases}$$

Thus, writing the bounds succinctly using the indicator function gives us,

$$(\Delta_{ij})_{ab}^L = -\delta\|(\mathbf{X}_j)_b\|_2 \mathbb{1}[a \in \mathcal{U}_i] - \delta\|(\mathbf{X}_i)_a\|_2 \mathbb{1}[b \in \mathcal{U}_j] - \delta^2 \mathbb{1}[a \in \mathcal{U}_i \wedge b \in \mathcal{U}_j \wedge a \neq b],$$

$$(\Delta_{ij})_{ab}^U = \delta\|(\mathbf{X}_j)_b\|_2 \mathbb{1}[a \in \mathcal{U}_i] + \delta\|(\mathbf{X}_i)_a\|_2 \mathbb{1}[b \in \mathcal{U}_j] + \delta^2 \mathbb{1}[a \in \mathcal{U}_i \wedge b \in \mathcal{U}_j \wedge a \neq b].$$

Using these bounds, similar to node classification, we can propagate them through the NTK computation to get the bounds on NTK. Using the NTK bounds, we can apply QPCert to get the certificate for graph classification. $\qquad\square$

