# OpenReview forum: "Provable Robustness of (Graph) Neural Networks Against Data Poisoning and Backdoors"
_ICLR.cc/2025/Conference — Submitted to ICLR 2025_

### Official Review · Reviewer_dGK8 · 2024-11-02

**Soundness:** 3
**Presentation:** 3
**Contribution:** 4
**Rating:** 6
**Confidence:** 2

**Summary:**

The authors leverage the Neural Tangent Kernel (NTK) to capture the complex training dynamics of GNNs and reformulate the bilevel optimization problem describing poisoning as a mixed-integer linear program. This allows them to provide white-box certificates for GNNs and neural networks in general, which was previously an unsolved problem.

**Strengths:**

- Provide a comprehensive theoretical analysis to support the methodology
- Very inspiring to leverage NTK to defend the adversarial and backdoor attacks
- Reformulated the bilevel optimization problem describing poisoning as a mixed-integer linear program

**Weaknesses:**

- this paper only considers the scenario where the adversaries manipulate the node features, but there are plenty of graph attack works that manipulate the graph structure and the node features at the same time. Could the method in this paper be leveraged for poisoned data with both graph structures and node features manipulated?
- The experiments only compare the GNNs with MLP. Could authors provide some adversarial/backdoor defense baselines? While the comparison with MLPs demonstrates the positive impact of QPCert, it does not adequately illustrate the extent of QPCert's effectiveness.
- In Figure 2, MLP consistently has the lowest accuracy when evaluated in the PL task, but MLP can have a better performance than some GNNs in Figure 3 (PU and BU tasks) when perturbation budges are large. Could the author explain what factors lead to these differences?

**Questions:**

the same as weaknesses

---

> ### Author Response · Authors · 2024-11-22
>
> Thank you for the positive review! We very much value the appreciative feedback and updated the draft to reflect your questions. Below, we answer to your weaknesses and questions in detail.
>
>
> ----
>
>
>
> > Could the method in this paper be leveraged for poisoned data with both graph structures and node features manipulated?
>
> We agree that tackling graph structure perturbations (or feature and structure perturbations jointly) is an important problem as we mention in our conclusion in Lines 501/502. It is thinkable to extend QPCert to also tackle structure certification, but this leads to several yet unsolved technical challenges. As a result, we have now included a detailed discussion on extending QPCert to structure perturbations in Appendix N.2, which we link to in the conclusion (Lines 502/503). In particular, the general strategy of QPCert to reformulate the bilevel optimization problem into a single-level problem is still applicable. However, an immediate application of the bounding scheme (Section 3.1) to structure perturbations would result into loose bounds that requires to include the NTK optimization into the optimization problem. Thus, we show in Appendix N.2 how this can lead to a mixed-integer bilinear problem describing structure perturbations and describe the open and yet unsolved technical challenges to make this problem tractable.
>
>
> ----
>
>
> >The experiments only compare the GNNs with MLP. Could authors provide some adversarial/backdoor defense baselines?
>
> In our analysis, we focus on commonly used message-passing GNNs with and without skip-connections, as they are widely used in practice and cover a representative spectrum of design choices. Thus, these networks allow us to demonstrate the effectiveness of QPCert with respect to the most commonly used GNNs for node classification. We agree that applying QPCert to adversarial/backdoor defense baselines is an interesting question. However, in order to include defense baselines, it is first necessary to derive their NTK in analytical form. This is a non-trivial task and while we provide such derivations for common message-passing GNNs, we are not aware of NTK derivation techniques for poisoning defenses so far (either for GNNs or general NNs). The moment the analytical NTK for a poisoning defense is available, it can be immediately integrated into our framework and we now also include a detailed discussion on extending QPCert to other architectures in Appendix N.4.
>
> ----
>
> > In Figure 2, MLP consistently has the lowest accuracy when evaluated in the PL task, but MLP can have a better performance than some GNNs in Figure 3 (PU and BU tasks) when perturbation budges are large. Could the author explain what factors lead to these differences?
>
> The reason is that in the poison unlabeled (PU) or backdoor unlabeled (BU) settings only the unlabeled nodes are attacked and as an MLP ignores unlabeled nodes during training, it is not affected by such a poisoning attack. We discuss this in Lines 406/407 in the draft and updated the discussion to make this more clear. Additionally, the reason why the MLP still loses some accuracy in the PU task is that due to the transductive setting, the attacked unlabeled nodes in the training graph correspond to the test nodes. Thus, the MLP is confronted with the attacked nodes at test time resulting in an evasion attack for the MLP. However, as the attacked nodes are only seen at test time for the MLP, they cannot affect the clean node predictions, and thus, the MLP accuracy plateaus for strong budgets (see e.g., Figure 3a $\delta=0.2$ and $\delta=0.5$). This is slightly different for the BU task. Due to the backdoor setting, every test node can be attacked, and thus, with a certain attack strength $\delta$, the MLP can't provide a guarantee even though its training was not affected. To provide a more clear experiment description, we now added this additional clarification into the draft in lines 411-413.

---

> ### Comment · Reviewer_dGK8 · 2024-11-22
> **Response to the authors**
>
> Thanks for the authors' detailed explanation. My questions for Q1 and Q3 are almost solved. However, for Q2, I believe the authors may have misunderstood my questions. I do not expect to apply QPCert to other defense baselines and also don't expect to see the analytical NTK for a poisoning defense. Actually, I want to compare the effectiveness (certified accuracy) of QPCert with other graph defense baselines. Therefore, could the authors test some existing graph defense methods, such as [1][2][3], on the datasets this paper uses?
>
> [1] GNNGuard: Defending Graph Neural Networks against Adversarial Attacks.
>
> [2] Elastic Graph Neural Networks.
>
> [3] Graph Information Bottleneck.

---

> > ### Author Response · Authors · 2024-11-25
> >
> > Thank you for your response! Indeed, it seems we misunderstood your Q2. We uploaded a new paper draft and now include **new experiments** resulting in a detailed comparison of the certified accuracy provided by QPCert with GNNGuard [1] and Elastic Graph Neural Networks [2] in Appendix I.7 and link to it in our baselines discussion in Lines 374/375. Currently, we included such a comparison for the CSBM dataset and are actively working on expanding this analysis to the other datasets in the paper which, due to time reasons, we plan on including in the camera-ready version.
> >
> > Note that poisoning defenses such as GNNGuard and Elastic GNN do *not* provide certified accuracies. Thus, measuring their robust accuracy with an adversarial attack as we do now in Appendix I.7 provides an upper bound on their true robustness. On the other hand, QPCert provides a certificate and thus, a lower bound to the actual robustness. As a result, the provided accuracies are not directly comparable but give insights on how much the gap between the best-certified accuracy and robust accuracy (independent of the method) is on a dataset.
> >
> > Interestingly, we find on CSBM for small to intermediate perturbation budgets that QPCert can provide higher certified robustness for a GCN than the robust accuracy provided by GNNGuard and Elastic GNN. We think this is due to a robustness-accuracy tradeoff these methods face due to their preprocessing of the graph / adapting of the message passing scheme, resulting in a slightly lower clean accuracy than the GCN on the CSBM dataset.
> >
> > [1] Zhang et al. "GNNGuard: Defending Graph Neural Networks against Adversarial Attacks", NeurIPS 2020
> > [2] Liu et al. "Elastic Graph Neural Networks", ICML 2021

---

> > > ### Comment · Reviewer_dGK8 · 2024-11-25
> > > **Response to Authors**
> > >
> > > Thank you for authors' response. The experimental results are very impressive, and I would like to increase my score to 7. However, since a score of 7 is not an option here, I will maintain my score at 6. Nonetheless, please note that my perceived score has effectively increased.

---

> > > > ### Author Response · Authors · 2024-12-02
> > > > **Thank you for wanting to raise the score to 7!**
> > > >
> > > > We want to thank the reviewer for the positive response, wanting to raise the score to 7 and calling our new experimental results impressive!

---

### Official Review · Reviewer_b7b5 · 2024-11-03

**Soundness:** 3
**Presentation:** 3
**Contribution:** 3
**Rating:** 8
**Confidence:** 3

**Summary:**

I'm rating this paper as "8: accept, good paper".

Summary: the paper proposes a certification against white-box and backdoor attacks, by making use of Graph NTK and the fact that for sufficiently wide networks the network is throughly understood given the input data and the neural tangent kernel.

**Strengths:**

- Clear writing
- Elegant ideas to make the bi-level optimisation feasible and tracktable

**Weaknesses:**

-

**Questions:**

- In line 369 it is mentioned that: "... We note as we are the first work to study white-box certificates for clean-label attacks on node features in graphs in general, there is no baseline prior work. ...". Even if there are no baselines for white-box attacks, aren't the certificates for black-box baselines still applicable? Specially because these methods are different in, e.g., their relaxation which may differently affect their performance and one cannot presume superiority of certificate in black-box versus white-box setting?

---

> ### Author Response · Authors · 2024-11-22
>
> Thank you for your positive review and recommendation to accept our paper! Below, we address your questions in detail.
>
> -----
>
> >Even if there are no baselines for white-box attacks, aren't the certificates for black-box baselines still applicable? Specially because these methods are different in, e.g., their relaxation which may differently affect their performance and one cannot presume superiority of certificate in black-box versus white-box setting?
>
> We agree that one cannot presume superiority of a certificate in black-box vs. white-box setting. Unfortunately, all so far developed black-box poisoning certificates have only been developed for the image domain (or in general, i.i.d data) and cannot be directly applied to graph learning tasks. There is one exception [1] who extend the randomized smoothing-based poisoning certificate framework to graph learning, but they only consider the case of injecting completely new nodes into the training graph (as we mentioned in Related Work line 488/489). Extending their framework to be applicable to $\ell_p$-bounded perturbations is non-trivial and would require a new method derivation. We agree that our initial argumentation in the manuscript did not make this point clear and thus, we added these arguments in lines 372-374 in the experimental results section for more clarity.
>
> [1] Lai et al. "Node-aware bi-smoothing: Certified robustness against graph injection attacks", IEEE S&P 2024

---

> > ### Author Response · Authors · 2024-11-25
> >
> > Dear Reviewer,
> >
> > We want to thank you again for your positive review and the recommendation for acceptance! As the discussion period closes soon, we would greatly appreciate a short response if we could answer your questions. If you have any further questions on which you want additional clarifications, please let us know and we will be happy to provide further answers.
> >
> > Best regards,
> > The Authors

---

### Official Review · Reviewer_NS1A · 2024-11-08

**Soundness:** 3
**Presentation:** 3
**Contribution:** 2
**Rating:** 6
**Confidence:** 4

**Summary:**

The paper introduces a white-box certification framework, QPCert, for evaluating the robustness of neural networks, specifically Graph Neural Networks (GNNs), against data poisoning and backdoor attacks. This framework, grounded in Neural Tangent Kernel (NTK) theory, reformulates poisoning detection as a mixed-integer linear program (MILP), enabling robustness certificates against node feature perturbations in GNNs. Thexperimental results to validate QPCert's effectiveness.

**Strengths:**

1. This paper studies an interesting problem: the certified robustness against data poisoning and backdoor attacks.
2. The extensive theoretical analysis is provided.
3. The paper is well written.

**Weaknesses:**

1. While the theoretical analysis is promising, the empirical evaluation could be strengthened to more comprehensively demonstrate the effectiveness of the proposed certification method. Currently, only one attack, APGD [1], is considered in the experiments. Since certification methods aim to provide provable guarantees against all attacks within a specific threat model, it would be beneficial for the authors to clarify how their certification framework addresses the threat models of other commonly studied attacks, such as graph poisoning and backdoor attacks [2, 3, 4]. This would provide a more insightful discussion of the method’s robustness and scope of applicability, beyond specific attack evaluations
2. It appears that the Neural Tangent Kernel (NTK) approach is currently limited to specific GNN architectures such as GCN, SGC, and GraphSage, which could restrict the broader applicability of the proposed method. It would be helpful if the authors could discuss any challenges or limitations in extending their method to other GNN architectures or clarify why these specific GNNs were chosen as representative examples. Additionally, outlining potential directions for adapting the approach to other types of GNNs could strengthen the paper's discussion and broaden its scope.
3. The proposed method addresses node feature perturbation attacks, yet structural perturbations are often more prevalent and impactful in real-world scenarios. It would strengthen the paper if the authors could discuss the challenges in adapting their method to structural perturbations or clarify their focus on feature perturbations. Additionally, a comparison of the practical implications of feature versus structural perturbations in real-world applications would provide valuable insights into the method's scope, limitations, and potential future directions.


[1] Reliable Evaluation of Adversarial Robustness with an Ensemble of Diverse Parameter-free Attacks.

[2] Adversarial Attacks on Graph Neural Networks via Meta Learning.

[3] Graph Backdoor.

[4] Unnoticeable Backdoor Attacks on Graph Neural Networks.

**Questions:**

1. Please add more experiments against some realistic graph poisoning and backdoor attacks.
2. Please discuss applying the proposed method to more advanced GNNs.
3. There are some works that study clean-label (graph) backdoor attacks [1, 2], can the proposed method be applied to this attack?

[1] Clean-Label Backdoor Attacks.

[2] A clean-label graph backdoor attack method in node classification task.

---

> ### Author Response · Authors · 2024-11-22
> **Response to W1, Q1, Q3**
>
> We thank you for the constructive review! Based on it, we now strengthened our empirical evaluation with new experiments and updated the draft with several important discussions. Below, we address all your mentioned weaknesses and questions in detail.
>
> ## W1 & Q1 - Add more realistic graph poisoning and backdoor attacks
>
> We have now added **three new attacks**: another graph poisoning attack namely MetaAttack [1] adapted to feature perturbations, as well as two graph backdoor attacks. This results in a thorough evaluation of the tightness of QPCert. As backdoor attacks, we implemented the clean-label graph backdoor attack from [2] as well as an attack that is allowed to use the complete $\ell_p$-ball budget freely for the poisoned nodes and target node. We refer to Appendix I.3 for details and new results and updated the experimental details discussion in Lines 352-356.
>
> We want to clarify that our usage of the APGD attack in the manuscript corresponds to a strong graph poisoning strategy where we directly differentiate through the training process by solving the SVM's quadratic problem associated to an infinitely-wide GNN with QPLayer [3] - a differentiable quadratic programming solver. APGD comes into play as a more sophisticated updating scheme regarding creating the poisoned graph based on the above-calculated gradient, compared to doing simple projected gradient descent.
>
> [1] Zügner et al. "Adversarial Attacks on Graph Neural Networks via Meta Learning", ICLR 2019
> [2] Xing et al. "A Clean-Label Graph Backdoor Attack Method in Node Classificaiton Task", Knowledge-Based Systems 2024
> [3] Bambade et al. "QPLayer: efficient differentiation of convex quadratic optimization", 2023
>
> ## W1 & Q3 - Clarification on application to common threat models
>
> To avoid any misunderstandings, we want to clarify that our method directly addresses any poisoning or backdoor attacks that perform $\ell_p$ bounded feature perturbations. In particular, the (certified) accuracies reported in Figures 2 and 3 are provable lower bounds for the robust accuracy achieved against the strongest adversary possible (within the given $\ell_p$-norm bounds) and thus, are valid for all conceivable attack strategies within the given $\ell_p$ bounds. As a result, the reported certified accuracies and thus, the applicability/effectiveness of QPCert, is *independent* of any specific attack evaluations.
>
> Regarding addressing threat models of commonly studied attacks:
> * QPCert is directly applicable to the clean-label (graph) backdoor attacks proposed in [1,2]. Both attacks perform bounded feature perturbations to labeled training nodes. This is exactly the setting we call "backdoor labeled" in our experiments, and e.g., Figure 13 (a) shows the certified accuracies in this setting that are immediately valid for any of the investigated NNs if they would be attacked with the attacks in [1,2]. As mentioned above, we now also implement the backdoor attack from [2] (see Appendix I.3).
> * QPCert is applicable to graph poisoning with MetaAttack if the gradient is taken w.r.t. to the graph features instead of the graph structure. Indeed, as mentioned previously, we adapt MetaAttack [1] to perform graph feature poisoning and the results can be found in Appendix I.3.
> * The backdoor attack in [3] changes features and training labels jointly, thus our method is only applicable to [3] if the training labels will be kept constant. Similarly, [4] changes the features and graph structure jointly. So far, QPCert does not allow to certify against graph structure changes and, as detailed in the answer to W3, we now detail the challenges to certify graph structure changes in Appendix N.2.
>
> We have now included a discussion on which threat models of commonly studied attacks are addressed by QPCert in Appendix N.1 and point to this discussion in line 475 (Section 5).
>
> [1] Turner et al. "Clean-Label Backdoor Attacks", 2018
> [2] Xing et al. "A Clean-Label Graph Backdoor Attack Method in Node Classificaiton Task", Knowledge-Based Systems 2024
> [3] Dai et al. "Unnoticeable Backdoor Attacks on Graph Neural Networks", ACM Web Conference 2023
> [4] Xi et al. "Graph Backdoor", USENIX Security 2021

---

> ### Author Response · Authors · 2024-11-22
> **Response to W2, W3 (a), Q2**
>
> ## W2 & Q2 - QPCert for more advanced GNNs and clarification on the choice of the GNNs
>
> **Applicability to other GNNs**
> While our analysis focused on commonly used GNNs with and without skip connections, QPCert is broadly applicable to any GNNs with a well-defined analytical form of the NTK. When extending QPCert to other architectures the following challenges and considerations can arise:
>
> 1. **NTK-network equivalence:** The equivalence between the network and NTK breaks down when the network has a non-linear last layer or bottleneck layers [1]. Consequently, our certificates do not hold for such networks.
> 2. **Analytical form of NTK:** Deriving a closed-form expression for the NTK is needed to derive bounds on the kernel. This might be challenging for networks with batch-normalizations or advanced pooling layers.
> 3. **Tight bounds for the NTK:** Ensuring non-trivial certificates requires deriving tight bounds for the NTK. Depending on the NTK, additional adaptation of our bounding strategy may be necessary.
>
> Despite these considerations, most message-passing networks satisfy these criteria, making QPCert readily applicable to a wide range of architectures, which we demonstrated by incorporating eight different GNNs (including MLP). We have added a discussion outlining these challenges in Appendix N.4 and reference it in Line 461.
>
> **Why the chosen architectures?**
> We chose to focus on common message-passing networks, as they are widely used in practice and cover a representative spectrum of design choices including skip connections. Thus, these networks allow us to demonstrate the applicability and potential of QPCert with respect to the most commonly used GNNs for node classification.
>
> [1] Liu et al. On the linearity of large non-linear models: when and why the tangent kernel is constant. NeurIPS 2020.
>
> ## W3 (a) - Discussing challenges of adapting QPCert to structural perturbations
>
> We now include a detailed technical discussion on the challenges arising when adapting QPCert to certify structure perturbations in Appendix N.2 and reference it in the conclusion (Line 503/504). On a technical level, to certify structure perturbations, the general strategy of QPCert to reformulate the bilevel optimization problem into a single-level problem is still applicable. However, as we argue in Appendix N.2., to avoid loose bounds and thus, a non-vacuous certificate, the NTK computation has to be included into the optimization problem. Therefore, we now derive and provide the associated optimization problem in Appendix N.2 (Eqs. 45-57), together with a discussion on the unsolved challenges that would allow to make this problem tractable. Broadly, the challenges are:
>
> 1. The resulting optimization problem is still untight and fundamentally bilinear with both the objective and constraints having many bilinearities (multiplications of two continuous variables) that cannot be modeled linearly.
> 2. While potential bilinear relaxations are conceptually possible, making them tight enough and scaling them to problem sizes of practical interest for certification are open challenges.
>
> We agree that tackling graph structure perturbations is an important problem as we discuss in our conclusion (Lines 502-504). We focus on feature certification as, next to open technical challenges regarding structure certification discussed in App. N.2, it $(i)$ has different and complementary real-world applications that we detail in our answer to W3b below, and $(ii)$ poses unique graph-related challenges due to the interconnectedness of nodes, which QPCert can elegantly handle through the use of the graph NTK. The unique challenge of certifying feature perturbations in the face of interconnected data points is also the reason why there is a distinct line of work focused on certifying node feature perturbations [1,2,3]
>
> [1] Zügner et al. "Certifiable robustness and robust training for graph convolutional networks", KDD 2019
> [2] Scholten et al. "Randomized Message-Interception Smoothing: Gray-box Certificates for Graph Neural Networks", NeurIPS 2022
> [3] Scholten et al. "Hierarchical Randomized Smoothing", NeurIPS 2023

---

> > ### Comment · Reviewer_NS1A · 2024-11-24
> >
> > Thanks, I will raise my score to 6. Hope you will include the above discussion in your future version.

---

> > > ### Author Response · Authors · 2024-11-25
> > >
> > > Thank you for raising the score and recommending acceptance! We would like to mention that all the discussions in our response are already included in the revised paper and we plan to keep these discussions in the paper as we sincerely think they improve upon our original submission. The referenced sections and line numbers in our response point to the respective updates in the revised paper.

---

> ### Author Response · Authors · 2024-11-22
> **Response to W3 (b)**
>
> ## W3 (b) - Practical implications of feature versus structural perturbations
>
> Indeed both feature and structure perturbations find different real-world applications. As particular examples, two important application areas for graph-based learning methods are fake news detection [1] and spam detection [2]. Regarding fake news detection, feature perturbations can model changes to the fake news content or to (controlled) user account profiles to avoid detection [1,3,4]. Structure perturbations allow to model a change in the propagation behavior (e.g., retweeting) to again, avoid detection [5]. A similar qualitative differentiation holds true for spam detection. Here, feature perturbations can model spammers trying to adapt their comments so as to mislead detectors [2,6,7]. On the other hand, structure perturbations can model a behavioral change in the posting pattern of a spammer, so as to post comments to different product items or stores to imitate real users [6,7]. We tried to touch upon this in Lines 499/500 in the conclusion and now included an extended discussion in Appendix N.3 linked in Lines 499/500.
>
> [1] Hu et al. "An overview of fake news detection: From a new perspective", Fundamental Research 2024
> [2] Li et al. "Spam Review Detection with Graph Convolutional Networks", CIKM 2019
> [3] Le et al. "MALCOM: Generating Malicious Comments to Attack Neural Fake News Detection Models", ICDM 2020
> [4] He et al. "PETGEN: Personalized Text Generation Attack on Deep Sequence Embedding-based Classification Models", SIGKDD 2021
> [5] Wang et al. "Attacking fake news detectors via manipulating news social engagement", WWW 2023
> [6] Wang et al. "Identify Online Store Review Spammers via Social Review Graph", ACM TIST 2012
> [7] Soliman et al. "AdaGraph: Adaptive Graph-Based Algorithms for Spam Detection in Social Networks", 2017
>
>
> **EDIT:** (25.11.) Updated Line Numbers in Responses

---

### Official Review · Reviewer_PTK3 · 2024-11-12

**Soundness:** 3
**Presentation:** 4
**Contribution:** 3
**Rating:** 6
**Confidence:** 3

**Summary:**

The paper introduces a novel framework, QPCert, for certifying the robustness of GNNs, against data poisoning and backdoor attacks. This work leverages the NTK for white-box certification and reformulates the poisoning problem as a MILP to provide formal guarantees for GNNs. The authors explore this certification's effectiveness across various GNN architectures and benchmark datasets, and also analyze the effect of graph structure on robustness, presenting insights for architectural choices in GNNs with robustness considerations.

**Strengths:**

The paper is generally well-organized and thorough, with sections on methodological background, detailed steps in QPCert’s derivation, and clear experimental results. Extensive experiments are conducted across multiple GNN architectures and datasets. The analysis includes diverse attack scenarios and perturbation models, highlighting the framework’s adaptability and utility.

**Weaknesses:**

Although the focus is on feature perturbations, robustness against graph structure modifications is a critical issue for GNNs. Future work on structural robustness would greatly enhance the scope and impact of this framework.

On the assumption on large Width: The reliance on the NTK’s emight limit applicability to smaller, practical network sizes.

How your work can be extended to graph classification?

Given the MILP reformulation, computational costs may scale poorly with large datasets,

**Questions:**

No questions

---

> ### Author Response · Authors · 2024-11-22
>
> Thank you for the positive review! We appreciate the feedback and below, address the weaknesses in detail.
>
> ----
> > Future work on structural robustness would greatly enhance the scope and impact of this framework
>
> We agree that future work on structure perturbations is important as we mention in the conclusion in Lines 502-504. We have now included an *extensive technical discussion* in Appendix N.2 on how our framework could be extended to graph structure perturbations and what the respective open technical challenges are. Broadly, extending the framework to structure perturbations introduces the following open challenges:
> 1. The resulting optimization problem will be untight and fundamentally bilinear with both the objective and constraints having many bilinearities (multiplications of two continuous variables) that cannot be modeled linearly.
> 2. While potential bilinear relaxations are conceptually possible, making them tight enough and scaling them to problem sizes of practical interest for certification are open challenges.
>
> We hope that our work and the added discussion lay the foundations for future work tackling this important but yet unsolved problem.
>
> ----
>
> > How your work can be extended to graph classification?
>
> Our work can be readily extended to graph classification using the graph NTK of the corresponding neural network trained for such a task. Note that in this case, the kernel is computed between all pairwise graphs instead of nodes. [1] provides a derivation of a graph NTK that can be applied in this context. By utilizing this NTK in our MILP framework, robustness certificates for graph classification can be obtained. We elaborate on the technical details in Appendix N.5 (linked in the discussion Line 461) and provide a brief outline here:
>
> The primary adaptation required is deriving entry-wise bounds for the NTK. Specifically, the NTK computation for graph classification involves pairwise feature matrix covariances, $X_iX_j^T$ where $X_i$ is the feature matrix of graph $i$, we adapt our derivation for this case. Detailed derivation for $\ell_2$ feature perturbation is given in the revised version, leading to the following bounds.
>
> For graph pairs $i,j$, let the adversarial feature matrix covariance be $X_i'X_j'^T=X_i X_j^T + \Delta_{ij}$. Let $\mathcal{U}\_i$ be the set of all potentially corrupted nodes for graph $i$ and $(X_j)\_b$ the node feature vector of node $b$ associated to graph $j$. Then the element-wise bounds on $\Delta_{ij}$ are given by
>
> $(\Delta_{ij})^L_{ab} = -\delta \lVert (X_j)_b \rVert_2 {1}[a \in \mathcal{U}_i] - \delta \lVert (X_i)_a \rVert_2 {1}[b \in \mathcal{U}_j] - \delta^2 {1}[a \in \mathcal{U}_i \land b \in \mathcal{U}_j \land a \ne b]$
>
> $(\Delta_{ij})^U_{ab} = \delta \lVert (X_j)_b \rVert_2 {1}[a \in \mathcal{U}_i] + \delta \lVert (X_i)_a \rVert_2 {1}[b \in \mathcal{U}_j] + \delta^2 {1}[a \in \mathcal{U}_i \land b \in \mathcal{U}_j \land a \ne b]$
>
> Using these bounds, similar to node classification, we can propagate them through the NTK computation to get the bounds on NTK. Using the NTK bounds, we can apply QPCert to get the certificate for graph classification.
>
> [1] Du et al. Graph neural tangent kernel: Fusing graph neural networks with graph kernels. NeurIPS 2019.
>
> ----
>
> > Given the MILP reformulation, computational costs may scale poorly with large datasets
>
> Indeed, as is true for all deterministic certificates [1] scaling to large datasets is challenging. Thus, we discuss this as an important avenue for future research in the discussion Lines 504-506.
>
> [1] Lie et al. "Sok: Certified robustness for deep neural networks", IEEE S&P 2023
>
> **EDIT:** (25.11.) Corrected Line Numbers

---

> > ### Author Response · Authors · 2024-11-25
> >
> > Dear Reviewer,
> >
> > We want to thank you again for the valuable suggestions that allowed us to improve the quality of our work. As the discussion period closes soon, we would greatly appreciate a response if we have satisfactorily addressed your questions and concerns. If you have any further questions/concerns on which you want additional clarifications, please let us know and we will be happy to provide further answers.
> >
> > Best regards,
> > The Authors

---

> > > ### Comment · Reviewer_PTK3 · 2024-11-25
> > >
> > > I acknowledge reading your response, Thanks a lot for the effort to answear. I keep my score

---

> > > > ### Author Response · Authors · 2024-12-02
> > > >
> > > > We want to thank you for your response and again for the positive review of our paper!

---

### Author Response · Authors · 2024-11-22
**Global response**

We thank all reviewers for their helpful feedback and insightful questions! Based on it, we now include interesting **new results and discussions** in the revised paper. We highlight changes to the original draft in blue and all line numbers in the rebuttal, if not indicated otherwise, refer to the updated draft.

Specifically, the revised paper now includes the following **new experiments**:
- **Three new attacks:** One additional graph poisoning attack [1] and two graph backdoor attacks [2] leading to a thorough evaluation of the tightness of QPCert (App. I.3)
- A **comparison** with **two poisoning defenses**: GNNGuard [3] and Elastic Graph Neural Networks [4] (App. I.7)

Furthermore, the revised paper now includes:
- An **extension** of QPCert **to graph classification** (App. N.5)
- A detailed discussion on **extending** QPCert to structure perturbation (App. N.2)
- Guidelines on the **broader applicability** of QPCert to arbitrary GNN architectures (App. N.4.)
- **Practical implications** of feature perturbations compared to structure perturbations (App. N.3)
- Clarifications on the **applicability** to commonly studied perturbation models and attacks (App. N.1)

[1] Zügner et al. "Adversarial Attacks on Graph Neural Networks via Meta Learning", ICLR 2019
[2] Xing et al. "A Clean-Label Graph Backdoor Attack Method in Node Classificaiton Task", Knowledge-Based Systems 2024
[3] Zhang et al. "GNNGuard: Defending Graph Neural Networks against Adversarial Attacks", NeurIPS 2020
[4] Liu et al. "Elastic Graph Neural Networks", ICML 2021

**EDIT:** (25.11.) Included comparisons to poisoning defenses.

---

### Meta-Review · Area_Chair_xQPU · 2024-12-19

**Metareview:**

This paper tries to design a white-box certificate for Graph Neural Networks (GNNs) against poisoning attacks, including backdoors, with a threat model on the node features. The solution first uses neural tangent kernel to turn a GNN training problem into an SVM problem, and then turns the bilevel optimization problem into a mixed-integer linear program (MILP).  The tightness of the worst-case NTK bounds for GNNs is then analyzed.  Extensive experiments are shown that demonstrate the effectiveness of QPCert, along with the role of graph data and the impact of different architectural components in GNNs.

The paper addresses an important problem.  Along with the rebuttal, it considered quite a number of extensions that arise from the real practice.  My major concern is the computational cost, which reviewer PTK3 raised, but was not sufficiently addressed in the rebuttal.  QPCert needs to solve a MILP to the global optimal (not local optimal), and in the worst case its complexity is exponential in the number of labeled nodes.  In the Cora-ML experiment with 140 labeled nodes, certifying a node can take between one minute and several hours using two CPUs.  In contrast, for CSBM and Cora-MLb which have 80 and 20 labeled nodes respectively, certifying one node typically takes several seconds up to one minute on a single CPU.  So it clearly scales superlinearly.  I think this computational cost needs to be addressed first, before considering other extensions of QPCert. For example, what about turning P(Q) in Theorem 1 into a maximization problem, which doesn’t need to be solved to the global optimal?  I understand strong duality can be tricky here though.

**Additional Comments On Reviewer Discussion:**

The rebuttal has been noted by the reviewers and have been taken into account by the AC in the recommendation of acceptance/rejection.

---

### Decision · Program_Chairs · 2025-01-22

Reject